# Truthfulness of Calibration Measures

**Nika Haghtalab, Mingda Qiao, Kunhe Yang, and Eric Zhao**
University of California, Berkeley
{nika,mingda.qiao,kunheyang,eric.zh}@berkeley.edu

## Abstract

We study calibration measures in a sequential prediction setup. In addition to rewarding accurate predictions (completeness) and penalizing incorrect ones (soundness), an important desideratum of calibration measures is *truthfulness*, a minimal condition for the forecaster not to be incentivized to exploit the system. Formally, a calibration measure is truthful if the forecaster (approximately) minimizes the expected penalty by predicting the conditional expectation of the next outcome, given the prior distribution of outcomes. We conduct a taxonomy of existing calibration measures. Perhaps surprisingly, all of them are far from being truthful. We introduce a new calibration measure termed the *Subsampled Smooth Calibration Error (SSCE)*, which is complete and sound, and under which truthful prediction is optimal up to a constant multiplicative factor. In contrast, under existing calibration measures, there are simple distributions on which a polylogarithmic (or even zero) penalty is achievable, while truthful prediction leads to a polynomial penalty.

## 1 Introduction

Probability forecasting is a central prediction task to a wide range of domains and applications, such as finance, meteorology, and medicine [MW84, DF83, WM68, JOKOM12, KSB21, VCV15, BF⁺02, CAT16]. For forecasts to be useful, a common minimum requirement is that they are *calibrated*, i.e., the predictions are unbiased conditioned on the predicted value. Formally, for a sequence of $T$ binary events, a forecaster who predicts probabilities in $[0, 1]$ is *perfectly calibrated* if for every $\alpha \in [0, 1]$, among the time steps on which $\alpha$ is predicted, an $\alpha$ fraction of the outcomes is indeed 1. Since perfectly calibrated forecasts are often unachievable, *calibration measures* have been introduced to quantify some form of deviation from perfectly calibrated forecasts. Common examples of these measures include the expected calibration error (ECE) [FV98], the smooth calibration error [KF08], and the distance from calibration [BGHN23].

As these calibration measures are commonly used to evaluate the performance of forecasters, it is important that their use encourages forecasters to incorporate the highest quality information available to them (e.g., via their expert knowledge or side information) about the next outcome. This desideratum, formally referred to as *truthfulness*, requires that a calibration measure incentivizes the forecasters to predict truthfully when the true distribution of the next outcome is known to them. Lack of truthfulness can have severe consequences: it serves as a poor measure of quality of forecasts, tempts forecasters to make deliberately biased predictions in order to game the system, and erodes trust in predictions provided by third-party forecasters. *Given the importance of truthfulness, we set out to identify calibration measures that demonstrate truthfulness.*

While truthfulness of calibration measures has not been systematically investigated to date, evidence of the lack of truthfulness of some calibration measures has emerged in recent literature. For example, [FH21, QV21] noted that a forecaster can lower their ECE by predicting according to the past. This observation was applied in the algorithm of [FH21] and motivated the "sidestepping" technique in the lower bound proof of [QV21]. More recently, [QZ24] highlighted a large gap in the truthfulness of a recently proposed calibration measure (called the *distance from calibration* [BGHN23]) by

showing that in a simple setup of predicting i.i.d. outcomes, the truthful forecaster incurs a distance of $\Omega(\sqrt{T})$ from calibration but there is a forecasting algorithm that achieves $\mathrm{polylog}(T)$ distance from calibration. We call this a $\mathrm{polylog}(T)$-$\Omega(\sqrt{T})$ *truthfulness gap*. On the other hand, we say that a calibration measure is $(\alpha, \beta)$-truthful if predicting the next outcome according to its conditional distribution incurs a measure that is no more than $\alpha\mathsf{OPT} + \beta$, where $\mathsf{OPT}$ is the minimum value of the calibration measure achievable by any forecaster. Faced with evidence that some calibration measures suffer from large truthfulness gaps, we will systematically examine the truthfulness (or a gap thereof) of a wide range of calibration measures.

For a truthful calibration measure to also be useful it must distinguish accurate predictions from inaccurate ones. After all, a measure that is uniformly 0 regardless of the quality of predictions is perfectly truthful (formally $(1, 0)$-truthful) but provides no insights into the quality of the predictions. We formalize the minimum requirement for a measure to be useful by its *completeness* and *soundness* when predicting i.i.d. Bernoulli outcomes. The former requires that predicting the outcomes according to the correct parameter of the generating Bernoulli distribution incurs no or $o(T)$ penalty, whereas the latter requires the penalty to be $\Omega(T)$ when predictions systematically deviate from the correct parameter. An equally important feature of a calibration measure is that it defines an ideal that could be asymptotically achieved for all prediction tasks. This is formalized by the existence of forecasting algorithms with an $o(T)$ penalty in the adversarial sequential prediction setting [FV98], where the sequence of outcomes is produced by an adaptive adversary.

With these desiderata in place (namely truthfulness, soundness, completeness, and asymptotic calibration), we ask *whether there are calibration measures that simultaneously satisfy all these criteria?* We answer this question in three parts:

**Part I: We show that existing calibration measures do not simultaneously meet these criteria.** We conduct a taxonomy of several existing calibration measures in terms of their completeness, soundness and truthfulness (formally defined in Section 2). We show that almost all of them have large *truthfulness gaps*: There are simple distributions on which an $O(1)$ (or even zero) penalty is achievable, while truthful predictions lead to a $\mathrm{poly}(T)$ penalty; see Table 1 for details.

Indeed, this lack of truthfulness is not limited to specific or contrived distributions. In the next theorem which we will prove in Appendix B, we strengthen these findings by showing that a commonly used notion of calibration systematically suffers large truthfulness gaps in most forecasting instances.

**Theorem 1.1** (Informal). *For every product distribution with marginals bounded away from 0 and 1, the truthful forecaster incurs $\Omega(\sqrt{T})$ smooth calibration error but there exists a forecasting algorithm that incurs only $\mathrm{polylog}(T)$ smooth calibration error.*

A notable exception in Table 1 is the class of calibration measures induced by *proper scoring rules*, i.e., loss functions for probabilistic predictions that are optimized by truthful forecasts. By definition, these calibration measures are $(1, 0)$-truthful. However, none of them is complete: as we show in Appendix A, even on i.i.d. Bernoulli trials, the optimal and truthful predictions incur an $\Omega(T)$ penalty.

**Part II: We introduce a new calibration measure, called SSCE, that is sound, complete, and approximately truthful.** We do this using a simple adjustment to an existing notion of calibration measure: we *subsample* a subset of the time steps and evaluate the *smooth calibration error* [KF08] on this sampled set only. We call this the *Subsampled Smooth Calibration Error (SSCE)* and formally define it in Section 2. Our main result is that SSCE is $(O(1), 0)$-truthful.

**Theorem 1.2** (Main Theorem). *There exists a universal constant $c > 0$ such that the SSCE is $(c, 0)$-truthful. Furthermore, the SSCE is complete and sound.*

As shown in Table 1, to the best of our knowledge, SSCE is the first calibration measure that simultaneously achieves completeness, soundness, and non-trivial truthfulness.

While our methodology for constructing this calibration measure is simple, the analytical steps required to establish the $(O(1), 0)$-truthfulness guarantee are far from simple. We dedicate most of the main body of this paper to illustrating the proof ideas in a series of warmups to Theorem 1.2.

**Part III: There is a forecasting algorithm that achieves $O(\sqrt{T})$ SSCE even in the adversarial setting.** While our study of truthfulness of calibration measures is necessarily focused on when the

| Calibration Measure | Complete? | Sound? | Truthful? |
|---|---|---|---|
| Expected Calibration Error, Maximum Swap Regret | ✓ | ✓ | $0$-$\Omega(T)$ gap |
| Smooth Calibration, Distance from Calibration, Interval Calibration, Laplace-Kernel Calibration | ✓ | ✓ | $0$-$\Omega(\sqrt{T})$ gap |
| U-Calibration Error | ✓ | ✓ | $O(1)$-$\Omega(\sqrt{T})$ gap |
| Proper Scoring Rules | × | ✓ | $(1,0)$-truthful |
| **Subsampled Smooth Calibration Error** | ✓ | ✓ | $(O(1),0)$-truthful |

Table 1: Evaluation of existing calibration measures along with SSCE, in terms of completeness, soundness and truthfulness (Definitions 2.2 and 2.5). An $\alpha$-$\beta$ truthfulness gap means that there is a prediction instance on which forecasting according to the true conditional distribution of the next outcome incurs more than $\beta$ penalty, but there is a forecasting strategy that incurs at most $\alpha$ penalty. See Appendix A for more details.

forecaster knows the conditional distribution of the next outcome, it is important to ensure that, even in the adversarial setting, a sublinear penalty can be achieved for this calibration measure. For this, we study the sequential calibration setting (e.g., [FV98]) where the outcome at time $t$ is chosen by an adaptive adversary who has observed the sequence of earlier outcomes and predictions. We show that an $O(\sqrt{T})$ SSCE is achievable.

**Theorem 1.3.** *In the adversarial sequential calibration setting, there is a deterministic strategy for the forecaster that achieves an $O(\sqrt{T})$ SSCE.*

An interesting and important feature of this result is that it achieves an $O(\sqrt{T})$ rate whereas an $O(\sqrt{T})$ rate for the expected calibration error is known to be impossible to achieve [QV21]. Together our Theorems 1.2 and 1.3 establish that SSCE is a truthful and useful calibration measure.

## 1.1 Related Work

There is a large body of work on calibration, a notion that dates back to the 1950s [Bri50, Daw82, Daw85] and has been applied to game theory [FV97, HPY23], machine learning [GPSW17], and algorithmic fairness [KMR17, PRW+17, HJKRR18, HJZ23]. We will restrict our discussion to sequential calibration and the systematic study of calibration measures, which are the closest to this work.

**Sequential calibration.** Foster and Vohra [FV98] first proved that one can achieve *asymptotic calibration* on arbitrary and adaptive outcomes. Formally, they gave a forecasting algorithm with an $O(T^{2/3})$ ECE in expectation, when predicting $T$ binary outcomes chosen by an adaptive adversary. Subsequent work gave alternative and simpler proofs of the result [FL99, Fos99, Har22], extended the result to other calibration measures [KF08, FH18, FH21, QZ24], and proved lower bounds on the optimal ECE [QV21]. Most closely related to our approach is the work of [FRST11], who studied a stronger notion that requires calibration on a family of *checking rules*, where each checking rule specifies a subset of the time horizon. Despite the apparent similarity, their notion is qualitatively different from the SSCE, since we take an expectation over the subsampled horizon, whereas they take the maximum. In particular, no forecaster can be calibrated in their definition if the checking rule family contains all subsets of $[T]$, since there always exists a checking rule that strongly correlates with the outcomes.

**Calibration measures.** The recent work of Błasiok, Gopalan, Hu and Nakkiran [BGHN23] initiated the rigorous study of calibration measures. Their work focused on the offline setup, where there is a known marginal distribution over the feature space, and each predictor maps the feature space to $[0,1]$. They proposed to use the *distance from calibration*—the $\ell_1$ distance from the predictor to the closest predictor that is perfectly calibrated—as the ground truth, and studied whether existing

calibration measures are consistent with it. Note that completeness and soundness are defined differently in [BGHN23]: a calibration measure is called complete (resp., sound) if it is upper (resp., lower) bounded by a polynomial of the distance from calibration. Since the distance from calibration is far from being truthful in the online setup (as shown by [QZ24]), our definition of completeness and soundness set up minimal conditions for an error metric to be regarded as measuring calibration, rather than enforcing closeness to the distance from calibration.

**Subsampling.** Our new calibration measure is derived from subsampling the time horizon. This simple idea has been shown to be effective in various different contexts, including privacy amplification in differential privacy (e.g.,[Ste22, Section 6]), handling adversarial corruptions [BLMT22], as well as adaptive data analysis [Bla23].

**Proper scoring rules.** Proper scoring rules [WM68] are error metrics for probabilistic forecasts that are optimized when the forecaster predicts according to the true distribution. While the error metrics induced by proper scoring rules are (perfectly) truthful by definition, as we show in Appendix A, they are qualitatively different from the usual calibration measures and, in particular, do not meet the completeness criterion. We note that a recent line of work [CY21, NNW21, LHSW22, PW22, HSLW23] studied the *optimization of scoring rules*, namely, finding the proper scoring rule that maximally incentivizes the forecaster to exert effort to obtain additional information.

## 2  Preliminaries

**Sequential prediction.** We consider the following prediction setup: First, a sequence $x \in \{0,1\}^T$ is sampled from distribution $\mathcal{D}$. At each step $t \in [T]$, the forecaster makes a prediction $p_t \in [0,1]$, after which $x_t$ is revealed. Formally, a deterministic forecaster is a function $\mathcal{A} : \bigcup_{t=1}^T \{0,1\}^{t-1} \to [0,1]$, where $\mathcal{A}(b_1, b_2, \ldots, b_{t-1})$ specifies the forecaster's prediction at step $t$ if the first $t-1$ observations match $b_{1:(t-1)}$. Distribution $\mathcal{D}$ and forecaster $\mathcal{A}$ naturally induce a joint distribution of $(x, p) \in \{0,1\}^T \times [0,1]^T$ via sampling $x \sim \mathcal{D}$ and predicting $p_t = \mathcal{A}(x_1, x_2, \ldots, x_{t-1})$.

Note that we could have defined the forecaster as a function of both the outcomes $x_{1:(t-1)}$ and the predictions $p_{1:(t-1)}$ in the past. This alternative definition is equivalent to ours, since $p_{1:(t-1)}$ would be uniquely determined by $x_{1:(t-1)}$. We could also have considered *randomized* forecasters, which are specified by distributions over deterministic forecasters. However, as we will see later, restricting our attention to deterministic forecasters does not affect the subsequent definitions.

**Calibration measures.** The quality of the forecaster's predictions in the setting above is quantified by calibration measures. Formally, a calibration measure CM is a family of functions $\{\mathsf{CM}_T : T \in \mathbb{N}\}$, where each $\mathsf{CM}_T$ maps $\{0,1\}^T \times [0,1]^T$ to $[0,T]$. We will frequently omit the subscript $T$, since it is usually clear from the context. With respect to calibration measure CM, the expected penalty incurred by forecaster $\mathcal{A}$ on distribution $\mathcal{D}$ is defined as $\mathsf{err}_{\mathsf{CM}}(\mathcal{D}, \mathcal{A}) := \mathbb{E}_{(x,p)\sim(\mathcal{D},\mathcal{A})}[\mathsf{CM}(x,p)]$, where $(x, p) \sim (\mathcal{D}, \mathcal{A})$ denotes sampling a sequence $x$ and predictions $p$ from the joint distribution induced by $\mathcal{D}$ and $\mathcal{A}$.

One example of calibration measures is the *smooth calibration error* introduced by [KF08] that is defined as $\mathsf{smCE}(x,p) := \sup_{f \in \mathcal{F}} \sum_{t=1}^T f(p_t)(x_t - p_t)$, where $\mathcal{F}$ is the family of 1-Lipschitz functions from $[0,1]$ to $[-1,1]$. In this work, we introduce a new calibration measure called *Subsampled Smooth Calibration Error (SSCE)* that is defined by subsampling a subset of the time horizon, and evaluating the smooth calibration error on it. We will formally define this measure next. In the following, $\mathsf{Unif}(S)$ denotes the uniform distribution over a finite set $S$. For a $T$-dimensional vector $x$ and $S \subseteq [T]$, $x|_S$ denotes the $|S|$-dimensional vector formed by the entries of $x$ indexed by $S$.

**Definition 2.1** (Subsampled Smooth Calibration Error). *For a sequence of outcomes $x \in \{0,1\}^T$ and predictions $p \in [0,1]^T$, the* Subsampled Smooth Calibration Error (SSCE) *is defined as*

$$\mathsf{SSCE}(x,p) := \mathop{\mathbb{E}}_{S\sim\mathsf{Unif}(2^{[T]})} [\mathsf{smCE}(x|_S, p|_S)] = \mathop{\mathbb{E}}_{y\sim\mathsf{Unif}(\{0,1\}^T)} \left[ \sup_{f\in\mathcal{F}} \sum_{t=1}^T y_t \cdot f(p_t) \cdot (x_t - p_t) \right].$$

**Completeness and soundness.** We give minimal conditions for a calibration measure to be regarded as complete (intuitively "accurate" predictions have a small penalty) and sound (intuitively "inaccurate" predictions have a large penalty).

**Definition 2.2** (Completeness and soundness)**.** *A calibration measure* $\mathsf{CM}$ *is complete if: (1) For any* $x \in \{0, 1\}^T$, $\mathsf{CM}_T(x, x) = 0$; *(2) For any* $\alpha \in [0, 1]$, $\mathbb{E}_{x_1, \ldots, x_T \sim \mathsf{Bernoulli}(\alpha)} \left[ \mathsf{CM}_T(x, \alpha \cdot \vec{1}_T) \right] = o_\alpha(T)$. *The calibration measure is sound if: (1) For any* $x \in \{0, 1\}^T$, $\mathsf{CM}_T(x, \vec{1}_T - x) = \Omega(T)$; *(2) For any* $\alpha, \beta \in [0, 1]$ *such that* $\alpha \neq \beta$, $\mathbb{E}_{x_1, \ldots, x_T \sim \mathsf{Bernoulli}(\alpha)} \left[ \mathsf{CM}_T(x, \beta \cdot \vec{1}_T) \right] = \Omega_{\alpha, \beta}(T)$. *Here,* $o_\alpha(\cdot)$ *and* $\Omega_{\alpha, \beta}(\cdot)$ *may hide constant factors that depend on the parameters in the subscript.*

**Truthfulness.** To define the truthfulness of a calibration measure, we introduce the *truthful forecaster* and the *optimal error* for a distribution $\mathcal{D}$.

**Definition 2.3** (Truthful forecaster)**.** *With respect to distribution* $\mathcal{D} \in \Delta(\{0, 1\}^T)$, *the truthful forecaster is defined as* $\mathcal{A}^{\mathrm{truthful}}(\mathcal{D})(b_1, b_2, \ldots, b_{t-1}) := \Pr_{x \sim \mathcal{D}} \left[ x_t = 1 \mid x_{1:(t-1)} = b_{1:(t-1)} \right]$.

Arguably, $\mathcal{A}^{\mathrm{truthful}}(\mathcal{D})$ is the only forecaster that makes the "right" predictions on distribution $\mathcal{D}$.

**Definition 2.4** (Optimal error)**.** *The optimal error on distribution* $\mathcal{D} \in \Delta(\{0, 1\}^T)$ *with respect to calibration measure* $\mathsf{CM}$ *is defined as* $\mathsf{OPT}_{\mathsf{CM}}(\mathcal{D}) := \inf_{\mathcal{A}} \mathsf{err}_{\mathsf{CM}}(\mathcal{D}, \mathcal{A})$, *where* $\mathcal{A}$ *ranges over all deterministic forecasters.*

Note that by an averaging argument, the definition of $\mathsf{OPT}_{\mathsf{CM}}(\mathcal{D})$ is unchanged if we take an infimum over randomized forecasters.

A calibration measure is truthful if, on every distribution, the truthful forecaster is near-optimal.

**Definition 2.5** (Truthfulness of calibration measures)**.** *A calibration measure* $\mathsf{CM}$ *is* $(\alpha, \beta)$-*truthful if, for every* $\mathcal{D} \in \Delta(\{0, 1\}^T)$, $\mathsf{err}_{\mathsf{CM}}(\mathcal{D}, \mathcal{A}^{\mathrm{truthful}}(\mathcal{D})) \leq \alpha \cdot \mathsf{OPT}_{\mathsf{CM}}(\mathcal{D}) + \beta$. *Conversely,* $\mathsf{CM}$ *is said to have an* $\alpha$-$\beta$ *truthfulness gap if, for some distribution* $\mathcal{D}$, $\mathsf{OPT}_{\mathsf{CM}}(\mathcal{D}) \leq \alpha$ *and* $\mathsf{err}_{\mathsf{CM}}(\mathcal{D}, \mathcal{A}^{\mathrm{truthful}}(\mathcal{D})) \geq \beta$.

## 3 Technical Overview

In this section, we briefly discuss the main technical ideas and challenges behind the proofs of Theorems 1.1, 1.2, and 1.3. We provide more details on our main result, i.e., that SSCE is $(O(1), 0)$-truthful, in Sections 4 through 6. Theorem 1.3 follows from a recent result of [ACRS24] on minimizing the distance from calibration in the adversarial setup, along with a new result connecting SSCE to distance from calibration, and is proved in Section 7. We defer the proof of Theorem 1.1 to Appendix B.

**A simple distribution that witnesses truthfulness gaps.** Inspired by [QV21, Example 2], we consider the distribution $\mathcal{D}$ specified as follows: The time horizon is divided into $T/3$ blocks of length 3, each with a uniformly random bit, followed by a zero and a one. Within each block, the truthful forecaster predicts $1/2$, 0 and 1 in order. Then, among the steps on which $1/2$ is predicted, the frequency of ones is typically $1/2 \pm \Theta(1/\sqrt{T})$. This deviation results in a $\Theta(\sqrt{T})$ penalty under most calibration measures (concretely, all calibration measures in the first two rows of Table 1).

However, there is a different strategy that ensures perfect calibration, and thus a zero penalty under most calibration measures. Within each block, the forecaster predicts $1/2$ on the first step. If the bit turns out to be 1, the forecaster maintains perfect calibration by predicting $1/2$ on the second step, on which the outcome is known to be 0; otherwise, the forecaster accomplishes the same by predicting $1/2$ on the third step. Therefore, the distribution $\mathcal{D}$ witnesses a $0$-$\Omega(\sqrt{T})$ truthfulness gap for every calibration measure in the first two rows of Table 1.

The importance of subsampling in the SSCE becomes apparent in light of the example above. On distribution $\mathcal{D}$, the truthful forecaster has to pay a $\Theta(\sqrt{T})$ cost for the mild deviation from the expectation, while a strategic forecaster avoids this deviation by correlating the predictions with the biases in the past. With the subsampling, however, the forecaster is no longer sure about the biases that factor into the penalty. This ensures that, compared to truth-telling, the benefit from predicting strategically is marginal, and thus makes the truthfulness guarantee in Theorem 1.2 possible.

**Establishing truthfulness via martingale inequalities.** We prove that the SSCE is $(O(1), 0)$-truthful in three steps: (1) Define a complexity measure $\sigma(\mathcal{D})$ of distribution $\mathcal{D}$; (2) Show that $\mathsf{err}_{\mathsf{SSCE}}(\mathcal{D}, \mathcal{A}^{\mathrm{truthful}}(\mathcal{D})) = O(\sigma(\mathcal{D}))$; (3) Show that $\mathsf{OPT}_{\mathsf{SSCE}}(\mathcal{D}) = \Omega(\sigma(\mathcal{D}))$.

As we elaborate in Section 5, the crux of Step (2) is to control the expected deviation of a martingale $(M_t)_{0 \le t \le T}$ with respect to filtration $(\mathbb{F}_t)$ by the its *realized variance* $\mathrm{Var}_t := \sum_{s=1}^t \mathrm{Var}\,[M_s | \mathbb{F}_{s-1}]$, which is highly non-trivial as the two processes $(M_t)$ and $(\mathrm{Var}_t)$ are correlated. In more detail, the filtration $(\mathbb{F}_t)$ corresponds to the randomness in $x \sim \mathcal{D}$, while $(M_t)$ tracks the biases in the predictions (on a subset of the time horizon) tested by a Lipschitz function. We note that such a bound would easily follow from "off-the-shelf" concentration inequalities for martingales (e.g., Freedman's inequality [Fre75]), if the total realized variance $\mathrm{Var}_T$ were uniformly bounded. However, in general, $\mathrm{Var}_T$ may vary drastically, and directly applying these concentration inequalities would introduce an extra super-constant factor. Our workaround is a "doubling trick" that divides the time horizon into *epochs*, the realized variances in which grow exponentially. We then apply Freedman's inequality to each epoch separately. In Section 5, we formulate a toy random walk problem that highlights this challenge and demonstrates our solution to it, which is of independent interest.

Similarly, as we show in Section 6, the crux of Step (3) is to establish another martingale inequality. We first show that for fixed $x$ and $p$, we have $\mathsf{SSCE}(x, p) = \Omega(\sqrt{N_T})$, where $N_t := \sum_{s=1}^t \mathbb{1}\left[|x_s - p_s| \ge 1/2\right]$. Furthermore, over the randomness in $x \sim \mathcal{D}$, the realized variance process $(\mathrm{Var}_t)$ defined above is shown to lower bound $(N_t)$, i.e., $(N_t - \mathrm{Var}_t)$ is a sub-martingale. However, the desired result requires the lower bound $\mathbb{E}\left[\sqrt{N_T}\right] \ge \Omega(1) \cdot \mathbb{E}\left[\sqrt{\mathrm{Var}_T}\right]$, which does *not* follow from $\mathbb{E}\left[N_T - \mathrm{Var}_T\right] \ge 0$ in general. This challenge necessitates a more careful analysis tailored to the specific properties of the processes $(N_t)$ and $(\mathrm{Var}_t)$.

**Deterministic forecasting strategy via reduction to** $\mathsf{smCE}$**.** We build on the result of [ACRS24] showing the existence of a deterministic forecasting strategy guaranteeing an $O(\sqrt{T})$ bound on $\mathsf{smCE}$. In particular, we show via a standard chaining argument that $\mathsf{SSCE}$ is upper bounded by $\mathsf{smCE}$ plus a variance term that can be upper bounded by $O(\sqrt{T})$. The result of [ACRS24] then implies a deterministic forecasting algorithm achieving an $O(\sqrt{T})$ $\mathsf{SSCE}$.

## 4 Warmup: The Product Distribution Case

As a warmup, in this section, we start by showing that $\mathsf{SSCE}$ is $(O(1), O(\log T))$-truthful for product distributions. This is a weaker version of Theorem 1.2 in terms of both the truthfulness parameters of $\mathsf{SSCE}$ and the restriction to product distributions. In Sections 5 and 6, we outline how we will remove these restrictions and improve the analysis of truthfulness.

For distribution $\mathcal{D} = \prod_{t=1}^T \mathsf{Bernoulli}(p_t^\star)$, take $\sigma^2 := \mathrm{Var}_{x \sim \mathcal{D}}\left[\sum_{t=1}^T x_t\right] = \sum_{t=1}^T p_t^\star(1 - p_t^\star)$ as a complexity measure of the distribution of outcomes. We will show that $\mathsf{err}_{\mathsf{SSCE}}(\mathcal{D}, \mathcal{A}^{\mathrm{truthful}}(\mathcal{D})) = O(\sigma + \log T)$ and $\mathsf{OPT}_{\mathsf{SSCE}}(\mathcal{D}) = \Omega(\sigma) - O(1)$.

### 4.1 Upper Bound the SSCE of the Truthful Forecaster

We first show that the truthful forecaster for $\mathcal{D}$, which predicts $p_t = p_t^\star$ at every step $t$, gives $\mathbb{E}_{x \sim \mathcal{D}}\left[\mathsf{SSCE}(x, p^\star)\right] = O(\sigma + \log T)$. For this purpose, it suffices to prove

$$\mathbb{E}_{x \sim \mathcal{D}}\left[\mathsf{smCE}(x, p^\star)\right] = O(\sigma + \log T), \tag{1}$$

since for each fixed $S \subseteq [T]$, applying (1) to $x|_S$ and $p^\star|_S$ gives $\mathbb{E}_{x \sim \mathcal{D}}\left[\mathsf{smCE}(x|_S, p^\star|_S)\right] \le O(\sigma + \log T)$, and taking an expectation over $S \sim \mathsf{Unif}(2^{[T]})$ gives the desired bound on $\mathsf{SSCE}$.

Recall that $\mathbb{E}\left[\mathsf{smCE}(x, p^\star)\right] = \mathbb{E}\left[\sup_{f \in \mathcal{F}} \sum_{t=1}^T f(p_t^\star) \cdot (x_t - p_t^\star)\right]$. If we replace $\mathcal{F}$ with the family of *constant* functions from $[0, 1]$ to $[-1, 1]$, the right-hand side would reduce to

$$\mathbb{E}_{x \sim \mathcal{D}}\left[\left|\sum_{t=1}^T (x_t - p_t^\star)\right|\right] \le \sqrt{\mathbb{E}_{x \sim \mathcal{D}}\left[\left(\sum_{t=1}^T (x_t - p_t^\star)\right)^2\right]} = \sqrt{\mathrm{Var}_{x \sim \mathcal{D}}\left[\sum_{t=1}^T x_t\right]} = \sigma.$$

Therefore, to prove the upper bound in (1), we need to show that the family of one-dimensional Lipschitz functions is not significantly richer than constant functions.

At a high level, this is done by taking finite coverings of Lipschitz functions and using Dudley's chaining technique [Dud87] to upper bound the value of this stochastic process. In more detail, let $\mathcal{F}_\delta$ be the smallest $\delta$-covering of $\mathcal{F}$ in the uniform norm, i.e., for each $f \in \mathcal{F}$, there exists $f_\delta \in \mathcal{F}_\delta$ such that $\|f - f_\delta\|_\infty \leq \delta$. It is well-known that $|\mathcal{F}_\delta| = e^{O(1/\delta)}$, and a chaining argument gives

$$\mathbb{E}_{x \sim \mathcal{D}} \left[ \sup_{f \in \mathcal{F}} \sum_{t=1}^{T} f(p_t^\star) \cdot (x_t - p_t^\star) \right] \leq 1 + \sum_{k=0}^{O(\log T)} \mathbb{E}_{x \sim \mathcal{D}} \left[ \max_{g \in \mathcal{G}_{2^{-k}}} \sum_{t=1}^{T} g(p_t^\star) \cdot (x_t - p_t^\star) \right], \quad (2)$$

where $\mathcal{G}_\delta := \{f_\delta - f_{\delta/2} : f_\delta \in \mathcal{F}_\delta, f_{\delta/2} \in \mathcal{F}_{\delta/2}, \|f_\delta - f_{\delta/2}\|_\infty \leq 3\delta/2\}$.

It remains to bound the second term of (2). Note that for a fixed $g$, because of the independence of $x_t$s, $g(p_t^\star) \cdot (x_t - p_t^\star)$ is independent across $t \in [T]$. Therefore, we can control the tail probability of $\sum_{t=1}^{T} g(p_t^\star) \cdot (x_t - p_t^\star)$ by Bernstein inequalities. For each fixed $\delta$, using a Bernstein tail bound, taking a union bound over $g \in \mathcal{G}_\delta$, and noting that $|\mathcal{G}_\delta| \leq |\mathcal{F}_\delta| \cdot |\mathcal{F}_{\delta/2}| = e^{O(1/\delta)}$, we have

$$\mathbb{E}_{x \sim \mathcal{D}} \left[ \max_{g \in \mathcal{G}_\delta} \sum_{t=1}^{T} g(p_t^\star) \cdot (x_t - p_t^\star) \right] \leq O(\delta) \cdot O\left( \sqrt{\sigma^2 \log |\mathcal{G}_\delta|} + \log |\mathcal{G}_\delta| \right) = O(\sigma\sqrt{\delta} + 1).$$

Plugging this into (2) proves (1) and thus the desired bound $\mathbb{E}_{x \sim \mathcal{D}} [\mathsf{SSCE}(x, p^\star)] = O(\sigma + \log T)$.

### 4.2 Lower Bound the Optimal SSCE

Next, we lower bound $\mathsf{OPT}_{\mathsf{SSCE}}(\mathcal{D})$ by showing that *every* forecasting strategy must incur an $\Omega(\sigma)$ SSCE on $\mathcal{D}$. Recall that $\mathsf{SSCE}(x, p)$ is given by

$$\mathbb{E}_{y \sim \mathsf{Unif}(\{0,1\}^T)} \left[ \sup_{f \in \mathcal{F}} \sum_{t=1}^{T} y_t \cdot f(p_t) \cdot (x_t - p_t) \right] \geq \mathbb{E}_{y \sim \mathsf{Unif}(\{0,1\}^T)} \left[ \left| \sum_{t=1}^{T} y_t \cdot (x_t - p_t) \right| \right],$$

where we use the fact that $\mathcal{F}$ contains the constant functions $1$ and $-1$.

Fix $x \in \{0,1\}^T$, $p \in [0,1]^T$ and let $N := \sum_{t=1}^{T} \mathbb{1}[|x_t - p_t| \geq 1/2]$. Over the randomness in $y \sim \mathsf{Unif}(\{0,1\}^T)$, the quantity $\sum_{t=1}^{T} y_t \cdot (x_t - p_t)$, by the central limit theorem, is approximately distributed as a normal distribution with variance $\sum_{t=1}^{T} \frac{1}{4}(x_t - p_t)^2 \geq \sum_{t=1}^{T} \frac{1}{16} \mathbb{1}[|x_t - p_t| \geq 1/2] = \Omega(N)$, so its expected absolute value is $\Omega(\sqrt{N})$.

Now it remains to lower bound the expectation of $\sqrt{N}$ induced by an arbitrary forecaster. Conditioning on $x_{1:(t-1)}$, $x_t$ always follows $\mathsf{Bernoulli}(p_t^\star)$. Thus, regardless of the choice of $p_t \in [0,1]$, the condition $|x_t - p_t| \geq 1/2$ holds with probability at least $\min\{p_t^\star, 1 - p_t^\star\} \geq p_t^\star(1 - p_t^\star)$. Then, over the $T$ steps, we expect that $N \geq \Omega(\sum_{t=1}^{T} p_t^\star(1 - p_t^\star)) = \Omega(\sigma^2)$ holds with probability $\Omega(1)$, as long as $\sigma = \Omega(1)$. This gives the desired lower bound $\mathbb{E}[\mathsf{SSCE}(x, p)] \gtrsim \mathbb{E}\left[\sqrt{N}\right] = \Omega(\sigma) - O(1)$.

## 5 Upper Bound the SSCE of the Truthful Forecaster

To extend the proof strategy sketched in Section 4 to non-product distributions, the first challenge is to define an appropriate complexity measure of a general distribution $\mathcal{D}$. Consider the stochastic process $(\mathsf{Var}_t)_{0 \leq t \leq T}$ defined as $\mathsf{Var}_t := \sum_{s=1}^{t} p_s^\star(1 - p_s^\star)$, where $x \sim \mathcal{D}$ and $p_t^\star := \mathbb{E}_{x' \sim \mathcal{D}}\left[x_t' \middle| x_{1:(t-1)}' = x_{1:(t-1)}\right]$ is now a random variable that denotes the conditional expectation of $x_t$ after observing $x_{1:(t-1)}$. The "right" definition turns out to be roughly $\sigma(\mathcal{D}) := \mathbb{E}\left[\sqrt{\mathsf{Var}_T}\right]$. In this section, we prove the following weaker upper bound on the SSCE incurred by the truthful forecaster. We provide a stronger bound (Theorem C.1) in Appendix C.

**Theorem 5.1.** *For any $\mathcal{D} \in \Delta(\{0,1\}^T)$, $\mathsf{err}_{\mathsf{SSCE}}(\mathcal{D}, \mathcal{A}^{\mathrm{truthful}}(\mathcal{D})) = O(\mathbb{E}\left[\sqrt{\mathsf{Var}_T}\right] + \log^2 T)$.*

*Proof sketch.* We begin by repeating the chaining argument in Section 4. Recall that, for any $\delta > 0$, there is a $\delta$-covering $\mathcal{F}_\delta$ of $\mathcal{F}$ in the $\infty$-norm that has size $e^{O(1/\delta)}$. Letting $\pi_\delta(f)$ denote the mapping

of a function $f$ onto the covering $\mathcal{F}_\delta$ such that $\|f - \pi_\delta(f)\|_\infty \leq \delta$, we can write for any $M \in \mathbb{Z}_+$:

$$\mathsf{SSCE}(x,p) \leq 2^{-M} \cdot T + \mathop{\mathbb{E}}_{y \sim \mathsf{Unif}(\{0,1\}^T)} \left[ \sum_{k=0}^{M} \underbrace{\sup_{f \in \mathcal{F}} \sum_{t=1}^{T} y_t \cdot (\pi_{2^{-k}}(f)(p_t) - \pi_{2^{1-k}}(f)(p_t)) \cdot (x_t - p_t)}_{=:W_k} \right].$$

To control the expectation of each $W_k$, we note that the set $\mathcal{G}_k := \{\pi_{2^{-k}}(f) - \pi_{2^{1-k}}(f) : f \in \mathcal{F}\}$ is of size at most $|\mathcal{F}_{2^{-k}}| \cdot |\mathcal{F}_{2^{1-k}}|$. Furthermore, every function $g \in \mathcal{G}_k$ satisfies

$$\|g\|_\infty = \|\pi_{2^{-k}}(f) - \pi_{2^{1-k}}(f)\|_\infty \leq \|\pi_{2^{-k}}(f) - f\|_\infty + \|f - \pi_{2^{1-k}}(f)\|_\infty = O(2^{-k})$$

for some $f \in \mathcal{F}$. We apply the following technical lemma, which we prove in Appendix C.

**Lemma 5.2.** *Given a function $f : [0,1] \rightarrow [-1,1]$ and $y \in \{0,1\}^T$, consider the martingale $M_t(f,y) := \sum_{s=1}^{t} y_s \cdot f(p_s^\star) \cdot (x_s - p_s^\star)$ where $x \sim \mathcal{D}$. Then, for any finite family $\mathcal{G}$ of functions from $[0,1]$ to $[-1,1]$ and any $y \in \{0,1\}^T$, we have*

$$\mathop{\mathbb{E}}_{x \sim \mathcal{D}} \left[ \max_{f \in \mathcal{G}} M_T(f,y) \right] \leq O \left( \log |\mathcal{G}| \cdot \log T + \sqrt{\log |\mathcal{G}|} \cdot \mathop{\mathbb{E}}_{x \sim \mathcal{D}} \left[ \sqrt{\mathrm{Var}_T} \right] \right).$$

Applying Lemma 5.2 to each $\mathcal{G}_k$ scaled up by a $\Theta(2^k)$ factor and noting that $\log |\mathcal{G}_k| \leq \log |\mathcal{F}_{2^{-k}}| + \log |\mathcal{F}_{2^{1-k}}| = O(2^k)$ gives

$$\mathsf{err}_{\mathsf{SSCE}}(\mathcal{D}, \mathcal{A}^{\mathrm{truthful}}(\mathcal{D})) \leq 2^{-M} \cdot T + \sum_{k=0}^{M} O(2^{-k}) \cdot O \left( 2^k \log T + 2^{k/2} \mathop{\mathbb{E}}_{x \sim \mathcal{D}} \left[ \sqrt{\mathrm{Var}_T} \right] \right)$$

$$\leq 2^{-M} \cdot T + \sum_{k=0}^{M} O \left( \log T + 2^{-k/2} \mathop{\mathbb{E}}_{x \sim \mathcal{D}} \left[ \sqrt{\mathrm{Var}_T} \right] \right)$$

$$\leq 2^{-M} \cdot T + O \left( M \log T + \mathop{\mathbb{E}}_{x \sim \mathcal{D}} \left[ \sqrt{\mathrm{Var}_T} \right] \right).$$

Choosing $M = \Theta(\log T)$ proves the theorem. $\qquad\square$

We remark that the proof of Lemma 5.2 is highly non-trivial. As mentioned in Section 3, such an upper bound would follow from Freedman's inequality, if $\mathrm{Var}_T$ were *always* bounded by $O\left( \left( \mathbb{E} \left[ \sqrt{\mathrm{Var}_T} \right] \right)^2 \right)$. However, in general, applying Freedman's inequality to each $M_T(f,y)$ necessarily requires an additional union bound over possible values of $\mathrm{Var}_T$, and introduces a super-constant multiplicative factor.

The challenge in dealing with the randomness in $\mathrm{Var}_T$ is captured by the following toy problem:

> **Random walk with early stopping:** Let $(X_t)_{0 \leq t \leq T}$ be the random walk such that $X_0 = 0$ and each $X_t - X_{t-1}$ independently follows $\mathsf{Unif}(\{\pm 1\})$. Let $\tau$ be an arbitrary stopping time with respect to $(X_t)$. Prove that $\mathbb{E}[|X_\tau|] \leq O(1) \cdot \mathbb{E}[\sqrt{\tau}]$.

Indeed, the above corresponds to a special case of Lemma 5.2 in which: (1) the sequence $p^\star$ starts with entry $1/2$, and may switch to entry $0$ at any point, depending on the realization of $x_t$s; (2) the family $\mathcal{G}$ consists of two constant functions $1$ and $-1$.

One might be tempted to prove $\mathbb{E}[|X_\tau|] \leq O(1) \cdot \mathbb{E}[\sqrt{\tau}]$ by first proving $\mathbb{E}[|X_\tau| | \tau = t] = O(\sqrt{t})$ for all $t \in [T]$, and then applying the law of total expectation. Such an approach is doomed to fail, because the stopping time $\tau$ might significantly bias the conditional expectation of $|X_\tau|$ on some event $\tau = t_0$, e.g., by stopping at time $t_0$ only if $|X_{t_0}| \gg \sqrt{t_0}$.

Our workaround is inspired by the standard doubling trick in online learning. We break the time horizon into *epochs* of geometrically increasing lengths: the $k$-th epoch contains $2^k$ steps. We break $|X_\tau|$ into the displacements accumulated in different epochs; their sum clearly upper bounds $|X_\tau|$.

Furthermore, we can show that, conditioning on reaching epoch $k$, the displacement within the epoch is $O(\sqrt{2^k})$. This allows us to establish the desired inequality via

$$\mathbb{E}\left[|X_\tau|\right] \leq O(1) \cdot \sum_{k=1}^{O(\log T)} \Pr\left[\tau \text{ reaches epoch } k\right] \cdot \sqrt{2^k} \leq O(1) \cdot \mathbb{E}\left[\sqrt{\tau}\right].$$

To prove Lemma 5.2, we extend this technique to a general martingale $M_T(f, y)$ by dividing the time horizon into epochs according to the doubling of $\mathrm{Var}_t$, and then applying Freedman's inequality to each epoch.

**Towards a stronger upper bound.** In our actual proof, we use a slightly different complexity measure $\sigma_\gamma(\mathcal{D}) := \mathbb{E}\left[\gamma(\mathrm{Var}_T)\right]$, where $\gamma(x) = x$ if $x < 1$ and $\gamma(x) = \sqrt{x}$ otherwise. Roughly speaking, this definition accounts for the fact that a sum of independent Bernoulli random variables behaves quite differently when its mean is close to $0$. To remove the extra $\log^2 T$ term in Theorem 5.1, our actual proof also uses a variant of Lemma 5.2, Lemma C.9, which involves a more careful application of Freedman's inequality tailored to specific coverings of Lipschitz functions.

## 6 Lower Bound the Optimal SSCE

In this section, we outline a weaker lower bound on the optimal SSCE achievable on a distribution.

**Theorem 6.1.** *For any $\mathcal{D} \in \Delta(\{0, 1\}^T)$, $\mathsf{OPT}_{\mathsf{SSCE}}(\mathcal{D}) = \Omega(\mathbb{E}\left[\sqrt{\mathrm{Var}_T}\right]) - O(1)$.*

Similar to the product distribution case (Section 4), the key quantity in the proof is the stochastic process $(N_t)_{0 \leq t \leq T}$ defined as $N_t := \sum_{s=1}^t n_s$ and $n_t := \mathbb{1}\left[|x_t - p_t| \geq 1/2\right]$. This is formalized by the following lemma, which applies to any realization of $x$, $p$, and $N_T = \sum_{t=1}^T \mathbb{1}\left[|x_t - p_t| \geq 1/2\right]$:

**Lemma 6.2.** *For any $x \in \{0, 1\}^T$ and $p \in [0, 1]^T$, we have $\mathsf{SSCE}(x, p) \geq \Omega\left(\sqrt{N_T}\right)$.*

It remains to lower bound the quantity $\mathbb{E}\left[\sqrt{N_T}\right]$ induced by an arbitrary forecaster. As argued earlier, conditioning on $x_{1:(t-1)}$, we always have $\Pr\left[n_t = 1\right] \geq p_t^\star(1 - p_t^\star) = \mathrm{Var}_t - \mathrm{Var}_{t-1}$, where $p_t^\star$ and $\mathrm{Var}_t$ are defined as in Section 5. Thus, $(N_t - \mathrm{Var}_t)$ is a sub-martingale, which implies $\mathbb{E}\left[N_T\right] \geq \mathbb{E}\left[\mathrm{Var}_T\right]$. However, this does *not* imply that $\mathbb{E}\left[\sqrt{N_T}\right] \geq \Omega(\mathbb{E}\left[\sqrt{\mathrm{Var}_T}\right])$. In fact, such an inequality does *not* hold in general: When $p_1^\star = \varepsilon \ll 1$ and $p_t^\star = 0$ for all $t \geq 2$, $\mathbb{E}\left[\sqrt{N_T}\right]$ could be $O(\varepsilon)$, yet $\mathbb{E}\left[\sqrt{\mathrm{Var}_T}\right] = \Omega(\sqrt{\varepsilon}) \gg O(\varepsilon)$.

The following technical lemma circumvents this counterexample by subtracting a constant term from the right-hand side:

**Lemma 6.3.** *The stochastic process $(N_t)_{t \in [T]}$ satisfies $\mathbb{E}\left[\sqrt{N_T}\right] \geq \Omega(\mathbb{E}\left[\sqrt{\mathrm{Var}_T}\right]) - O(1)$.*

Note that Theorem 6.1 directly follows from Lemmas 6.2 and 6.3, which we prove in Appendix D.1. To avoid the extra $-O(1)$ term in the lower bound, our actual proof (deferred to Appendix D.3) works with the slightly different complexity measure $\sigma_\gamma(\mathcal{D}) := \mathbb{E}\left[\gamma(\mathrm{Var}_T)\right]$ defined in Section 5.

## 7 Forecasting with $O(\sqrt{T})$ SSCE

In this section, we prove Theorem 1.3, which states the existence of a deterministic forecaster that incurs an $O(\sqrt{T})$ SSCE against all adaptive adversaries. Recall the definition of the smooth calibration error (smCE) from Section 2. Using standard chaining arguments, we can show the following relation between SSCE and smCE, whose proof we defer to Appendix F.

**Lemma 7.1.** *For any $x \in \{0, 1\}^T$ and $p \in [0, 1]^T$,*

$$\mathsf{SSCE}(x, p) \leq \frac{1}{2}\mathsf{smCE}(x, p) + O(\sqrt{T}),$$

*where the $O(\cdot)$ notation hides a universal constant that does not depend on $T$, $x$ or $p$.*

Theorem 1.3 follows from the lemma above and a recent result of [ACRS24].

*Proof of Theorem 1.3.* It was shown by [ACRS24] that there exists a deterministic forecaster with an $O(\sqrt{T})$ distance from calibration ($\mathsf{CalDist}(x,p)$) against every adaptive adversary in the adversarial sequential calibration setup. Lemma 7.1 together with the inequality $\frac{1}{2}\mathsf{smCE}(x,p) \leq \mathsf{CalDist}(x,p)$ from [BGHN23, Lemma 5.4 and Theorem 7.3] implies that

$$\mathsf{SSCE}(x,p) \leq \mathsf{CalDist}(x,p) + O(\sqrt{T}),$$

so the same forecaster incurs an SSCE of $O(\sqrt{T})$ as well. □

## 8  Discussion

We formulate three natural desiderata of calibration measures that evaluate the quality of probabilistic forecasts: truthfulness, completeness, and soundness. They serve as minimal requirements for an error metric to be considered as measuring calibration and not to create a significant incentive for forecasters to predict untruthfully. While existing calibration measures fail to simultaneously meet all these criteria, we propose the new calibration measure (SSCE) that is shown to be approximately truthful via a non-trivial analysis. In the following, we discuss two natural directions of future work.

**Inherent trade-offs among different desiderata?**  As shown in Table 1, the SSCE and the error metrics induced by proper scoring rules give a trade-off between truthfulness and completeness: The former is complete and approximately truthful, while the latter is perfectly truthful but not complete. Is there a calibration measure that achieves the best of both worlds? Taking a step back, while our definition of truthfulness seems natural, the completeness and soundness criteria, as defined, only serve as minimal requirements. It still remains to explore ways to formally quantify the latter two, and investigate the inherent quantitative trade-offs among truthfulness, completeness and soundness.

**Truthfulness against adaptive adversaries?**  One may wonder whether the truthfulness guarantee of SSCE can be extended to handle *adaptive* adversaries as well. Assuming that the forecaster is given an adversary's (randomized) strategy for choosing $x_t$ based on $x_{1:(t-1)}$ and $p_{1:(t-1)}$, is it still approximately optimal to always predict the conditional probability? Here, "adaptive" emphasizes that $x_t$ may depend on both $x_{1:(t-1)}$ and $p_{1:(t-1)}$; the formulation in Section 2 is equivalent to that $x_t$ only depends on $x_{1:(t-1)}$.

Unfortunately, as we show in Appendix G, such a guarantee does not hold for SSCE, and is unlikely to hold for any natural calibration measure: An adversary can "force" the forecaster to predict untruthfully by "threatening" to increase the variance of the subsequent bits. However, this adversary is highly contrived and unrealistic for practical scenarios. We may thus identify reasonable restrictions on the adaptive adversary to sidestep this counterexample.

## 9  Acknowledgements

This work is supported in part by the National Science Foundation under grants CCF-2145898 and the Graduate Research Fellowship Program under grant DGE 2146752, the Office of Naval Research under grant N00014-24-1-2159, an Alfred P. Sloan fellowship, a Schmidt Sciences AI2050 fellowship, and a Google Research Scholars award. Any opinions, findings, and conclusions or recommendations expressed in this material are those of the author(s) and do not necessarily reflect the views of the funding agencies.

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

| Calibration Measure | Complete? | Sound? | Truthful? |
|---|:---:|:---:|:---:|
| Expected Calibration Error | ✓ | ✓ | $0$-$\Omega(T)$ gap |
| Maximum Swap Regret | ✓ | ✓ | $0$-$\Omega(T)$ gap |
| Smooth Calibration Error | ✓ | ✓ | $0$-$\Omega(\sqrt{T})$ gap |
| Distance from Calibration | ✓ | ✓ | $0$-$\Omega(\sqrt{T})$ gap |
| Interval Calibration Error | ✓ | ✓ | $0$-$\Omega(\sqrt{T})$ gap |
| Laplace-Kernel Calibration Error | ✓ | ✓ | $0$-$\Omega(\sqrt{T})$ gap |
| U-Calibration Error | ✓ | ✓ | $O(1)$-$\Omega(\sqrt{T})$ gap |
| Proper Scoring Rules | × | ✓ | $(1,0)$-truthful |
| smCE $+ \sqrt{T}$ | × | ✓ | $(O(1),0)$-truthful |
| **Subsampled Smooth Calibration Error** | ✓ | ✓ | $(O(1),0)$-truthful |

Table 2: Evaluation of previous calibration measures along with SSCE, in terms of completeness, soundness and truthfulness (Definitions 2.2 and 2.5). Every calibration measure, except SSCE, either lacks completeness or has a significant truthfulness gap.

## A  Taxonomy of Existing Calibration Measures

In this section, we prove that the existing calibration measures in Table 2 either have a large truthfulness gap or lack completeness.

In these proofs, the *biases* induced by specific outcomes and predictions will be frequently used: With respect to outcomes $x \in \{0,1\}^T$ and predictions $p \in [0,1]^T$, the bias associated with value $\alpha \in [0,1]$ is defined as

$$\Delta_\alpha := \sum_{t=1}^{T}(x_t - p_t) \cdot \mathbb{1}\left[p_t = \alpha\right].$$

### A.1  Existing Calibration Measures

**The expected calibration error.**  A common calibration measure is the sum of $L_1$ errors of each level set, known as the $L_1$ calibration error or the *Expected Calibration Error (ECE)*: On $x \in \{0,1\}^T$ and $p \in [0,1]^T$, the expected calibration error is defined as

$$\mathsf{ECE}(x,p) := \sum_{\alpha \in [0,1]} \left| \sum_{t=1}^{T}(x_t - p_t) \cdot \mathbb{1}[p_t = \alpha] \right| = \sum_{\alpha \in [0,1]} |\Delta_\alpha|.$$

Note that the summand $|\Delta_\alpha|$ is non-zero only if $\alpha \in \{p_1, p_2, \ldots, p_T\}$, so the summations above are essentially finite and well-defined.

**The smooth calibration error.**  The *smooth calibration error* [KF08] is defined as

$$\mathsf{smCE}(x,p) := \sup_{f \in \mathcal{F}} \sum_{t=1}^{T} f(p_t) \cdot (x_t - p_t) = \sup_{f \in \mathcal{F}} \sum_{\alpha \in [0,1]} f(\alpha) \cdot \Delta_\alpha,$$

where $\mathcal{F}$ is the family of 1-Lipschitz functions from $[0,1]$ to $[-1,1]$. Again, since $\Delta_\alpha \neq 0$ holds only if $\alpha \in \{p_1, p_2, \ldots, p_T\}$, the summation above is finite and well-defined.

**The distance from calibration.** The *distance from calibration*, introduced by [BGHN23] and extended to the sequential setup by [QZ24], is defined as:

$$\mathsf{CalDist}(x, p) := \min_{q \in \mathcal{C}(x)} \|p - q\|_1,$$

where

$$\mathcal{C}(x) := \left\{ p \in [0,1]^T : \forall a \in [0,1], \sum_{t=1}^{T} (x_t - p_t) \cdot \mathbb{1}[p_t = \alpha] = 0 \right\}$$

is the set of predictions that are perfectly calibrated for $x$.

**Interval calibration.** The *interval calibration error* of [BGHN23] relaxes the ECE to a binned version while penalizing the use of long intervals. Formally, an interval partition $\mathcal{I}$ is a finite collection of intervals $\{I_1, I_2, \ldots, I_{|\mathcal{I}|}\}$ that form a partition of $[0,1]$. The interval calibration error is defined as:

$$\mathsf{intCE}(x, p) := \inf_{\mathcal{I}} \left[ \sum_{i=1}^{|\mathcal{I}|} \left| \sum_{t=1}^{T} (x_t - p_t) \cdot \mathbb{1}[p_t \in I_i] \right| + \sum_{t=1}^{T} \sum_{i=1}^{|\mathcal{I}|} \mathsf{len}(I_i) \cdot \mathbb{1}[p_t \in I_i] \right],$$

where the infimum is over all interval partitions $\mathcal{I}$, and $\mathsf{len}(I)$ denotes the length of interval $I$. Note that the first summation inside the infimum is analogous to the ECE, except that the biases associated with all values within the same interval are added together. The second summation gives the total lengths of the intervals into which the $T$ predictions fall.

**Laplace-kernel calibration.** The *Laplace-kernel calibration error* [BGHN23] is a special case of the *maximum mean calibration error* introduced by [KSJ18]. It can be viewed as a variant of the smooth calibration error, in which the family $\mathcal{F}$ of Lipschitz functions is replaced by

$$\widetilde{\mathcal{F}} := \left\{ f : \mathbb{R} \to \mathbb{R} : \|f\|_2^2 + \|f'\|_2^2 \leq 1 \right\},$$

where $\| \cdot \|_2$ denotes the $\ell_2$ norm of functions, and $f'$ is the derivative of $f$. Namely,

$$\mathsf{kCE}^{\mathsf{Lap}}(x, p) := \sup_{f \in \widetilde{\mathcal{F}}} \sum_{t=1}^{T} f(p_t) \cdot (x_t - p_t).$$

**U-calibration.** The definition of the *U-calibration error* [KLST23] is based on *proper scoring rules*. A (bounded) scoring rule is a function $S : \{0,1\} \times [0,1] \to [-1,1]$. A scoring rule is proper if it holds for every $\alpha \in [0,1]$ that

$$\alpha \in \arg\min_{\beta \in [0,1]} \mathbb{E}_{x \sim \mathsf{Bernoulli}(\alpha)} [S(x, \beta)].$$

In other words, when the outcome $x$ is drawn from follow $\mathsf{Bernoulli}(\alpha)$, predicting the true parameter $\alpha$ minimizes the expected loss. The U-calibration error is then defined as

$$\mathsf{UCal}(x, p) := \sup_{S} \left[ \sum_{t=1}^{T} S(x_t, p_t) - \inf_{\alpha \in [0,1]} \sum_{t=1}^{T} S(x_t, \alpha) \right],$$

where the supremum is over all proper scoring rules. Note that for each fixed $S$, the expression inside the supremum is exactly the *external regret* of the forecaster, i.e., the excess loss compared to the best fixed prediction in hindsight.

**Maximum swap regret.** A recent line of work [NRRX23, RS24, HW24] considers a strengthening of U-calibration, in which the external regret is replaced with the *swap regret*. In particular, [HW24] showed that the resulting calibration measure, termed the Maximum Swap Regret (MSR), is polynomially related to the ECE after scaling by a factor of $1/T$:

$$\left[ \frac{\mathsf{ECE}(x, p)}{T} \right]^2 \leq \frac{\mathsf{MSR}(x, p)}{T} \leq \frac{2\mathsf{ECE}(x, p)}{T}.$$

## A.2  0-$\Omega(T)$ Truthfulness Gaps

We first prove the 0-$\Omega(T)$ truthfulness gaps of the ECE and the MSR.

**Proposition A.1.** *Both the expected calibration error and the maximum swap regret have a 0-$\Omega(T)$ truthfulness gap.*

To establish Proposition A.1, we follow a similar argument to the one in Section 3: We divide the time horizon into $T/3$ triples, each containing a random bit followed by a zero and a one. The truthful forecaster would predict the true probabilities for the $T/3$ random bits, which are designed to be close to $1/2$ but distinct. This leads to a linear ECE. On the other hand, a strategic forecaster may always predict $1/2$ on the random bit. Then, based on the realization of the random bit, they use the subsequent deterministic bits to offset the bias. The resulting predictions are perfectly calibrated, and thus have a zero ECE. Finally, the relation between the ECE and the MSR gives the same truthfulness gap for the MSR.

*Proof of Proposition A.1.* Consider the distribution $\mathcal{D}$ defined as follows:

- Let $\varepsilon_1, \varepsilon_2, \ldots, \varepsilon_{T/3}$ be distinct values in $[-1/4, 1/4]$ chosen arbitrarily.

- For each $k \in [T/3]$, set $(p_{3k-2}^\star, p_{3k-1}^\star, p_{3k}^\star) = (1/2 + \varepsilon_k, 0, 1)$.

- $\mathcal{D}$ is the product distribution $\prod_{t=1}^{T} \mathsf{Bernoulli}(p_t^\star)$.

By definition, the predictions made by the truthful forecaster are exactly given by $p^\star$. Then, for each $k \in [T/3]$ and $\alpha = 1/2 + \varepsilon_k \in [1/4, 3/4]$, we have $|\Delta_\alpha| = |x_{3k-2} - \alpha| \geq 1/4$. This shows $\mathsf{err}_{\mathsf{ECE}}(\mathcal{D}, \mathcal{A}^{\mathrm{truthful}}(\mathcal{D})) \geq (T/3) \cdot (1/4) = \Omega(T)$. By the inequality $\frac{\mathsf{MSR}(x,p)}{T} \geq \left[\frac{\mathsf{ECE}(x,p)}{T}\right]^2$, we also have $\mathsf{err}_{\mathsf{MSR}}(\mathcal{D}, \mathcal{A}^{\mathrm{truthful}}(\mathcal{D})) = \Omega(T)$.

On the other hand, consider the following alternative forecaster for $\mathcal{D}$:

- For each $k \in [T/3]$, predict $p_{3k-2} = 1/2$.

- If $x_{3k-2} = 0$, predict $p_{3k-1} = 0$ and $p_{3k} = 1/2$; otherwise, predict $p_{3k-1} = 1/2$ and $p_{3k} = 1$.

Clearly, for each $k \in [T/3]$, the steps $t \in \{3k - 2, 3k - 1, 3k\}$ have zero contribution to $\Delta_0$, $\Delta_1$ and $\Delta_{1/2}$. Therefore, this forecaster achieves a zero ECE on $\mathcal{D}$. This proves $\mathsf{OPT}_{\mathsf{ECE}}(\mathcal{D}) = 0$ and establishes the 0-$\Omega(T)$ truthfulness gap for the ECE. Finally, the inequality $\frac{\mathsf{MSR}(x,p)}{T} \leq \frac{2\mathsf{ECE}(x,p)}{T}$ implies that the same forecaster achieves a zero MSR, which establishes $\mathsf{OPT}_{\mathsf{MSR}}(\mathcal{D}) = 0$ and the 0-$\Omega(T)$ truthfulness gap for the MSR. $\square$

## A.3  0-$\Omega(\sqrt{T})$ Truthfulness Gaps

Next, we prove the 0-$\Omega(\sqrt{T})$ truthfulness gap for several calibration measures. The proof follows the argument outlined in Section 3.

**Proposition A.2.** *The smooth calibration error, the distance from calibration, the interval calibration error, and the Laplace-kernel calibration error all have a 0-$\Omega(\sqrt{T})$ truthfulness gap.*

*Proof.* The truthfulness gaps of the four calibration measures are witnessed by the same product distribution $\mathcal{D} = \prod_{t=1}^{T} \mathsf{Bernoulli}(p_t^\star)$, where $(p_{3k-2}^\star, p_{3k-1}^\star, p_{3k}^\star) = (1/2, 0, 1)$ for every $k \in [T/3]$.

**Truthful forecaster has an $\Omega(\sqrt{T})$ penalty.**  The truthful forecaster makes predictions that are identical to $p^\star$. As a result, we have $\Delta_0 = \Delta_1 = 0$, while $\Delta_{1/2}$ is distributed as the difference between a sample from $\mathsf{Binomial}(T/3, 1/2)$ and its mean $T/6$. It then follows that $|\Delta_{1/2}| \geq \Omega(\sqrt{T})$ holds with probability $\Omega(1)$. We will show that all four calibration measures evaluate to $\Omega(\sqrt{T})$ in expectation.

For the smooth calibration error, we have

$$\mathsf{err}_{\mathsf{smCE}}(\mathcal{D}, \mathcal{A}^{\mathrm{truthful}}(\mathcal{D})) = \underset{x \sim \mathcal{D}}{\mathbb{E}} \left[ |\Delta_{1/2}| \right] = \underset{X \sim \mathsf{Binomial}(T/3, 1/2)}{\mathbb{E}} [|X - T/6|] = \Omega(\sqrt{T}).$$

For the distance from calibration, by [BGHN23, Lemma 5.4 and Theorem 7.3], we have the inequality $\frac{1}{2}\mathsf{smCE}(x, p) \leq \mathsf{CalDist}(x, p)$ for any $x \in \{0, 1\}^T$ and $p \in [0, 1]^T$, so the truthful forecaster also gives $\mathsf{err}_{\mathsf{CalDist}}(\mathcal{D}, \mathcal{A}^{\mathrm{truthful}}(\mathcal{D})) = \Omega(\sqrt{T})$.

For interval calibration, let $\mathcal{I}$ be an arbitrary interval partition, and $I \in \mathcal{I}$ be the interval that contains $1/2$. If $I$ contains either $0$ or $1$, we must have $\mathsf{len}(I) \geq 1/2$, and the term $\sum_{t=1}^{T} \sum_{i=1}^{|\mathcal{I}|} \mathsf{len}(I_i) \cdot \mathbb{1}[p_t \in I_i]$ will be at least $2T/3 \cdot 1/2 = \Omega(T)$. If $I$ does not contain $0$ or $1$, the summation $\sum_{t=1}^{T}(x_t - p_t) \cdot \mathbb{1}[p_t \in I]$ will be exactly $\Delta_{1/2}$, and the first term in the definition will be at least $|\Delta_{1/2}|$. It follows that $\mathsf{intCE}(x, p) \geq \Omega(\sqrt{T})$ with probability $\Omega(1)$, so we have the lower bound $\mathsf{err}_{\mathsf{intCE}}(\mathcal{D}, \mathcal{A}^{\mathrm{truthful}}(\mathcal{D})) = \Omega(\sqrt{T})$.

For Laplace-kernel calibration, let $f_0$ be an arbitrary function in $\widetilde{\mathcal{F}}$ such that $f_0(1/2) > 0$, e.g., we can take $f_0(x) = ce^{-x^2}$ for a sufficiently small constant $c > 0$. Then, we have

$$\mathsf{kCE}^{\mathsf{Lap}}(x, p) \geq \sup_{f \in \{f_0, -f_0\}} \sum_{\alpha \in [0,1]} f(\alpha) \cdot \Delta_\alpha \geq \Omega(1) \cdot |\Delta_{1/2}|.$$

It follows that $\mathsf{err}_{\mathsf{kCE}^{\mathsf{Lap}}}(\mathcal{D}, \mathcal{A}^{\mathrm{truthful}}(\mathcal{D})) \geq \Omega(1) \cdot \mathbb{E}\left[|\Delta_{1/2}|\right] = \Omega(\sqrt{T})$.

**Strategic forecaster has a zero penalty.** Consider the same strategic forecaster as in the proof of Proposition A.1: For each $k \in [T/3]$,

- Predict $p_{3k-2} = 1/2$.

- If $x_{3k-2} = 0$, predict $(p_{3k-1}, p_{3k}) = (0, 1/2)$; otherwise, predict $(p_{3k-1}, p_{3k}) = (1/2, 1)$.

Clearly, this guarantees that $\Delta_\alpha = 0$ holds for all $\alpha \in [0, 1]$. By definition, we have $\mathsf{OPT}_{\mathsf{smCE}}(\mathcal{D}) = \mathsf{OPT}_{\mathsf{CalDist}}(\mathcal{D}) = 0$. It also easily follows that both $\mathsf{intCE}$ and $\mathsf{kCE}^{\mathsf{Lap}}$ evaluate to $0$. For $\mathsf{intCE}$, we consider the interval partition $\mathcal{I} = \{\{0\}, (0, 1/2), \{1/2\}, (1/2, 1), \{1\}\}$, which witnesses $\mathsf{intCE}(x, p) = 0$. For $\mathsf{kCE}^{\mathsf{Lap}}$, the summation $\sum_{t=1}^{T} f(p_t) \cdot (x_t - p_t) = \sum_{\alpha \in [0,1]} f(\alpha) \cdot \Delta_\alpha$ evaluates to $0$ for all $f \in \widetilde{\mathcal{F}}$. This proves $\mathsf{OPT}_{\mathsf{intCE}}(\mathcal{D}) = \mathsf{OPT}_{\mathsf{kCE}^{\mathsf{Lap}}}(\mathcal{D}) = 0$. $\qquad\square$

## A.4 $O(1)$-$\Omega(\sqrt{T})$ Truthfulness Gap of U-Calibration

For the U-calibration error, we prove a slightly smaller truthfulness gap of $O(1)$-$\Omega(\sqrt{T})$, via a more involved analysis.

**Proposition A.3.** *The U-calibration error has an $O(1)$-$\Omega(\sqrt{T})$ truthfulness gap.*

*Proof.* We use a slightly different construction: the product distribution $\mathcal{D} = \prod_{t=1}^{T} \mathsf{Bernoulli}(p_t^\star)$ where $p_t^\star = 1/2$ for $t \leq T/2$ and $p_t^\star = 1$ for $t > T/2$.

**Truthful forecaster has an $\Omega(\sqrt{T})$ penalty.** We first show that the truthful forecaster has an $\Omega(\sqrt{T})$ U-calibration error. Let random variable $X := \sum_{t=1}^{T/2} x_t$ denote the number of ones among the first $T/2$ random bits. Note that $X$ follows $\mathsf{Binomial}(T/2, 1/2)$. Consider the scoring rule defined as:

$$S(0, \alpha) = \mathsf{sgn}(\alpha - 1/2) \quad \text{and} \quad S(1, \alpha) = \mathsf{sgn}(1/2 - \alpha).$$

Note that $S$ is proper, since for any $\alpha \in [0, 1]$, we have

$$\underset{x \sim \mathsf{Bernoulli}(\alpha)}{\mathbb{E}} [S(x, \beta)] = (1 - \alpha) \cdot \mathsf{sgn}(\beta - 1/2) + \alpha \cdot \mathsf{sgn}(1/2 - \beta) = (1 - 2\alpha) \cdot \mathsf{sgn}(\beta - 1/2),$$

which is always minimized at $\beta = \alpha$.

The total loss (w.r.t. $S$) incurred by the forecaster is then

$$\sum_{t=1}^{T} S(x_t, p_t) = X \cdot S(1, 1/2) + (T/2 - X) \cdot S(0, 1/2) + T/2 \cdot S(1, 1)$$
$$= X \cdot 0 + (T/2 - X) \cdot 0 + T/2 \cdot (-1)$$
$$= -T/2.$$

On the other hand, the total loss incurred by a fixed prediction $\beta \in [0, 1]$ is given by:

$$\sum_{t=1}^{T} S(x_t, \beta) = (T/2 + X) \cdot S(1, \beta) + (T/2 - X) \cdot S(0, \beta)$$
$$= (T/2 + X) \cdot \mathrm{sgn}(1/2 - \beta) + (T/2 - X) \cdot \mathrm{sgn}(\beta - 1/2)$$
$$= 2X \cdot \mathrm{sgn}(1/2 - \beta).$$

By choosing $\beta = 1$, we can obtain a total loss of $-2X$. Therefore, whenever $X \geq T/4$, we have

$$\mathsf{UCal}(x, p) \geq -T/2 - (-2X) = 2(X - T/4).$$

When $X < T/4$, we always have $\mathsf{UCal}(x, p) \geq 0$, since the trivial scoring rule $S \equiv 0$ is proper. This shows that the truthful forecaster gives

$$\mathsf{err}_{\mathsf{UCal}}(\mathcal{D}, \mathcal{A}^{\mathrm{truthful}}(\mathcal{D})) \geq \underset{X \sim \mathsf{Binomial}(T/2, 1/2)}{\mathbb{E}} [\max\{2(X - T/4), 0\}] = \Omega(\sqrt{T}).$$

**Strategic forecaster with an $O(1)$ penalty.** We consider an alternative forecaster $\mathcal{A}$, which is slightly more involved:

- At every step $t \leq T/2$, predict $p_t = 5/8$.

- For $t = T/2 + 1, T/2 + 2, \ldots, T$, predict $p_t = 5/8$ until $|\Delta_{5/8}| \leq 1$ at some time $t$. After that step, predict $p_t = 1$.

We first argue that the condition $|\Delta_{5/8}| \leq 1$ must hold at some point. Recall that $X = \sum_{t=1}^{T/2} x_t$. By a Chernoff bound, $X$ falls into $[T/8, 5T/16]$ except with probability $e^{-\Omega(T)}$. Assuming this, we have $\Delta_{5/8} = X - (T/2) \cdot (5/8) \leq 0$ at time $t = T/2$. Furthermore, if we hypothetically predict $5/8$ for each of the last $T/2$ steps, we would have

$$\Delta_{5/8} = (X + T/2) - T \cdot (5/8) \geq T/8 + T/2 - 5T/8 = 0$$

after all the $T$ steps. Since $\Delta_{5/8}$ changes by at most $1$ at each step, we must hit the condition $|\Delta_{5/8}| \leq 1$ at some point.

Therefore, except with an $e^{-\Omega(T)}$ probability, we end up with $\Delta_{5/8} \in [-1, 1]$. Furthermore, we predict at most two different values: $5/8$ and $1$. For every fixed proper scoring rule $S : \{0, 1\} \times [0, 1] \to [-1, 1]$, we have

$$\sum_{t=1}^{T} S(x_t, p_t) - \inf_{\beta \in [0,1]} \sum_{t=1}^{T} S(x_t, \beta)$$
$$\leq \sum_{\alpha \in \{5/8, 1\}} \left[ \sum_{t=1}^{T} S(x_t, p_t) \cdot \mathbb{1}[p_t = \alpha] - \inf_{\beta \in [0,1]} \sum_{t=1}^{T} S(x_t, \beta) \cdot \mathbb{1}[p_t = \alpha] \right].$$

In the above, we divide the time horizon $[T]$ into two parts, based on whether $5/8$ or $1$ is predicted. The inequality holds since the right-hand side allows different values of $\beta$ for different parts. Clearly, the term corresponding to $\alpha = 1$ has zero contribution, since it reduces to $S(1, 1) - \inf_{\beta \in [0,1]} S(1, \beta)$ times the number of times $1$ is predicted, which evaluates to $0$ by definition of proper scoring rules.

The term corresponding to $\alpha = 5/8$, on the other hand, is given by

$$N_0 \cdot S(0, 5/8) + N_1 \cdot S(1, 5/8) - \inf_{\beta \in [0,1]} [N_0 \cdot S(0, \beta) + N_1 \cdot S(1, \beta)],$$

where each $N_b$ denotes the number of steps on which $5/8$ is predicted and the outcome is $b \in \{0, 1\}$. By definition of proper scoring rules, the infimum is achieved by $\beta^\star = \frac{N_1}{N_0 + N_1}$, and the above can be further simplified into

$$(N_0 + N_1) \cdot [S(\beta^\star, 5/8) - S(\beta^\star, \beta^\star)],$$

where $S(\alpha, \beta) := \alpha \cdot S(1, \beta) + (1 - \alpha) \cdot S(0, \beta)$ is the linear extension of $S$ to $[0, 1]^2$.

Let $\ell(\alpha) := S(\alpha, \alpha)$ denote the *uni-variate form* of $S$. The following is a standard fact about proper scoring rules (see e.g., [KLST23, Lemma 1 and Corollary 2]).

**Lemma A.4.** *For any proper scoring rule $S : [0, 1]^2 \to [-1, 1]$ and its uni-variate form $\ell : [0, 1] \to [-1, 1]$, it holds for all $\alpha, \beta \in [0, 1]$ that*

- $S(\alpha, \beta) = \ell(\beta) + (\alpha - \beta) \cdot \ell'(\beta)$

- $|\ell'(\alpha)| \leq 2$ *for all $\alpha \in [0, 1]$.*

In particular, we have

$$|S(\beta^\star, 5/8) - S(5/8, 5/8)| = |\beta^\star - 5/8| \cdot \ell'(5/8) \leq 2|\beta^\star - 5/8|$$

and

$$|S(5/8, 5/8) - S(\beta^\star, \beta^\star)| = |\ell(5/8) - \ell(\beta^\star)| \leq 2|\beta^\star - 5/8|.$$

It follows that, assuming $X \in [T/8, 5T/16]$,

$$\mathsf{UCal}(x, p) \leq 4(N_0 + N_1)|\beta^\star - 5/8| = 4 \left| N_1 - \frac{5}{8}(N_0 + N_1) \right| = 4|\Delta_{5/8}| \leq 4.$$

When $X \in [T/8, 5T/16]$ does not hold (which happens with probability $e^{-\Omega(T)}$), the U-calibration error is trivially upper bounded by $O(T)$. It follows that

$$\mathsf{OPT}_{\mathsf{UCal}}(\mathcal{D}) \leq \mathsf{err}_{\mathsf{UCal}}(\mathcal{D}, \mathcal{A}) \leq 4 + O(T) \cdot e^{-\Omega(T)} = O(1).$$

$\square$

## A.5 Lack of Completeness

Every scoring rule $S : \{0, 1\} \times [0, 1] \to [0, 1]$ induces a calibration measure $\mathsf{CM}^{(S)}(x, p) := \sum_{t=1}^{T} S(x_t, p_t)$.[1] When $S$ is proper, it is easy to show that the resulting $\mathsf{CM}^{(S)}$ is perfectly truthful, i.e., $(1, 0)$-truthful.

A drawback of such calibration measures is that they all lack completeness. Concretely, consider the squared loss $S(x, p) := (x - p)^2$. When the outcomes $x_1, x_2, \ldots, x_T$ are independent and uniformly random bits, the "right" prediction $p_t \equiv 1/2$ gives a total penalty of $T/4$, which is only a constant factor away from the maximum possible penalty of $T$. This violates the completeness property in Definition 2.2. In contrast, as shown in Table 2, almost all the other calibration measures would evaluate to $\ll T$ in this case. Such an *asymptotic gap* better justifies the intuition that $p_t \equiv 1/2$ is a much better prediction than, say, $p_t \equiv 0$.

More generally, unless the proper scoring rule $S$ is trivial, we may find $(x_0, p_0) \in \{0, 1\} \times (0, 1)$ such that $S(x_0, p_0) > 0$. Then, on a sequence of independent samples from $\mathsf{Bernoulli}(p_0)$, we have

$$\mathbb{E}_{x_1, \ldots, x_T \sim \mathsf{Bernoulli}(p_0)} \left[ \mathsf{CM}_T^{(S)}(x, p_0 \cdot \vec{1}) \right] \geq T \cdot S(x_0, p_0) \cdot \Pr_{X \sim \mathsf{Bernoulli}(p_0)} [X = x_0]$$

$$\geq T \cdot S(x_0, p_0) \cdot \min\{p_0, 1 - p_0\} = \Omega(T),$$

which violates the completeness condition in Definition 2.2.

We also note that $\mathsf{smCE}(x, p) + \sqrt{T}$ gives a calibration measure that is trivially truthful: Implicit in the proof of [QZ24, Theorem 3], the truthful forecaster gives an $O(\sqrt{T})$ smooth calibration error on every distribution $\mathcal{D} \in \Delta(\{0, 1\}^T)$, so it immediately gives a constant approximation of the optimal

---

[1]Here, we consider scoring rules with co-domain $[0, 1]$, since our definition of calibration measures (in Section 2) requires them to be bounded between 0 and $T$ on length-$T$ sequences.

error, which is at least $\sqrt{T}$. However, this metric is not complete in the sense of Definition 2.2, since it evaluates to $\sqrt{T}$ instead of 0 when $p = x$ (i.e., the predictions are binary and perfect). While SSCE also discourages the forecaster from "over-optimizing" the metric by introducing some additional noise, the subsampling procedure is arguably more "organic" and better-justified than adding a $\sqrt{T}$ term.

# B  Proof of Theorem 1.1

We prove Theorem 1.1, which we formally restate below.

**Theorem B.1** (Formal version of Theorem 1.1). *For every $p^\star \in [0,1]^T$, on the product distribution $\mathcal{D} = \prod_{t=1}^T \mathsf{Bernoulli}(p_t^\star)$, there is a forecaster that achieves an $O(\log^{3/2} T)$ smooth calibration error and distance from calibration. Moreover, assuming that $p^\star \in [\delta, 1-\delta]^T$ for a fixed constant $\delta \in (0, 1/2]$, both $\mathsf{OPT}_{\mathsf{smCE}}(\mathcal{D})$ and $\mathsf{OPT}_{\mathsf{CalDist}}(\mathcal{D})$ are $\Omega(\sqrt{T})$.*

## B.1  The Upper Bound Part

We start by proving the upper bound part of Theorem B.1 by designing a forecasting algorithm.

**The forecasting algorithm.**  Our proof is based on an algorithm of [QZ24] that works for the special case that $p_t^\star \equiv 1/2$. Their algorithm starts by predicting $1/2$ on the first $T/2$ steps. Depending on the realization of these $T/2$ random bits, it predicts a slightly biased value for the next $T/2$ steps, until the total bias (i.e., the partial sum of $x_t - p_t$) becomes close to 0 at some point. If there is still time left, the algorithm repeats the above strategy for the remainder of the time horizon.

Roughly speaking, [QZ24] shows that a $\mathrm{polylog}(T)$ distance from calibration can be achieved by designing a sub-routine with the following three properties:

- **Small bias:** With high probability, the total bias is $O(1)$ in magnitude at some time $t \in [T/2, T]$.
- **Proximity of predictions:** During the sub-routine, the values being predicted lie in a short interval of length $\mathrm{polylog}(T)/\sqrt{T}$.
- **Sparsity of predictions:** During the sub-routine, only $O(1)$ different values are predicted.

To handle the general case that $p^\star \in [0,1]^T$ is arbitrary, we design an alternative sub-routine, the behavior of which depends on whether the sequence $p^\star$ is "sufficiently stationary" in some sense. Let $\mu_{\mathrm{first}} := \frac{1}{T/2} \sum_{t=1}^{T/2} p_t^\star$ and $\mu_{\mathrm{second}} := \frac{1}{T/2} \sum_{t=T/2+1}^{T} p_t^\star$ be the averages of the first and the second halves of the sequence, respectively. Let $\mu = (\mu_{\mathrm{first}} + \mu_{\mathrm{second}})/2$ be the overall average.

- **Case 1:** $|\mu_{\mathrm{first}} - \mu| > \mathrm{polylog}(T)/\sqrt{T}$. When $\mu_{\mathrm{first}}$ and $\mu$ are far away, we predict $\alpha := \frac{\mu_{\mathrm{first}} + \mu}{2}$ at every step. Without loss of generality, suppose that $\mu_{\mathrm{first}} < \mu$, in which case we have
$$\mu_{\mathrm{first}} < \alpha < \mu,$$
where both inequalities hold with a margin $> \mathrm{polylog}(T)/\sqrt{T}$. Then, with high probability the following two events happen: (1) The total bias is negative at time $T/2$, i.e., it holds that $\sum_{t=1}^{T/2} x_t < \alpha \cdot (T/2)$; (2) If we (hypothetically) predict the same value $\alpha$ for the second half, the bias will be positive in the end with high probability, i.e., $\sum_{t=1}^{T} x_t > \alpha \cdot T$. Therefore, with high probability, the bias must be close to 0 at some point in $[T/2, T]$. In this case, this sub-routine has all the desired properties.

- **Case 2:** $|\mu_{\mathrm{first}} - \mu| \leq \mathrm{polylog}(T)/\sqrt{T}$. When $\mu_{\mathrm{first}}$ and $\mu$ are close, we use a strategy that is more similar to the algorithm of [QZ24]. For the first half of the sequence, we predict $\alpha := \mu_{\mathrm{first}}$. Let $\Delta_{\mathrm{first}} := \sum_{t=1}^{T/2} (x_t - \alpha)$ denote the total bias at time $T/2$. Say that $\Delta_{\mathrm{first}} \geq 0$. Then, we will predict $\beta := \mu_{\mathrm{second}} + \frac{\Delta_{\mathrm{first}}}{T/2} + \frac{\mathrm{polylog}(T)}{\sqrt{T}}$ in the second half of the sequence. The value of $\beta$ is chosen such that we can offset the bias incurred in the first half (i.e., the $\Delta_{\mathrm{first}}/(T/2)$ term). We also introduce some additional bias (i.e.,

the $\mathrm{polylog}(T)/\sqrt{T}$ term), so that we can return to a zero bias with high probability. In this case, our sub-routine predicts two different values ($\alpha$ and $\beta$), and they only differ by $\mathrm{polylog}(T)/\sqrt{T}$ with high probability.

Formally, our algorithm is given in Algorithm 1. The actual algorithm is significantly more involved than the outline above. The complication is due to the constraint that all predictions must lie in $[0, 1]$, while our choice of $\beta$ in Case 2 above might be invalid. We circumvent this issue by noting that $\beta$ can be invalid only if $\mu_{\mathrm{first}}$ is too close to either 0 or 1. In that case, we will choose a different value of $\alpha$ (i.e., the prediction for the first half), so that the sign of the bias at time $T/2$ is more predictable, and the resulting choice of $\beta$ will likely be valid.

---

**Algorithm 1:** Forecaster for Product Distributions

**Input:** Parameters $p_1^\star, p_2^\star, \ldots, p_T^\star$. Outcomes $x_1, x_2, \ldots, x_T$ observed sequentially.
**Output:** Predictions $p_1, p_2, \ldots, p_T$.

1   $t \leftarrow 0;\ r \leftarrow 0$;
2   **while** $t < T$ **do**
3     $r \leftarrow r + 1;\ T^{(r)} \leftarrow T - t;\ H^{(r)} \leftarrow \lfloor T^{(r)}/2 \rfloor$;
4     **if** $T^{(r)} = 1$ **then** predict $p_T = 0$ and **break**;
5     $\mu_{\mathrm{first}}^{(r)} \leftarrow \frac{1}{H^{(r)}} \sum_{s=t+1}^{t+H^{(r)}} p_s^\star;\ \mu_{\mathrm{second}}^{(r)} \leftarrow \frac{1}{H^{(r)}} \sum_{s=t+H^{(r)}+1}^{t+2H^{(r)}} p_s^\star$;
6     $\mu^{(r)} \leftarrow [\mu_{\mathrm{first}}^{(r)} + \mu_{\mathrm{second}}^{(r)}]/2;\ \Delta^{(r)} \leftarrow 0$;
7     **if** $|\mu_{\mathrm{first}}^{(r)} - \mu^{(r)}| \geq \sqrt{\frac{2 \ln T^{(r)}}{H^{(r)}}}$ **then**
8       $\alpha^{(r)} \leftarrow [\mu_{\mathrm{first}}^{(r)} + \mu^{(r)}]/2$;
9       **for** $i = 1, 2, \ldots, 2H^{(r)}$ **do**
10         $t \leftarrow t + 1$; Predict $p_t \leftarrow \alpha^{(r)}$;
11         Observe $x_t;\ \Delta^{(r)} \leftarrow \Delta^{(r)} + (x_t - p_t)$;
12         **if** $i > H^{(r)}$ *and* $|\Delta^{(r)}| \leq 1$ **then break**;
13       **end**
14     **else**
15       **if** $\mu_{\mathrm{first}}^{(r)} \leq 1/2$ **then**
16         **if** $\mu_{\mathrm{first}}^{(r)} \geq 10\sqrt{\frac{\ln T^{(r)}}{H^{(r)}}}$ **then** $\alpha^{(r)} \leftarrow \mu_{\mathrm{first}}^{(r)}$ ;
17         **else** $\alpha^{(r)} \leftarrow \max\left\{\mu_{\mathrm{first}}^{(r)} - \sqrt{\frac{2\mu_{\mathrm{first}}^{(r)} \ln T^{(r)}}{H^{(r)}}}, 0\right\}$ ;
18       **else**
19         **if** $1 - \mu_{\mathrm{first}}^{(r)} \geq 10\sqrt{\frac{\ln T^{(r)}}{H^{(r)}}}$ **then** $\alpha^{(r)} \leftarrow \mu_{\mathrm{first}}^{(r)}$ ;
20         **else** $\alpha^{(r)} \leftarrow \min\left\{\mu_{\mathrm{first}}^{(r)} + \sqrt{\frac{2[1-\mu_{\mathrm{first}}^{(r)}] \ln T^{(r)}}{H^{(r)}}}, 1\right\}$ ;
21       **for** $i = 1, 2, \ldots, H^{(r)}$ **do**
22         $t \leftarrow t + 1$; Predict $p_t \leftarrow \alpha^{(r)}$;
23         Observe $x_t;\ \Delta^{(r)} \leftarrow \Delta^{(r)} + (x_t - p_t)$;
24       **end**
25       **if** $\Delta^{(r)} \geq 0$ **then** $\beta^{(r)} \leftarrow \min\left\{\mu_{\mathrm{second}}^{(r)} + \Delta^{(r)}/H^{(r)} + \sqrt{\frac{\ln T^{(r)}}{2H^{(r)}}}, 1\right\}$ ;
26       **else** $\beta^{(r)} \leftarrow \max\left\{\mu_{\mathrm{second}}^{(r)} + \Delta^{(r)}/H^{(r)} - \sqrt{\frac{\ln T^{(r)}}{2H^{(r)}}}, 0\right\}$ ;
27       **for** $i = 1, 2, \ldots, H^{(r)}$ **do**
28         $t \leftarrow t + 1$; Predict $p_t \leftarrow \beta^{(r)}$;
29         Observe $x_t;\ \Delta^{(r)} \leftarrow \Delta^{(r)} + (x_t - p_t)$;
30         **if** $|\Delta^{(r)}| \leq 1$ **then break**;
31       **end**
32   **end**

---

**The analysis.** We analyze Algorithm 1 and prove the upper bound in Theorem B.1 in the following three steps:

- First, we break the execution of Algorithm 1 into different rounds of the while-loop, and show that each round brings a $\mathrm{polylog}(T)$ smooth calibration error in expectation.

- Then, using the simple observation that the smooth calibration error is sub-additive, we obtain an upper bound on the overall smooth calibration error.

- Finally, we use a relation between $\mathsf{smCE}(x, p)$ and $\mathsf{CalDist}(x, p)$ when $p$ only contains a few different values (shown by [QZ24]) to translate the upper bound to one on the distance from calibration.

The first step is the most technical. We fix $r$ and condition on the value of $t$ (equivalently, the value of $T^{(r)}$) at the beginning of the $r$-th round. Note that the event $t = t_0$ is solely determined by the realization of $x_1, x_2, \ldots, x_{t_0}$, so conditioning on the value of $t$, the subsequent bits $x_{t+1}$ through $x_T$ are still distributed according to $\mathcal{D}$. Let sequences $x^{(r)}$ and $p^{(r)}$ denote the outcomes and predictions made in the $r$-th round. Note that the two sequences are of the same length, though the length might vary.

We classify the rounds into three different types as follows:

- **Type 1:** The condition $|\mu_{\mathrm{first}}^{(r)} - \mu^{(r)}| \geq \sqrt{\frac{2 \ln T^{(r)}}{H^{(r)}}}$ holds in the if-statement on Line 7.

- **Type 2:** $|\mu_{\mathrm{first}}^{(r)} - \mu^{(r)}| < \sqrt{\frac{2 \ln T^{(r)}}{H^{(r)}}}$, and $\alpha^{(r)}$ is set to $\mu_{\mathrm{first}}^{(r)}$ on either Line 16 or Line 19.

- **Type 3:** $|\mu_{\mathrm{first}}^{(r)} - \mu^{(r)}| < \sqrt{\frac{2 \ln T^{(r)}}{H^{(r)}}}$, and $\alpha^{(r)}$ is not set to $\mu_{\mathrm{first}}^{(r)}$.

Note that for fixed $p^\star$, the type of a round is deterministic given $r$ and $T^{(r)}$.

The three lemmas below give high-probability bounds on the smooth calibration error incurred during each round.

**Lemma B.2.** *Conditioning on the value of $T^{(r)}$, if the $r$-th round is Type 1, it holds with probability $1 - O(1/T^{(r)})$ that*
$$\mathsf{smCE}(x^{(r)}, p^{(r)}) \leq 1.$$

**Lemma B.3.** *Conditioning on the value of $T^{(r)}$, if the $r$-th round is Type 2, it holds with probability $1 - O(1/T^{(r)})$ that*
$$\mathsf{smCE}(x^{(r)}, p^{(r)}) \leq 1 + O\left(\frac{1}{T^{(r)}}\right) \cdot \left[\Delta_{\mathrm{first}}^{(r)}\right]^2 + O\left(\sqrt{\frac{\log T^{(r)}}{T^{(r)}}}\right) \cdot \left|\Delta_{\mathrm{first}}^{(r)}\right|,$$

*where $\Delta_{\mathrm{first}}^{(r)}$ denotes the value of $\Delta^{(r)}$ at the end of the first for-loop (on Line 25).*

**Lemma B.4.** *Conditioning on the value of $T^{(r)}$, if the $r$-th round is Type 3, it holds with probability $1 - O(1/T^{(r)})$ that*
$$\mathsf{smCE}(x^{(r)}, p^{(r)}) \leq 1 + O\left(\frac{1}{T^{(r)}}\right) \cdot \left[\Delta_{\mathrm{first}}^{(r)}\right]^2 + O\left(\sqrt{\frac{\log T^{(r)}}{T^{(r)}}}\right) \cdot \left|\Delta_{\mathrm{first}}^{(r)}\right|,$$

*where $\Delta_{\mathrm{first}}^{(r)}$ denotes the value of $\Delta^{(r)}$ at the end of the first for-loop (on Line 25).*

We first prove the upper bound part of Theorem B.1 using the lemmas above.

*Proof of Theorem B.1, the upper bound part.* By Lemmas B.2 through B.4, regardless of the type of the $r$-th round, it holds with probability $1 - O\left(1/T^{(r)}\right)$ that
$$\mathsf{smCE}(x^{(r)}, p^{(r)}) \leq 1 + O\left(\frac{1}{T^{(r)}}\right) \cdot \left[\Delta_{\mathrm{first}}^{(r)}\right]^2 + O\left(\sqrt{\frac{\log T^{(r)}}{T^{(r)}}}\right) \cdot \left|\Delta_{\mathrm{first}}^{(r)}\right|,$$

where $\Delta_{\text{first}}^{(r)}$ is regarded as 0 if the $r$-th round is Type 1. We say that the round *fails* if this upper bound on smCE does not hold. Conditioning on that $T^{(r)} = L$, we always have $\text{smCE}(x^{(r)}, p^{(r)}) \leq L$, since there are at most $L$ steps in the $r$-th round. Therefore, we have the inequality

$$\text{smCE}(x^{(r)}, p^{(r)}) \leq 1 + O\left(\frac{1}{L}\right) \cdot \left[\Delta_{\text{first}}^{(r)}\right]^2 + O\left(\sqrt{\frac{\log L}{L}}\right) \cdot \left|\Delta_{\text{first}}^{(r)}\right| + L \cdot \mathbb{1}\left[\text{round } r \text{ fails}\right].$$

We will upper bound the value of $\mathbb{E}\left[\text{smCE}(x^{(r)}, p^{(r)})\right]$ by taking an expectation over both sides of the above. Therefore, we examine the expectation of $|\Delta_{\text{first}}^{(r)}|$ and $[\Delta_{\text{first}}^{(r)}]^2$ conditioning on $T^{(r)} = L$.

When the round is Type 1, there is nothing to upper bound. For Type 2 rounds, $\Delta_{\text{first}}^{(r)}$ is the difference between $X_{\text{first}} = \sum_{s=t+1}^{t+H} x_s$ and its mean $\mu_{\text{first}} H$. Since the variance of $X_{\text{first}}$ is $O(L)$, we have $\mathbb{E}\left[\left|\Delta_{\text{first}}^{(r)}\right|\right] = O(\sqrt{L})$ and $\mathbb{E}\left[\left[\Delta_{\text{first}}^{(r)}\right]^2\right] = O(L)$.

Type 3 rounds are trickier. We assume that $\mu_{\text{first}} \leq 1/2$; this is without loss of generality since the $\mu_{\text{first}} > 1/2$ case can be handled by a completely symmetric argument. Then, $\Delta_{\text{first}}^{(r)}$ is the difference between $X_{\text{first}} = \sum_{s=t+1}^{t+H} x_s$ and $\alpha H$, and $\alpha$ may differ from $\mu_{\text{first}}$ by at most $\sqrt{\frac{2\mu_{\text{first}} \ln L}{H}}$. This gives

$$\mathbb{E}\left[\left[\Delta_{\text{first}}^{(r)}\right]^2\right] = \mathbb{E}\left[(X_{\text{first}} - \mu_{\text{first}} H)^2\right] + (\mu_{\text{first}} H - \alpha H)^2$$
$$\leq O(L) + O(\mu_{\text{first}} H \ln L).$$

Now we use the fact that when $\mu_{\text{first}} \leq 1/2$, the round is Type 3 only if $\mu_{\text{first}} < 10\sqrt{\frac{\ln T^{(r)}}{H}}$. This implies

$$O(\mu_{\text{first}} H \ln L) \leq O(\sqrt{L} \cdot \log^{3/2} L),$$

which is dominated by the $O(L)$ term. It then follows from Jensen's inequality that

$$\mathbb{E}\left[\left|\Delta_{\text{first}}^{(r)}\right|\right] \leq \sqrt{\mathbb{E}\left[\left[\Delta_{\text{first}}^{(r)}\right]^2\right]} = O(\sqrt{L}).$$

**Put everything together.** Therefore, we have the upper bound

$$\mathbb{E}\left[\text{smCE}(x^{(r)}, p^{(r)})\Big| T^{(r)} = L\right]$$

$$\leq 1 + \mathbb{E}\left[O\left(\frac{1}{L}\right) \cdot [\Delta_{\text{first}}^{(r)}]^2 + O\left(\sqrt{\frac{\log L}{L}}\right) \cdot |\Delta_{\text{first}}^{(r)}|\Big| T^{(r)} = L\right] + L \cdot \Pr\left[\text{round } r \text{ fails}\Big| T^{(r)} = L\right]$$

$$\leq 1 + O(\sqrt{\log L}) + L \cdot O(1/L) = O(\sqrt{\log T}).$$

The second step applies our earlier conclusion that $\mathbb{E}\left[|\Delta_{\text{first}}^{(r)}|\right] = O(\sqrt{L})$ and $\mathbb{E}\left[[\Delta_{\text{first}}^{(r)}]^2\right] = O(L)$ conditioning on $T^{(r)} = L$. Taking another expectation over the randomness in $T^{(r)}$ shows that $\text{smCE}(x^{(r)}, p^{(r)}) = O(\sqrt{\log T})$ for every $r$. Note that we have

$$\text{smCE}(x, p) = \sup_{f \in \mathcal{F}} \sum_{t=1}^{T} f(p_t) \cdot (x_t - p_t)$$

$$= \sup_{f \in \mathcal{F}} \sum_{r} \sum_{t} f(p_t^{(r)}) \cdot (x_t^{(r)} - p_t^{(r)})$$

$$\leq \sum_{r} \sup_{f \in \mathcal{F}} \sum_{t} f(p_t^{(r)}) \cdot (x_t^{(r)} - p_t^{(r)})$$

$$= \sum_{r} \text{smCE}(x^{(r)}, p^{(r)}).$$

Furthermore, there are at most $O(\log T)$ rounds. It follows that $\mathbb{E}\left[\mathsf{smCE}(x,p)\right] = O(\log^{3/2} T)$.

Finally, we note that in each round of the while-loop, the forecaster predicts at most 2 different values (namely, $\alpha^{(r)}$ and $\beta^{(r)}$). Therefore, the predictions $p_1, p_2, \ldots, p_T$ contain at most $O(\log T)$ different values. By [QZ24, Theorem 2], we conclude that

$$\mathbb{E}\left[\mathsf{CalDist}(x,p)\right] \leq O(1) \cdot \mathbb{E}\left[\mathsf{smCE}(x,p) + |\{p_1, p_2, \ldots, p_T\}|\right] = O(\log^{3/2} T).$$

$\square$

Now we prove Lemmas B.2 through B.4. In the proofs below, we frequently drop the superscript $(r)$ since we only refer to the $r$-th round.

*Proof of Lemma B.2.* Recall that a Type 1 round is one in which the condition $|\mu_{\mathrm{first}} - \mu| \geq \sqrt{\frac{2\ln T^{(r)}}{H}}$ holds in the if-statement. We say that the round *succeeds*, if we exit the for-loop using the "break" statement on Line 12, i.e., the condition $i > H$ and $|\Delta^{(r)}| \leq 1$ holds at some point (including in the last iteration where $i = 2H$); otherwise, the round *fails*.

Note that only one value (namely, $\alpha^{(r)}$) is predicted within the round. Thus, if the round succeeds, we have

$$\mathsf{smCE}(x^{(r)}, p^{(r)}) = |\Delta_{\alpha^{(r)}}| = \left|\Delta^{(r)}\right| \leq 1.$$

It remains to control the probability for a Type 1 round to fail. Consider random variables

$$X_{\mathrm{first}} := \sum_{s=t+1}^{t+H} x_s \quad \text{and} \quad X := \sum_{s=t+1}^{t+2H} x_s.$$

Note that both are sums of independent Bernoulli random variables, with $\mathbb{E}\left[X_{\mathrm{first}}\right] = \mu_{\mathrm{first}} H$ and $\mathbb{E}\left[X\right] = 2\mu H$. Also note that since $\alpha = (\mu_{\mathrm{first}} + \mu)/2$, we have

$$|\mu_{\mathrm{first}} - \alpha| = |\mu - \alpha| = \frac{1}{2}|\mu_{\mathrm{first}} - \mu| \geq \sqrt{\frac{\ln T^{(r)}}{2H}}.$$

Without loss of generality, suppose that $\mu_{\mathrm{first}} \leq \mu$. By an additive Chernoff bound, we have

$$\Pr\left[X_{\mathrm{first}}/H \geq \alpha\right] \leq \exp\left(-2H\left(\alpha - \mu_{\mathrm{first}}\right)^2\right) \leq \frac{1}{T^{(r)}}.$$

and

$$\Pr\left[X/(2H) \leq \alpha\right] \leq \exp\left(-4H\left(\alpha - \mu\right)^2\right) \leq \frac{1}{T^{(r)}}.$$

Therefore, except with probability $O(1/T^{(r)})$, we have both $X_{\mathrm{first}} < \alpha H$ and $X > 2\alpha H$. In other words, if the for-loop (hypothetically) runs all the $2H$ iterations, we would have $\Delta^{(r)} < 0$ at the end of the $H$-th iteration, and $\Delta^{(r)} > 0$ at the end of the $2H$-th iteration. Since $\Delta^{(r)}$ changes by $|x_t - p_t| \leq 1$ within each iteration, there must be an iteration $i \in \{H+1, H+2, \ldots, 2H\}$ at the end of which $\Delta^{(r)}$ falls into $[0, 1]$. By definition of Algorithm 1, we exit the for-loop at that time, and the $r$-th round succeeds. $\square$

*Proof of Lemma B.3.* Recall that in a Type 2 round, we have $|\mu_{\mathrm{first}} - \mu| < \sqrt{\frac{2\ln T^{(r)}}{H}}$ and $\alpha = \mu_{\mathrm{first}}$. Without loss of generality, suppose that $\mu_{\mathrm{first}} \leq 1/2$; the case that $\mu_{\mathrm{first}} > 1/2$ follows from a completely symmetric argument. We say that a Type 2 round *succeeds* if both conditions below are satisfied:

- When $\beta$ is chosen, the clipping (i.e., taking the minimum with 1 or taking the maximum with 0) is not effective.

- We exit the second for-loop through the break statement on Line 30.

Otherwise, the round *fails*.

Again, we first upper bound the smooth calibration error incurred within a successful round, and then control the probability for a round to fail. Since only $\alpha$ and $\beta$ are predicted in this round, we have

$$\mathsf{smCE}(x^{(r)}, p^{(r)}) = \sup_{f \in \mathcal{F}}[f(\alpha) \cdot \Delta_\alpha + f(\beta) \cdot \Delta_\beta],$$

where $\Delta_\alpha$ and $\Delta_\beta$ are defined with respect to $x^{(r)}$ and $p^{(r)}$. The above is further given by

$$\sup_{f \in \mathcal{F}}[f(\beta) \cdot (\Delta_\alpha + \Delta_\beta) + [f(\alpha) - f(\beta)] \cdot \Delta_\alpha]$$
$$\leq \sup_{f \in \mathcal{F}}[f(\beta) \cdot (\Delta_\alpha + \Delta_\beta)] + \sup_{f \in \mathcal{F}}[(f(\alpha) - f(\beta)) \cdot \Delta_\alpha]$$
$$= |\Delta_\alpha + \Delta_\beta| + |\alpha - \beta| \cdot |\Delta_\alpha|.$$

Note that $\Delta_\alpha + \Delta_\beta$ is exactly the value of $\Delta^{(r)}$ at the end of the second for-loop, while $\Delta_\alpha$ is its value after the first for-loop, i.e., $\Delta_{\mathrm{first}}^{(r)}$. Then, assuming that the round succeeds, we have $|\Delta_\alpha + \Delta_\beta| \leq 1$ and

$$|\alpha - \beta| = |\mu_{\mathrm{first}} - \beta| \leq |\mu_{\mathrm{first}} - \mu_{\mathrm{second}}| + |\mu_{\mathrm{second}} - \beta|$$
$$\leq \sqrt{\frac{2 \ln T^{(r)}}{H}} + \left( \frac{|\Delta_{\mathrm{first}}^{(r)}|}{H} + \sqrt{\frac{\ln T^{(r)}}{2H}} \right)$$
$$= O\left( \frac{1}{T^{(r)}} \right) \cdot |\Delta_{\mathrm{first}}^{(r)}| + O\left( \sqrt{\frac{\log T^{(r)}}{T^{(r)}}} \right).$$

Plugging the above back into the upper bound on $\mathsf{smCE}(x^{(r)}, p^{(r)})$ shows that in a successful Type 2 round,

$$\mathsf{smCE}(x^{(r)}, p^{(r)}) \leq 1 + O\left( \frac{1}{T^{(r)}} \right) \cdot [\Delta_{\mathrm{first}}^{(r)}]^2 + O\left( \sqrt{\frac{\log T^{(r)}}{T^{(r)}}} \right) \cdot |\Delta_{\mathrm{first}}^{(r)}|.$$

In the following, we show that a Type 2 round succeeds with probability $1 - O(1/T^{(r)})$. Let $X_{\mathrm{first}} := \sum_{s=t+1}^{t+H} x_s$. Note that $X_{\mathrm{first}}$ is a sum of $H$ independent Bernoulli random variables and $\mathbb{E}[X_{\mathrm{first}}] = \mu_{\mathrm{first}} H$. Furthermore, we have $\Delta_{\mathrm{first}}^{(r)} = X_{\mathrm{first}} - \mu_{\mathrm{first}} H$. By an additive Chernoff bound, we have

$$\Pr\left[ |\Delta_{\mathrm{first}}^{(r)}| \leq \sqrt{\frac{H \ln T^{(r)}}{2}} \right] = \Pr\left[ |X_{\mathrm{first}}/H - \mu_{\mathrm{first}}| \leq \sqrt{\frac{\ln T^{(r)}}{2H}} \right] \geq 1 - \frac{2}{T^{(r)}}. \quad (3)$$

Recall that we need to argue that no clipping is applied when $\beta$ is chosen. We analyze the following two cases:

- **Case 1.** $\Delta_{\mathrm{first}}^{(r)} \geq 0$. In this case, we need to show that

$$\mu_{\mathrm{second}} + \frac{\Delta_{\mathrm{first}}^{(r)}}{H} + \sqrt{\frac{\ln T^{(r)}}{2H}} \leq 1.$$

Recall that we assumed $\mu_{\mathrm{first}} \leq 1/2$ and $|\mu_{\mathrm{first}} - \mu| < \sqrt{\frac{2 \ln T^{(r)}}{H}}$. The latter further implies $|\mu_{\mathrm{first}} - \mu_{\mathrm{second}}| = 2|\mu_{\mathrm{first}} - \mu| < \sqrt{\frac{8 \ln T^{(r)}}{H}}$. Thus, it suffices to prove that

$$\sqrt{\frac{8 \ln T^{(r)}}{H}} + \frac{|\Delta_{\mathrm{first}}^{(r)}|}{H} + \sqrt{\frac{\ln T^{(r)}}{2H}} \leq \frac{1}{2}.$$

When $|\Delta_{\mathrm{first}}^{(r)}| \leq \sqrt{\frac{H \ln T^{(r)}}{2}}$ (i.e., the event in Equation (3) holds), the left-hand side above is $O\left( \sqrt{\frac{\log T^{(r)}}{T^{(r)}}} \right)$, which is below $1/2$ as long as $T^{(r)}$ exceeds some universal constant $T_0$.

Therefore, the probability that a clipping is applied is at most $O(1/T^{(r)})$, where we absorb the constraint $T^{(r)} \geq T_0$ into the hidden constant in $O(\cdot)$.

- **Case 2.** $\Delta_{\text{first}}^{(r)} < 0$. In this case, we need to show that

$$\mu_{\text{second}} + \frac{\Delta_{\text{first}}^{(r)}}{H} - \sqrt{\frac{\ln T^{(r)}}{2H}} \geq 0.$$

Recall that the definition of Type 2 rounds implies $\mu_{\text{first}} \geq 10\sqrt{\frac{\ln T^{(r)}}{H}}$. Thus, it suffices to prove that

$$10\sqrt{\frac{\ln T^{(r)}}{H}} - \sqrt{\frac{2\ln T^{(r)}}{H}} - \frac{|\Delta_{\text{first}}^{(r)}|}{H} - \sqrt{\frac{\ln T^{(r)}}{2H}} \geq 0.$$

The above holds whenever the event in Equation (3) happens, since $10 - \sqrt{2} - 1/\sqrt{2} - 1/\sqrt{2} > 0$.

Finally, we argue that, with high probability, we exit the second for-loop via the break statement. Let $X_{\text{second}} \coloneqq \sum_{s=t+H+1}^{t+2H} x_s$ denote the total outcome in the second half. By symmetry, we only deal with the case that $\Delta_{\text{first}}^{(r)} \geq 0$, where we have $\beta = \mu_{\text{second}} + \Delta_{\text{first}}^{(r)}/H + \sqrt{\frac{\ln T^{(r)}}{2H}}$. If the second for-loop runs all the $H$ iterations in full, at the end of it, the value of $\Delta^{(r)}$ will be given by

$$\Delta_{\text{first}}^{(r)} + X_{\text{second}} - \beta H = X_{\text{second}} - \mu_{\text{second}} H - \sqrt{\frac{H \ln T^{(r)}}{2}}.$$

Note that the above is non-negative only if $X_{\text{second}} \leq \mu_{\text{second}} H + \sqrt{\frac{H \ln T^{(r)}}{2}}$, which, by an additive Chernoff bound, holds with probability at most $1/T^{(r)}$. Therefore, with probability $1 - 1/T^{(r)}$, the value of $\Delta^{(r)}$ must fall into $[-1, 0]$ during the second for-loop, and we will take the break statement accordingly. $\qquad\square$

*Proof of Lemma B.4.* Again, without loss of generality, suppose that $\mu_{\text{first}} \leq 1/2$; the other case follows from a completely symmetric argument. In contrast to Type 1 and Type 2 rounds, we say that a Type 3 round *succeeds* if all the following conditions hold simultaneously:

- $\Delta_{\text{first}}^{(r)} \geq 0$, i.e., $\Delta^{(r)} \geq 0$ holds at the end of the first for-loop (on Line 25).

- When $\beta$ is chosen, the clipping (i.e., taking the minimum with 1) is not effective.

- We exit the second for-loop through the break statement on Line 30.

Otherwise, the round *fails*.

By the same argument as in the proof of Lemma B.3, in a successful Type 3 round, we have

$$\mathsf{smCE}(x^{(r)}, p^{(r)}) \leq 1 + O\left(\frac{1}{T^{(r)}}\right) \cdot [\Delta_{\text{first}}^{(r)}]^2 + O\left(\sqrt{\frac{\log T^{(r)}}{T^{(r)}}}\right) \cdot |\Delta_{\text{first}}^{(r)}|.$$

The only change in the argument is the upper bound on $|\alpha - \beta|$, since $\alpha$ is no longer equal to $\mu_{\text{first}}$. Nevertheless, we still have

$$|\alpha - \beta| \leq |\alpha - \mu_{\text{first}}| + |\mu_{\text{first}} - \mu_{\text{second}}| + |\mu_{\text{second}} - \beta|$$
$$\leq \sqrt{\frac{2\mu_{\text{first}} \ln T^{(r)}}{H}} + \sqrt{\frac{2\ln T^{(r)}}{H}} + \left(\frac{|\Delta_{\text{first}}^{(r)}|}{H} + \sqrt{\frac{\ln T^{(r)}}{2H}}\right)$$
$$= O\left(\frac{1}{T^{(r)}}\right) \cdot |\Delta_{\text{first}}^{(r)}| + O\left(\sqrt{\frac{\log T^{(r)}}{T^{(r)}}}\right),$$

and the rest of the analysis goes through.

Thus, it remains to show that a Type 3 round succeeds with probability $1 - O(1/T^{(r)})$. Let $X_{\text{first}} \coloneqq \sum_{s=t+1}^{t+H} x_s$. Note that $X_{\text{first}}$ is a sum of independent Bernoulli random variables and $\mathbb{E}[X_{\text{first}}] = \mu_{\text{first}} H$. By a multiplicative Chernoff bound, for any $\delta \geq 0$, we have

$$\Pr[X_{\text{first}}/H \leq (1 - \delta)\mu_{\text{first}}] \leq \exp\left(-\delta^2 \mu_{\text{first}} H/2\right).$$

In particular, plugging $\delta = \sqrt{\frac{2 \ln T^{(r)}}{\mu_{\text{first}} H}}$ into the above gives

$$\Pr\left[ X_{\text{first}}/H \le \mu_{\text{first}} - \sqrt{\frac{2\mu_{\text{first}} \ln T^{(r)}}{H}} \right] \le \frac{1}{T^{(r)}}.$$

Recall that $\alpha$ is chosen as the maximum between $\mu_{\text{first}} - \sqrt{\frac{2\mu_{\text{first}} \ln T^{(r)}}{H}}$ and 0. Thus, with probability at least $1 - 1/T^{(r)}$, we have $X_{\text{first}}/H \ge \alpha$, which is equivalent to $\Delta^{(r)} \ge 0$ at the end of the first for-loop.

Then, we need to argue that when $\beta$ is chosen, we have $\mu_{\text{second}} + \Delta^{(r)}/H + \sqrt{\frac{\ln T^{(r)}}{2H}} \le 1$. We will show the equivalent inequality:

$$(\mu_{\text{second}} - 1/2) + \Delta^{(r)}/H + \sqrt{\frac{\ln T^{(r)}}{2H}} \le 1/2.$$

For the first term, we note that since $\mu = (\mu_{\text{first}} + \mu_{\text{second}})/2$, the assumption $|\mu_{\text{first}} - \mu| < \sqrt{\frac{2 \ln T^{(r)}}{H}}$ implies $|\mu_{\text{first}} - \mu_{\text{second}}| = O\left(\sqrt{\frac{\log T^{(r)}}{T^{(r)}}}\right)$. With the additional assumption that $\mu_{\text{first}} \le 1/2$, we have

$$\mu_{\text{second}} - 1/2 \le (\mu_{\text{first}} - 1/2) + |\mu_{\text{first}} - \mu_{\text{second}}| \le O\left(\sqrt{\frac{\log T^{(r)}}{T^{(r)}}}\right).$$

For the second term, we note that, at the end of the first for-loop, $\Delta^{(r)}/H$ is given by

$$\frac{X_{\text{first}} - \alpha H}{H} = \left(\frac{X_{\text{first}}}{H} - \mu_{\text{first}}\right) + (\mu_{\text{first}} - \alpha).$$

By an additive Chernoff bound, $\frac{X_{\text{first}}}{H} - \mu_{\text{first}} \le O\left(\sqrt{\frac{\log T^{(r)}}{T^{(r)}}}\right)$ holds with probability $1 - O(1/T^{(r)})$. By our choice of $\alpha$, $\mu_{\text{first}} - \alpha$ is always $O\left(\sqrt{\frac{\log T^{(r)}}{T^{(r)}}}\right)$. Finally, the last term is clearly $O\left(\sqrt{\frac{\log T^{(r)}}{T^{(r)}}}\right)$. Therefore, as long as $T^{(r)}$ is larger than a universal constant $T_0$, the total $O\left(\sqrt{\frac{\log T^{(r)}}{T^{(r)}}}\right)$ term is upper bounded by $1/2$. Again, we can absorb the condition $T^{(r)} \ge T_0$ into the big-$O$ notation, so the second condition (that $\beta$ is not clipped) is satisfied with probability $1 - O(1/T^{(r)})$.

Finally, we argue that we exit the second for-loop via the break statement with high probability. Let $X_{\text{second}} := \sum_{s=t+H+1}^{t+2H} x_s$ denote the total outcome in the second half. Recall that we have $\Delta^{(r)} \ge 0$ at the end of the first for-loop, and that $\beta = \mu_{\text{second}} + \Delta^{(r)}/H + \sqrt{\frac{\ln T^{(r)}}{2H}}$. If the second for-loop runs all the $H$ iterations in full, at the end of it, the value of $\Delta^{(r)}$ will be given by

$$\Delta_{\text{first}}^{(r)} + X_{\text{second}} - \beta H = X_{\text{second}} - \mu_{\text{second}} H - \sqrt{\frac{H \ln T^{(r)}}{2}}.$$

Note that the above is non-negative only if $X_{\text{second}} \le \mu_{\text{second}} H + \sqrt{\frac{H \ln T^{(r)}}{2}}$, which, by an additive Chernoff bound, holds with probability at most $1/T^{(r)}$. Therefore, with probability $1 - 1/T^{(r)}$, the value of $\Delta^{(r)}$ must fall into $[-1, 0]$ during the second for-loop, and we will take the break statement accordingly. $\qquad\square$

## B.2 The Lower Bound Part

We prove the lower bound part of Theorem B.1 via a central limit theorem.

*Proof of Theorem B.1, the lower bound part.* On the product distribution $\mathcal{D} = \prod_{t=1}^{T} \text{Bernoulli}(p_t^{\star})$, the truthful forecaster predicts $p_t = p_t^{\star}$ at every step $t$. Then, we have

$$\underset{x \sim \mathcal{D}}{\mathbb{E}} [\text{smCE}(x, p^{\star})] = \underset{x \sim \mathcal{D}}{\mathbb{E}} \left[ \sup_{f \in \mathcal{F}} \sum_{t=1}^{T} f(p_t^{\star}) \cdot (x_t - p_t^{\star}) \right] \geq \underset{x \sim \mathcal{D}}{\mathbb{E}} \left[ \left| \sum_{t=1}^{T} (x_t - p_t^{\star}) \right| \right],$$

where we use the fact that $\mathcal{F}$ contains the constant functions $f \equiv 1$ and $f \equiv -1$.

Applying the Berry-Esseen theorem [She10] to the random variable $X := \sum_{t=1}^{T} (x_t - p_t^{\star})$ gives:

$$\forall x \in \mathbb{R}, \ \left| \Pr\left[X \leq x \cdot \sigma_0\right] - \Phi(x) \right| \leq C_0 \cdot \sigma_0^{-1} \cdot \rho_0,$$

where $\Phi(x)$ is CDF of the standard normal distribution, $C_0 \leq 0.56$ is a universal constant, and

$$\sigma_0 = \sqrt{\sum_{t=1}^{T} \mathbb{E}\left[(x_t - p_t^{\star})^2\right]} = \sqrt{\sum_{t=1}^{T} p_t^{\star}(1 - p_t^{\star})} \geq \sqrt{T\delta(1-\delta)};$$

$$\rho_0 = \max_{t \in [T]} \frac{\mathbb{E}\left[|x_t - p_t^{\star}|^3\right]}{\mathbb{E}\left[|x_t - p_t^{\star}|^2\right]} = \max_{t \in [T]} \frac{p_t^{\star}(1 - p_t^{\star}) \cdot [(p_t^{\star})^2 + (1 - p_t^{\star})^2]}{p_t^{\star}(1 - p_t^{\star})} \leq 1.$$

In particular, taking $x = -1$ gives:

$$\Pr\left[X \leq -\sigma_0\right] \geq \Phi(-1) - C_0 \cdot \sigma_0^{-1} \cdot \rho_0 = \Omega(1) - O(1/\sqrt{T}).$$

For all sufficiently large $T$, the $O(1/\sqrt{T})$ term is dominated by the $\Omega(1)$ term, in which case we have

$$\underset{x \sim \mathcal{D}}{\mathbb{E}} [\text{smCE}(x, p^{\star})] \geq \mathbb{E}\left[|X|\right] \geq \sigma_0 \cdot \Pr\left[X \leq -\sigma_0\right] = \Omega(\sqrt{T}).$$

Finally, by the inequality $\frac{1}{2}\text{smCE}(x, p) \leq \text{CalDist}(x, p)$ [BGHN23, Lemma 5.4 and Theorem 7.3], the distance from calibration incurred by the truthful forecaster is also $\Omega(\sqrt{T})$. $\qquad\square$

## C  Supplemental Materials for Section 5

The following is a tighter version of Theorem 5.1.

**Theorem C.1.** *For any* $\mathcal{D} \in \Delta(\{0,1\}^T)$, $\text{err}_{\text{SSCE}}(\mathcal{D}, \mathcal{A}^{\text{truthful}}(\mathcal{D})) = O(\mathbb{E}\left[\gamma(\text{Var}_T)\right])$, *where*
$$\gamma(x) := \begin{cases} x, & x < 1, \\ \sqrt{x}, & x \geq 1. \end{cases}$$

*Proof.* Given a function $f : [0,1] \to [-1,1]$ and binary vector $y \in \{0,1\}^T$, we define the martingale $M_t(f, y) := \sum_{s=1}^{t} y_s \cdot f(p_s^{\star}) \cdot (x_s - p_s^{\star})$ where $x \sim \mathcal{D}$ and we use $\mathbb{F}_t$ to denote the filtration describing the randomness of $M_T(f, y)$ up to time $t$ and $p_t^{\star} := \mathbb{E}\left[x_t | \mathbb{F}_{t-1}\right]$. Note that, conditioned on $\mathbb{F}_{t-1}$, $x_t$ is distributed as a Bernoulli with parameter $p_t^{\star}$.

We can write the SSCE of a truthful forecaster in terms of $M_T(f, y)$ as

$$\text{SSCE}(x, p^{\star}) := \underset{y \sim \text{Unif}(\{0,1\}^T)}{\mathbb{E}} \left[ \sup_{f \in \mathcal{F}} M_T(f, y) \right].$$

We now proceed via chaining and define the dyadic scale $\varepsilon_k = 2^{1-k}$ for $k = 0, 1, 2, \ldots$. To cover the set of Lipschitz functions $\mathcal{F}$, we will use the sets of piecewise constant functions $\{\mathcal{F}_\delta\}_{\delta>0}$ described in Lemma C.2. For each function $f \in \mathcal{F}$, let $\pi_k(f)$ be a close function in $\mathcal{F}_{\varepsilon_k}$ such that $d(f, \pi_k(f)) \leq 2\varepsilon_k$. Observe that the covering $\mathcal{F}_{\varepsilon_0}$ is a singleton and that $\pi_k(f)$ always exists as $\mathcal{F}_{\varepsilon_k}$ is a $2\varepsilon_k$-covering of $\mathcal{F}$. Telescoping then gives

$$f(x) = (f(x) - \pi_M(f)(x)) + \pi_0(f)(x) + \sum_{i=1}^{M} [\pi_i(f)(x) - \pi_{i-1}(f)(x)],$$

meaning that we have

$$\mathsf{SSCE}(x, p^\star) \leq \underbrace{\mathbb{E}_{y \sim \mathrm{Unif}(\{0,1\}^T)} \left[ \sup_{f \in \mathcal{F}} \sum_{t=1}^{T} y_t \cdot (f(p_t^\star) - \pi_M(f)(p_t^\star)) \cdot (x_t - p_t^\star) \right]}_{(\text{Term A})}$$

$$+ \underbrace{\mathbb{E}_{y \sim \mathrm{Unif}(\{0,1\}^T)} \left[ \sup_{f \in \mathcal{F}} \sum_{t=1}^{T} y_t \cdot \pi_0(f)(p_t^\star) \cdot (x_t - p_t^\star) \right]}_{(\text{Term B})}$$

$$+ \underbrace{\mathbb{E}_{y \sim \mathrm{Unif}(\{0,1\}^T)} \left[ \sup_{f \in \mathcal{F}} \sum_{i=1}^{M} \sum_{t=1}^{T} y_t \cdot (\pi_i(f)(p_t^\star) - \pi_{i-1}(f)(p_t^\star)) \cdot (x_t - p_t^\star) \right]}_{(\text{Term C})}. \quad (4)$$

First, we can use that $d(f(p_t^\star) - \pi_M(f)(p_t^\star)) \leq 2^{2-M}$ to deterministically bound Term A by

$$\mathbb{E}_{y \sim \mathrm{Unif}(\{0,1\}^T)} \left[ \sup_{f \in \mathcal{F}} \sum_{t=1}^{T} y_t \cdot (f(p_t^\star) - \pi_M(f)(p_t^\star)) \cdot (x_t - p_t^\star) \right] \leq 2^{2-M} \cdot T.$$

Second, we can observe that the image of $\pi_0(f)$ is a singleton: $|\{\pi_0(f) \mid f \in \mathcal{F}\}| = 1$; let this unique function be denoted by $f^\star$. Then, Term B reduces to $\mathbb{E}_{y \sim \mathrm{Unif}(\{0,1\}^T)}[M_T(f^\star, y)]$, which evaluates to $0$ after taking an expectation over $x \sim \mathcal{D}$, since for every $y \in \{0,1\}^T$, $(M_t(f^\star, y))_{0 \leq t \leq T}$ forms a martingale. Third, we can observe that $\pi_i(f) - \pi_{i-1}(f)$ is a function from $[0,1] \to \{-2^{1-i}, 0, 2^{1-i}\}$ that takes a constant value along the segments $[(j-1)2^{1-i}, j2^{1-i})$ for all $j \in [2^{i-1}]$. Thus, we can bound the summands of Term C by

$$\mathbb{E}_{y \sim \mathrm{Unif}(\{0,1\}^T)} \left[ \sup_{f \in \mathcal{F}} \sum_{t=1}^{T} y_t \cdot (\pi_i(f)(p_t^\star) - \pi_{i-1}(f)(p_t^\star)) \cdot (x_t - p_t^\star) \right]$$

$$\leq \sum_{j=0}^{2^{i-1}} \mathbb{E}_{y \sim \mathrm{Unif}(\{0,1\}^T)} \left[ \sup_{v \in \{0, \pm 2^{1-i}\}} \sum_{t=1}^{T} y_t \cdot v \cdot (x_t - p_t^\star) \cdot \mathbb{1} \left[ j2^{1-i} \leq p_t^\star < (j+1)2^{1-i} \right] \right]$$

$$\leq \sum_{j=0}^{2^{i-1}} 2^{1-i} \mathbb{E}_{y \sim \mathrm{Unif}(\{0,1\}^T)} \left[ \sup_{v \in \{\pm 1\}} \underbrace{\sum_{t=1}^{T} y_t \cdot v \cdot (x_t - p_t^\star) \cdot \mathbb{1} \left[ j2^{1-i} \leq p_t^\star < (j+1)2^{1-i} \right]}_{=: M_T(v,y,i,j)} \right].$$

Invoking Lemma C.9 with $\mathcal{G} = \{x \mapsto 1, x \mapsto -1\}$ and $\mathcal{I} = [j2^{1-i}, (j+1)2^{1-i})$, we have that for all $i \in [M]$, $j \in \{0, 1, \ldots, 2^{i-1}\}$, and $y \in \{0,1\}^T$:

$$\mathbb{E}_{x \sim \mathcal{D}} \left[ \sup_{v \in \{\pm 1\}} M_T(v, y, i, j) \right] \leq (48 + 8 \ln 2) \mathbb{E}_{x \sim \mathcal{D}} \left[ \gamma \left( \sum_{t=1}^{T} p_t^\star (1 - p_t^\star) \mathbb{1} \left[ j2^{1-i} \leq p_t^\star < (j+1)2^{1-i} \right] \right) \right].$$

Plugging this into Term C, we have

$$\mathbb{E}_{x \sim \mathcal{D}} [\text{Term C}] \leq (48 + 8 \ln 2) \sum_{i=1}^{M} 2^{1-i} \sum_{j=0}^{2^{i-1}} \mathbb{E}_{x \sim \mathcal{D}} \left[ \gamma \left( \sum_{t=1}^{T} p_t^\star (1 - p_t^\star) \mathbb{1} \left[ j2^{1-i} \leq p_t^\star < (j+1)2^{1-i} \right] \right) \right]$$

$$= (48 + 8 \ln 2) \sum_{i=1}^{M} 2^{1-i} \mathbb{E}_{x \sim \mathcal{D}} \left[ \sum_{j=0}^{2^{i-1}} \gamma \left( \sum_{t=1}^{T} p_t^\star (1 - p_t^\star) \mathbb{1} \left[ j2^{1-i} \leq p_t^\star < (j+1)2^{1-i} \right] \right) \right].$$

Using Lemma C.3, we can simplify

$$\mathop{\mathbb{E}}_{x\sim\mathcal{D}}[\text{Term C}] \le (48+8\ln 2)\sum_{i=1}^{M} 2^{1-i}\cdot\sqrt{2^{i-1}+1}\ \mathop{\mathbb{E}}_{x\sim\mathcal{D}}\left[\gamma\left(\sum_{j=0}^{2^{i-1}}\sum_{t=1}^{T} p_t^\star(1-p_t^\star)\mathbb{1}\left[j2^{1-i}\le p_t^\star < (j+1)2^{1-i}\right]\right)\right]$$

$$\le (48+8\ln 2)\sum_{i=1}^{M} 2^{1-i/2}\ \mathop{\mathbb{E}}_{x\sim\mathcal{D}}\left[\gamma\left(\sum_{t=1}^{T} p_t^\star(1-p_t^\star)\right)\right]$$

$$= (48+8\ln 2)\cdot(2+2\sqrt{2})\cdot\mathop{\mathbb{E}}_{x\sim\mathcal{D}}\left[\gamma\left(\mathrm{Var}_T\right)\right].$$

Plugging this into (4) and observing that we can choose $M$ to be arbitrarily large, we have as desired

$$\mathop{\mathbb{E}}_{x\sim\mathcal{D}}[\text{SSCE}(x,p^\star)] \le \inf_{M\in\mathbb{N}}\left[2^{2-M}\cdot T + \mathop{\mathbb{E}}_{x\sim\mathcal{D}}[\text{Term C}]\right]$$

$$\le (48+8\ln 2)\cdot(2+2\sqrt{2})\cdot\mathop{\mathbb{E}}_{x\sim\mathcal{D}}\left[\gamma\left(\mathrm{Var}_T\right)\right].$$

$\square$

## C.1 Auxillary Lemmas

**Covering Lipschitz functions.** Let us first recall a standard covering of the class of Lipschitz functions $\mathcal{F}\subseteq[-1,1]^{[0,1]}$. We will work with the metric $d$ on the functions $\mathcal{F}$ induced by the $\infty$-norm; that is, for any $f,g\in\mathcal{F}$, $d(f,g):=\sup_{x\in[0,1]}|f(x)-g(x)|$. In this section, for $\delta>0$ and $b>a$ where $\frac{b-a}{\delta}\in\mathbb{Z}$, we will use the shorthand $[a,b]_\delta:=\{a,a+\delta,\dots,b\}$ to denote endpoints of partitioning of $[a,b]$ into segments of length $\delta$. We will also use the shorthand $\lfloor x\rfloor_\delta:=\max\{i\delta\mid i\delta\le x, i\in\mathbb{Z}\}$ to denote rounding down to the nearest multiple of $\delta$.

**Lemma C.2.** *For $\delta>0$ where $\frac{1}{\delta}\in\mathbb{Z}$, consider all functions $f:[0,1]\to[-1,1]$ that satisfy conditions*

$$\begin{aligned}
(1)\quad &\forall x\in[0,1]_\delta : f(x)\in[-1,1]_\delta\\
(2)\quad &\forall x\in[0,1]_\delta\setminus\{1\} : |f(x+\delta)-f(x)|\le\delta\\
(3)\quad &\forall x\in[0,1] : f(x)=f(\lfloor x\rfloor_\delta).
\end{aligned}$$

*This set of functions, which we will denote by $\mathcal{F}_\delta$, is a $2\delta$-covering of the set of 1-Lipschitz functions $\mathcal{F}:[0,1]\to[-1,1]$ in the metric $d$.*

*Proof.* Fix a 1-Lipschitz function $f\in\mathcal{F}$. Let $f'\in\mathcal{F}_\delta$ be the function in our covering where, for all $x\in[0,1]_\delta$, $f'(x)=\lfloor f(x)\rfloor_\delta$. Note that $f'$ is unique because the elements of $\mathcal{F}_\delta$ can be identified by their image on $[0,1]_\delta$. For any $x\in[0,1]$, we have

$$\begin{aligned}
|f(x)-f'(x)| &\le |f(x)-f(\lfloor x\rfloor_\delta)| + |f'(x)-f'(\lfloor x\rfloor_\delta)| + |f(\lfloor x\rfloor_\delta)-f'(\lfloor x\rfloor_\delta)|\\
&\le |x-\lfloor x\rfloor_\delta| + 0 + |f(\lfloor x\rfloor_\delta)-f'(\lfloor x\rfloor_\delta)|\\
&\le 2\delta,
\end{aligned}$$

where the first inequality is the triangle inequality, the second inequality uses the 1-Lipschitzness of $f$ and that $f'(x)=f'(\lfloor x\rfloor_\delta)$, and the third inequality uses the fact that $|f(z)-f'(z)|\le\delta$ and $|z-\lfloor z\rfloor_\delta|\le\delta$ for all $z\in[0,1]$. $\square$

**Bounding sums of $\gamma$.** Consider the piecewise function $\gamma(x):=\begin{cases} x, & x<1,\\ \sqrt{x}, & x\ge 1.\end{cases}$

**Lemma C.3.** *For all values $x_1,\dots,x_n\ge 0$, we can upper bound $\sum_{i=1}^n\gamma(x_i)\le\sqrt{n}\cdot\gamma(\sum_{i=1}^n x_i)$.*

*Proof.* First, suppose that $\sum_{i=1}^n x_i\le 1$. Then, $\gamma(\sum_{i=1}^n x_i)=\sum_{i=1}^n x_i$ and $x_i\le 1$ for all $i\in[n]$. The claim is therefore equivalent to the trivial statement $\sum_{i=1}^n x_i\le\sqrt{n}\cdot\sum_{i=1}^n x_i$.

Now suppose that $\sum_{i=1}^{n} x_i > 1$. The Cauchy-Schwarz inequality gives

$$\sum_{i=1}^{n} \sqrt{x_i} \leq \sqrt{n} \sqrt{\sum_{i=1}^{n} x_i}.$$

By our assumption that $\sum_{i=1}^{n} x_i > 1$, we have $\gamma(\sum_{i=1}^{n} x_i) = \sqrt{\sum_{i=1}^{n} x_i}$. We separately have that

$$\sum_{i=1}^{n} \gamma(x_i) \leq \sum_{i=1}^{n} \sqrt{x_i},$$

because $\gamma(x) = x \leq \sqrt{x}$ for $x \in [0,1]$ and $\gamma(x) = \sqrt{x} = \sqrt{x}$ for $x \geq 1$. Thus,

$$\sum_{i=1}^{n} \gamma(x_i) \leq \sum_{i=1}^{n} \sqrt{x_i} \leq \sqrt{n} \sqrt{\sum_{i=1}^{n} x_i} = \sqrt{n} \cdot \gamma\left(\sum_{i=1}^{n} x_i\right).$$

$\square$

## C.2 Epochs of Doubling Realized Variance

**Definition C.4.** *For $\mathcal{I} \subseteq [0,1]$, consider the stochastic process $(\mathrm{Var}_t(\mathcal{I}))_{0 \leq t \leq T}$ defined as*

$$\mathrm{Var}_t(\mathcal{I}) := \sum_{s=1}^{t} p_s^\star (1 - p_s^\star) \cdot \mathbb{1}\left[p_s^\star \in \mathcal{I}\right],$$

*where $x \sim \mathcal{D}$ and $p_t^\star := \mathrm{Pr}_{x' \sim \mathcal{D}}\left[x'_t = 1 | x'_{1:(t-1)} = x_{1:(t-1)}\right]$. We define the* epochs *with respect to $\mathcal{I}$ as the sequence $\tau_0, \tau_1, \cdots \in \mathbb{N}$ where $\tau_0 = 0$ and, for each $k \in [\lceil \log_2(T) \rceil + 2]$,*

$$\tau_k := \min\left\{t \in [\tau_{k-1} + 1, T] \mid \mathrm{Var}_t(\mathcal{I}) - \mathrm{Var}_{\tau_{k-1}}(\mathcal{I}) \geq 2^{k-1}\right\} \cup \{\infty\}. \tag{5}$$

The epochs $\tau_0, \tau_1, \ldots$ defined in Definition C.4 partition the $T$ time steps of a martingale into epochs such that the realized variance $\mathrm{Var}_t(\mathcal{I})$ increases by approximately $2^{k-1}$ within the $k$-th epoch. In particular, we can understand $\tau_k$ as pointing to the last time step of the $k$th epoch. The definition of $\tau$ ensures that:

- Epoch 1 starts from time step 1, and ends at the earliest time step $t$ such that $\mathrm{Var}_t(\mathcal{I}) \geq 1 = 2^0$.
- For $k \geq 2$, Epoch $k$ starts from the time step after the last step of Epoch $k - 1$, and ends at the earliest time step such that the total variance within the epoch reaches $2^{k-1}$.

We have the following technical facts about the epochs $\tau$.

**Fact C.5.** *The $(\lceil \log_2(T) \rceil + 2)$-th epoch is never complete, i.e., $\tau_{\lceil \log_2(T) \rceil + 2} = \infty$.*

*Proof.* Our definition of $\mathrm{Var}_t(\mathcal{I})$ clearly guarantees $\mathrm{Var}_T(\mathcal{I}) \leq T$, which implies

$$\mathrm{Var}_T(\mathcal{I}) - \mathrm{Var}_{\tau_{\lceil \log_2(T) \rceil + 1}}(\mathcal{I}) \leq T < 2^{\lceil \log_2(T) \rceil + 1},$$

and therefore, $\tau_{\lceil \log_2(T) \rceil + 2} = \infty$. $\square$

**Fact C.6.** *For every epoch $k \in [\lceil \log_2(T) \rceil + 2]$, the change in realized variance in epoch $k$ is deterministically upper bounded by $\mathrm{Var}_{\tau_k}(\mathcal{I}) - \mathrm{Var}_{\tau_{k-1}}(\mathcal{I}) < 2^{k-1} + 1$.*

*Proof.* By definition, $\mathrm{Var}_{\tau_k - 1}(\mathcal{I}) - \mathrm{Var}_{\tau_{k-1}}(\mathcal{I}) < 2^{k-1}$. Because $p_t^\star \in [0,1]$ for all $t \in [T]$, the realized variance increases by at most $p_t^\star(1 - p_t^\star) \leq 1$ in each timestep, i.e. $\mathrm{Var}_{\tau_k}(\mathcal{I}) - \mathrm{Var}_{\tau_k - 1}(\mathcal{I}) \leq 1$. The fact follows by summing the two inequalities. $\square$

**Fact C.7.** *For any epoch $k \in [\lceil \log_2(T) \rceil + 2]$, the probability that the $k$th epoch ends is at most*

$$\mathrm{Pr}\left[\tau_k < \infty\right] \leq \min\left\{\frac{\mathbb{E}\left[\mathbb{1}[\mathrm{Var}_T(\mathcal{I}) \geq 1] \cdot \sqrt{\mathrm{Var}_T(\mathcal{I})}\right]}{\sqrt{2^{k-1}}}, 1\right\}.$$

*Proof.* The sequence of realized variances $\mathrm{Var}_1(\mathcal{I}), \ldots, \mathrm{Var}_T(\mathcal{I})$ is deterministically non-decreasing. Thus, for every epoch $k \in [[\lceil \log_2(T) \rceil + 2]$,

$$\Pr[\tau_k < \infty] \leq \Pr\left[\mathrm{Var}_T(\mathcal{I}) - \mathrm{Var}_{\tau_{k-1}} \geq 2^{k-1} \wedge \mathrm{Var}_T(\mathcal{I}) \geq 1\right]$$

$$\leq \Pr\left[\mathbb{1}\left[\mathrm{Var}_T(\mathcal{I}) \geq 1\right] \cdot \mathrm{Var}_T(\mathcal{I}) \geq 2^{k-1}\right]$$

$$= \Pr\left[\mathbb{1}\left[\mathrm{Var}_T(\mathcal{I}) \geq 1\right] \cdot \sqrt{\mathrm{Var}_T(\mathcal{I})} \geq \sqrt{2^{k-1}}\right],$$

with the second inequality following as $\tau_1 < \infty$ implies $\mathrm{Var}_T(\mathcal{I}) \geq 1$. We can next invoke Markov's inequality $\Pr[X \geq a] \leq \frac{\mathbb{E}[X]}{a}$ with $a = \sqrt{2^{k-1}}$ and $X = \mathbb{1}\left[\mathrm{Var}_T(\mathcal{I}) \geq 1\right] \cdot \sqrt{\mathrm{Var}_T(\mathcal{I})}$ to recover

$$\Pr\left[\mathbb{1}\left[\mathrm{Var}_T(\mathcal{I}) \geq 1\right] \cdot \sqrt{\mathrm{Var}_T} \geq \sqrt{2^{k-1}}\right] \leq \min\left\{\frac{\mathbb{E}\left[\mathbb{1}\left[\mathrm{Var}_T(\mathcal{I}) \geq 1\right] \cdot \sqrt{\mathrm{Var}_T(\mathcal{I})}\right]}{\sqrt{2^{k-1}}}, 1\right\}.$$

$\square$

**Fact C.8.** *The exponentially weighted sum of probabilities that each epoch ends is at most*

$$\sum_{k=2}^{\lceil \log_2(T) \rceil + 2} \sqrt{2^{k-1}} \Pr[\tau_{k-1} < \infty] \leq (2\sqrt{2} + 2) \mathbb{E}\left[\mathbb{1}\left[\mathrm{Var}_T(\mathcal{I}) \geq 1\right] \cdot \sqrt{\mathrm{Var}_T(\mathcal{I})}\right].$$

*Proof.* We will prove the deterministic inequality

$$\sum_{k=2}^{\lceil \log_2(T) \rceil + 2} \sqrt{2^{k-1}} \cdot \mathbb{1}\left[\tau_{k-1} < \infty\right] \leq (2\sqrt{2} + 2)\mathbb{1}\left[\mathrm{Var}_T(\mathcal{I}) \geq 1\right] \cdot \sqrt{\mathrm{Var}_T(\mathcal{I})};$$

the fact follows from taking an expectation on both sides.

Let $K = \max\{k \mid \tau_k < \infty\}$ be the number of completed epochs. When $K = 0$, we have $\mathrm{Var}_T(\mathcal{I}) < 1$, and both sides of the above reduce to 0. Now, suppose that $K \geq 1$, in which case we have $\mathrm{Var}_T(\mathcal{I}) \geq 1$. By telescoping, we can lower bound the realized variance by

$$\mathrm{Var}_T(\mathcal{I}) \geq \sum_{k=1}^{K} 2^{k-1} \geq 2^{K-1}.$$

Separately, by definition of $K$, we have

$$\sum_{k=2}^{\lceil \log_2(T) \rceil + 2} \sqrt{2^{k-1}} \cdot \mathbb{1}\left[\tau_{k-1} < \infty\right] = \sum_{k=2}^{K+1} \sqrt{2^{k-1}}$$

$$= \mathbb{1}\left[\mathrm{Var}_T(\mathcal{I}) \geq 1\right] \sum_{k=2}^{K+1} \sqrt{2^{k-1}}$$

$$\leq \mathbb{1}\left[\mathrm{Var}_T(\mathcal{I}) \geq 1\right] \sqrt{2^K}(\sqrt{2} + 2)$$

with the second equality following from $\mathrm{Var}_T(\mathcal{I}) \geq 1$. Combining the previous two inequalities gives the desired inequality

$$\sum_{k=2}^{\lceil \log_2(T) \rceil + 2} \sqrt{2^{k-1}} \cdot \mathbb{1}\left[\tau_{k-1} < \infty\right] \leq (2\sqrt{2} + 2)\mathbb{1}\left[\mathrm{Var}_T(\mathcal{I}) \geq 1\right] \cdot \sqrt{\mathrm{Var}_T(\mathcal{I})}.$$

$\square$

## C.3 Random Walks with Early Stopping

We now prove a technical result that the magnitude of a random walk with random variance can be upper bounded by its (expected) standard deviation. Compared to Lemma 5.2, the lemma below gives a bound that depends on $\gamma(\mathrm{Var}_T(\mathcal{I}))$ (rather than the square root), and avoids the extra $\log |\mathcal{G}| \cdot \log T$ term. While the leading factor ($\approx \log |\mathcal{G}|$) is larger than the one in Lemma 5.2 ($\approx \sqrt{\log |\mathcal{G}|}$), we will only apply the bound to the case that $|\mathcal{G}| = O(1)$, where the difference between the two is only a constant factor.

**Lemma C.9.** *Given a function* $f : [0,1] \to [-1,1]$, $y \in \{0,1\}^T$, *and set* $\mathcal{I} \subseteq [0,1]$, *consider the martingale* $M_t(f, y, \mathcal{I}) := \sum_{s=1}^{t} y_t \cdot f(p_s^\star) \cdot (x_s - p_s^\star) \cdot \mathbb{1}\,[p_s^\star \in \mathcal{I}]$, *where* $x \sim \mathcal{D}$, *and* $p_t^\star = \Pr_{x' \sim \mathcal{D}}\left[x_t' = 1 | x_{1:(t-1)}' = x_{1:(t-1)}\right]$. *Then, for any finite family* $\mathcal{G}$ *of functions from* $[0,1]$ *to* $[-1,1]$, *any* $y \in \{0,1\}^T$, *and any* $\mathcal{I} \subseteq [0,1]$, *we have*

$$\mathbb{E}_{x \sim \mathcal{D}}\left[\max_{f \in \mathcal{G}} M_T(f, y, \mathcal{I})\right] \le 8\big(6 + \log(|\mathcal{G}|)\big) \mathbb{E}_{x \sim \mathcal{D}}\left[\gamma(\mathrm{Var}_T(\mathcal{I}))\right].$$

*where* $\mathrm{Var}_t(\mathcal{I}) := \sum_{s=1}^{t} p_s^\star(1 - p_s^\star) \cdot \mathbb{1}\,[p_s^\star \in \mathcal{I}]$ *is the realized variance restricted to subset* $\mathcal{I}$, *and*
$$\gamma(x) := \begin{cases} x, & x < 1, \\ \sqrt{x}, & x \ge 1. \end{cases}$$

*Proof.* Let us decompose the horizon into epochs of doubling realized variance with respect to the subset $\mathcal{I}$ as per Definition C.4. Using $\tau$ as defined in (5), we will write $I_k := [\tau_{k-1} + 1 : \min\{T, \tau_k\}]$ to denote the time steps composing epoch $k$ and write $K := \max\{k \mid \tau_k < \infty\}$ to denote the number of completed epochs.

We will separately handle the contributions of epoch 1 and those of later epochs.

**First epoch.** Since $y_t \in \{0,1\}$ and $\|f\|_\infty \le 1$ holds for every $f \in \mathcal{G}$, we can bound the expected contribution from the first epoch as follows:

$$\mathbb{E}_{x \sim \mathcal{D}}\left[\max_{f \in \mathcal{G}} \sum_{t=1}^{\tau_1} y_t \cdot f(p_t^\star) \cdot (x_t - p_t^\star) \cdot \mathbb{1}\,[p_t^\star \in \mathcal{I}]\right] \le \mathbb{E}_{x \sim \mathcal{D}}\left[\sum_{t=1}^{\tau_1} |x_t - p_t^\star| \cdot \mathbb{1}\,[p_t^\star \in \mathcal{I}]\right]. \quad (6)$$

Note that for any $p \in [0,1]$ and Bernoulli random variable $x \sim \mathrm{Bernoulli}(p)$,

$$\mathbb{E}\,[|x - p|] = \Pr\,[x = 0] \cdot |0 - p| + \Pr\,[x = 1] \cdot |1 - p| = 2p(1 - p).$$

It thus follows that the process $(X_t)_{0 \le t \le T}$ where

$$X_t := \sum_{s=1}^{t} [|x_s - p_s^\star| - 2p_s^\star(1 - p_s^\star)] \cdot \mathbb{1}\,[p_s^\star \in \mathcal{I}]$$

is a martingale, as conditioning on any realization of $x_{1:(t-1)}$, we have

$$\mathbb{E}_{x \sim \mathcal{D}}\,[|x_t - p_t^\star| - 2p_t^\star(1 - p_t^\star) \mid x_{1:t-1}] = \mathbb{E}_{x \sim \mathrm{Bernoulli}(p_t^\star)}\,[|x - p_t^\star|] - 2p_t^\star(1 - p_t^\star) = 0.$$

By the optional stopping theorem, we have

$$\mathbb{E}_{x \sim \mathcal{D}}\left[\sum_{t=1}^{\tau_1} [|x_t - p_t^\star| - 2p_t^\star(1 - p_t^\star)] \cdot \mathbb{1}\,[p_t^\star \in \mathcal{I}]\right] = \mathbb{E}\,[X_{\tau_1}] = 0.$$

Plugging this identity into (6) gives

$$\mathbb{E}_{x \sim \mathcal{D}}\left[\max_{f \in \mathcal{G}} \sum_{t=1}^{\tau_1} y_t \cdot f(p_t^\star) \cdot (x_t - p_t^\star) \cdot \mathbb{1}\,[p_t^\star \in \mathcal{I}]\right] \le \mathbb{E}_{x \sim \mathcal{D}}\left[\sum_{t=1}^{\tau_1} 2p_t^\star(1 - p_t^\star) \cdot \mathbb{1}\,[p_t^\star \in \mathcal{I}]\right]$$

$$= 2\,\mathbb{E}_{x \sim \mathcal{D}}\,[\mathrm{Var}_{\tau_1}(\mathcal{I})] \quad (7)$$

$$\le 2\,\mathbb{E}_{x \sim \mathcal{D}}\,[\min\{2, \mathrm{Var}_T(\mathcal{I})\}]$$

$$\le 4\,\mathbb{E}_{x \sim \mathcal{D}}\,[\min\{1, \mathrm{Var}_T(\mathcal{I})\}],$$

where the third step applies Fact C.6 with $k = 1$.

**Later epochs.** Applying a triangle inequality and the law of total expectation gives

$$\mathbb{E}_{x \sim \mathcal{D}}\left[\max_{f \in \mathcal{G}} \sum_{t=\tau_1+1}^{T} y_t \cdot f(p_t^\star) \cdot (x_t - p_t^\star) \cdot \mathbb{1}\left[p_t^\star \in \mathcal{I}\right]\right]$$

$$= \mathbb{E}_{x \sim \mathcal{D}}\left[\max_{f \in \mathcal{G}} \sum_{k=2}^{K+1} \sum_{t \in I_k} y_t \cdot f(p_t^\star) \cdot (x_t - p_t^\star) \cdot \mathbb{1}\left[p_t^\star \in \mathcal{I}\right]\right]$$

$$\leq \mathbb{E}_{x \sim \mathcal{D}}\left[\sum_{k=2}^{K+1} \max_{f \in \mathcal{G}} \sum_{t \in I_k} y_t \cdot f(p_t^\star) \cdot (x_t - p_t^\star) \cdot \mathbb{1}\left[p_t^\star \in \mathcal{I}\right]\right] \tag{8}$$

$$= \sum_{k=2}^{\lceil \log_2(T) \rceil + 2} \mathbb{E}_{x \sim \mathcal{D}}\left[\max_{f \in \mathcal{G}} \sum_{t=1}^{T} y_t \cdot f(p_t^\star) \cdot (x_t - p_t^\star) \cdot \mathbb{1}\left[p_t^\star \in \mathcal{I} \wedge t \in I_k\right]\right]$$

$$= \sum_{k=2}^{\lceil \log_2(T) \rceil + 2} \Pr\left[\tau_{k-1} < \infty\right] \cdot \mathbb{E}_{x \sim \mathcal{D}}\left[\max_{f \in \mathcal{G}} M_T^{k,f} \mid \tau_{k-1} < \infty\right],$$

where we define the process

$$M_T^{k,f} := \sum_{t=1}^{T} y_t \cdot f(p_t^\star) \cdot (x_t - p_t^\star) \cdot \mathbb{1}\left[p_t^\star \in \mathcal{I} \wedge t \in I_k\right]. \tag{9}$$

In the above, the third step uses Fact C.5, namely that $\tau_{\lceil \log_2(T) \rceil + 2} = \infty$. We can use Freedman's inequality to obtain a maximal inequality for each of these $M_T^{k,f}$ processes.

**Fact C.10.** *For every $y \in \{0,1\}^T$ and $k \geq 2$, we can uniformly bound the process $M_T^{k,f}$ defined in (9) over a finite class $\mathcal{G}$ of functions from $[0,1]$ to $[-1,1]$ by*

$$\mathbb{E}_{x \sim \mathcal{D}}\left[\max_{f \in \mathcal{G}} M_T^{k,f} \mid \tau_{k-1} < \infty\right] \leq \sqrt{2^{k-1}}(2 + 2\sqrt{\log |\mathcal{G}|}) + 2 + 2\log |\mathcal{G}|.$$

Applying Fact C.10 to each of the martingales $M_T^{k,f}$ in (8) gives us

$$\mathbb{E}_{x \sim \mathcal{D}}\left[\max_{f \in \mathcal{G}} \sum_{t=\tau_1+1}^{T} y_t \cdot f(p_t^\star) \cdot (x_t - p_t^\star) \cdot \mathbb{1}\left[p_t^\star \in \mathcal{I}\right]\right]$$

$$\leq \sum_{k=2}^{\lceil \log_2(T) \rceil + 2} \Pr\left[\tau_{k-1} < \infty\right] \left(\sqrt{2^{k-1}}(2 + 2\sqrt{\log |\mathcal{G}|}) + 2 + 2\log |\mathcal{G}|\right). \tag{10}$$

To upper bound the right-hand side above, we use Fact C.8 to bound

$$\sum_{k=2}^{\lceil \log_2(T) \rceil + 2} \Pr\left[\tau_{k-1} < \infty\right] \sqrt{2^{k-1}} \leq \mathbb{E}\left[\mathbb{1}\left[\mathrm{Var}_T(\mathcal{I}) \geq 1\right] \cdot \sqrt{\mathrm{Var}_T(\mathcal{I})}\right] (2 + 2\sqrt{2}),$$

and use Fact C.7 to bound

$$\sum_{k=2}^{\lceil \log_2(T) \rceil + 2} \Pr\left[\tau_{k-1} < \infty\right] \leq \sum_{k=2}^{\lceil \log_2(T) \rceil + 2} \min\left\{1, \frac{\mathbb{E}\left[\mathbb{1}\left[\mathrm{Var}_T(\mathcal{I}) \geq 1\right] \cdot \sqrt{\mathrm{Var}_T(\mathcal{I})}\right]}{2^{(k-2)/2}}\right\}$$

$$\leq \mathbb{E}\left[\mathbb{1}\left[\mathrm{Var}_T(\mathcal{I}) \geq 1\right] \cdot \sqrt{\mathrm{Var}_T(\mathcal{I})}\right] (2 + \sqrt{2}).$$

Plugging these into (10) gives

$$\mathbb{E}\left[\max_{f \in \mathcal{G}}\left[M_T(f, y, \mathcal{I}) - M_{\tau_1}(f, y, \mathcal{I})\right]\right]$$

$$\leq \mathbb{E}\left[\mathbb{1}\left[\mathrm{Var}_T(\mathcal{I}) \geq 1\right] \cdot \sqrt{\mathrm{Var}_T(\mathcal{I})}\right] (2 + 2\sqrt{2})(2 + 2\sqrt{\log |\mathcal{G}|} + \sqrt{2} + \sqrt{2}\log |\mathcal{G}|)$$

$$\leq \mathbb{E}\left[\mathbb{1}\left[\mathrm{Var}_T(\mathcal{I}) \geq 1\right] \cdot \sqrt{\mathrm{Var}_T(\mathcal{I})}\right] 8\left(5 + \log |\mathcal{G}|\right). \tag{11}$$

**Combine bounds.** Combining (7) and (11) and recalling the definition of $\gamma$, we recover our main claim

$$\mathop{\mathbb{E}}_{x\sim\mathcal{D}}\left[\max_{f\in\mathcal{G}} M_T(f,y,\mathcal{I})\right]$$

$$\leq \mathop{\mathbb{E}}_{x\sim\mathcal{D}}\left[\max_{f\in\mathcal{G}} M_{\tau_1}(f,y,\mathcal{I})\right] + \mathop{\mathbb{E}}_{x\sim\mathcal{D}}\left[\max_{f\in\mathcal{G}} M_T(f,y,\mathcal{I}) - M_{\tau_1}(f,y,\mathcal{I})\right]$$

$$\leq 4\mathop{\mathbb{E}}_{x\sim\mathcal{D}}\left[\min\left\{1,\mathrm{Var}_T(\mathcal{I})\right\}\right] + 8\big(5+\log|\mathcal{G}|\big)\cdot\mathbb{E}\left[\mathbb{1}\left[\mathrm{Var}_T(\mathcal{I})\geq 1\right]\sqrt{\mathrm{Var}_T(\mathcal{I})}\right]$$

$$\leq 4\mathop{\mathbb{E}}_{x\sim\mathcal{D}}\left[\gamma(\mathrm{Var}_T(\mathcal{I}))\right] + 8\big(5+\log|\mathcal{G}|\big)\cdot\mathop{\mathbb{E}}_{x\sim\mathcal{D}}\left[\gamma(\mathrm{Var}_T(\mathcal{I}))\right]$$

$$\leq 8\cdot(6+\log|\mathcal{G}|)\cdot\mathop{\mathbb{E}}_{x\sim\mathcal{D}}\left[\gamma(\mathrm{Var}_T(\mathcal{I}))\right].$$

The second step above applies Inequalities (7) and (11). The third holds since $\min\{1,x\}\leq\gamma(x)$ and $\mathbb{1}\left[x\geq 1\right]\sqrt{x}\leq\gamma(x)$ hold for all $x\geq 0$. $\qquad\square$

Let us recall Freedman's inequality [Fre75].

**Lemma C.11.** *Consider a martingale $M_n\sim\mathcal{D}$ with filtration $(\mathbb{F}_t)$ where $|M_t - M_{t-1}|\leq 1$ for all $t\in[n]$. For all $x,y>0$, we have the following high-probability bound on $M_n$:*

$$\Pr\left[\exists n, M_n\geq x \ \wedge\ \sum_{t=1}^n\mathbb{E}\left[(M_t-M_{t-1})^2\,|\mathbb{F}_{t-1}\right]\leq y\right]\leq\exp\left(-\frac{x^2}{2(x+y)}\right).$$

We now prove Fact C.10.

**Fact C.10.** *For every $y\in\{0,1\}^T$ and $k\geq 2$, we can uniformly bound the process $M_T^{k,f}$ defined in (9) over a finite class $\mathcal{G}$ of functions from $[0,1]$ to $[-1,1]$ by*

$$\mathop{\mathbb{E}}_{x\sim\mathcal{D}}\left[\max_{f\in\mathcal{G}} M_T^{k,f}\mid\tau_{k-1}<\infty\right]\leq\sqrt{2^{k-1}}(2+2\sqrt{\log|\mathcal{G}|})+2+2\log|\mathcal{G}|.$$

*Proof.* Fix any $f\in\mathcal{G}$. For $t\notin I_k$, we have trivially that for any $x_{1:t-1}\in\{0,1\}^{t-1}$:

$$\mathop{\mathbb{E}}_{x'\sim\mathcal{D}}\left[y_t\cdot f(p_t^\star)\cdot(x_t'-p_t^\star)\cdot\mathbb{1}\left[t\in I_k\wedge p_t^\star\in\mathcal{I}\right]\mid x_{1:t-1}'=x_{1:t-1}\right]=0.$$

For $t\in I_k$, since $\mathbb{1}\left[\tau_{k-1}<\infty\right]$ and $p_t^\star$ is measurable by $x_{1:t-1}$, we again have that

$$\mathop{\mathbb{E}}_{x'\sim\mathcal{D}}\left[y_t\cdot f(p_t^\star)\cdot(x_t-p_t^\star)\cdot\mathbb{1}\left[t\in I_k\wedge p_t^\star\in\mathcal{I}\right]\mid x_{1:t-1}'=x_{1:t-1}\right]$$

$$=y_t\cdot f(p_t^\star)\cdot\mathbb{1}\left[t\in I_k\wedge p_t^*\in\mathcal{I}\right]\cdot\left(\mathop{\mathbb{E}}_{x'\sim\mathcal{D}}\left[x_t'\mid x_{1:t-1}'=x_{1:t-1}\right]-p_t^\star\right)$$

$$=0.$$

This means that $M_T^{k,f}$ is a martingale even conditioned on the event that $\tau_{k-1}<\infty$.

Our construction of epoch $k$ in (5) further guarantees that the realized variance of $M_T^{k,f}$ is deterministically upper bounded by $\mathrm{Var}_{\tau_k}(\mathcal{I})-\mathrm{Var}_{\tau_{k-1}}(\mathcal{I})\leq 2^{k-1}+1$ (Fact C.6). Thus,

$$2^{k-1}+1\geq\sum_{t=1}^T p_t^\star(1-p_t^\star)\cdot\mathbb{1}\left[t\in I_k\wedge p_t^\star\in\mathcal{I}\right]$$

$$=\sum_{t=1}^T\mathop{\mathbb{E}}_{x\sim\mathsf{Bernoulli}(p_t^\star)}\left[(x-p_t^\star)^2\right]\cdot\mathbb{1}\left[t\in I_k\wedge p_t^\star\in\mathcal{I}\right]$$

$$=\sum_{t=1}^T\mathop{\mathbb{E}}_{x_t'\sim\mathcal{D}_t}\left[(x_t'-p_t^\star)^2\mid x_{1:t-1}'=x_{1:t-1}\right]\cdot\mathbb{1}\left[t\in I_k\wedge p_t^\star\in\mathcal{I}\right]^2$$

$$\geq\sum_{t=1}^T y_t^2\cdot f(p_t^\star)^2\cdot\mathop{\mathbb{E}}_{x_t'\sim\mathcal{D}_t}\left[(x_t'-p_t^\star)^2\mid x_{1:t-1}'=x_{1:t-1}\right]\cdot\mathbb{1}\left[t\in I_k\wedge p_t^\star\in\mathcal{I}\right]^2$$

$$=\sum_{t=1}^T\mathop{\mathbb{E}}_{x_t'\sim\mathcal{D}_t}\left[(M_t^{k,f}-M_{t-1}^{k,f})^2\mid x_{1:t-1}'=x_{1:t-1}\right]. \tag{12}$$

where the first equality uses the definition of a Bernoulli's variance; the second equality uses that, conditioned on $\mathbb{F}_{t-1}$, $x_t \sim \text{Bernoulli}(p_t^\star)$; and the second inequality uses that $y_t^2 \leq 1$ and $|f(x)| \leq 1$ for all $x \in [0, 1]$.

We can thus use Freedman's inequality to bound the deviation of each martingale $M_T^{k,f}$. First, observe that the quadratic formula gives the inequality $\exp\left(-\frac{x^2}{2(x+y)}\right) \leq p$ if $x \geq \log(1/p) + \sqrt{\log^2(p) + 2y\log(1/p)}$. We can therefore invoke Lemma C.11 with $y = 2^{k-1} + 1$ and

$$x = 2\log(1/p) + \sqrt{2y\log(1/p)} \geq \log(1/p) + \sqrt{\log^2(p) + 2y\log(1/p)}$$

to show that

$$p \geq \Pr\left[M_T^{k,f} \geq x \wedge \sum_{t=1}^T \mathop{\mathbb{E}}_{x_t' \sim \mathcal{D}_t}\left[(M_t^{k,f} - M_{t-1}^{k,f})^2 \mid x_{1:t-1}' = x_{1:t-1}\right] \leq 2^{k-1} + 1 \mid \tau_{k-1} < \infty\right].$$

Applying (12), we can simplify this to

$$p \geq \Pr\left[M_T^{k,f} \geq \sqrt{2(2^{k-1} + 1)\log(1/p)} + 2\log(1/p) \mid \tau_{k-1} < \infty\right]$$

$$\geq \Pr\left[M_T^{k,f} \geq \sqrt{2^{k+1}\log(1/p)} + 2\log(1/p) \mid \tau_{k-1} < \infty\right].$$

We can then take a union bound over $\mathcal{G}$ for

$$p \geq \Pr\left[\max_{f \in \mathcal{G}} M_T^{k,f} \geq \sqrt{2^{k+1}\log(|\mathcal{G}|/p)} + 2\log(|\mathcal{G}|/p) \mid \tau_{k-1} < \infty\right].$$

Using the layer cake representation of expectation, we can convert this high-probability bound into the expectation bound through a change of variables

$$\mathbb{E}\left[\max_{f \in \mathcal{G}} M_T^{k,f} \mid \tau_{k-1} < \infty\right] = \int_0^\infty \Pr\left[\max_{f \in \mathcal{G}} M_T^{k,f} \geq t \mid \tau_{k-1} < \infty\right] \, \mathrm{d}t$$

$$= \int_0^1 \sqrt{2^{k+1}\log(|\mathcal{G}|/p)} + 2\log(|\mathcal{G}|/p) \, \mathrm{d}p$$

$$= \sqrt{2^{k+1}}\left(\frac{|\mathcal{G}|}{2}\sqrt{\pi} \cdot \text{erfc}(\sqrt{\log|\mathcal{G}|}) + \sqrt{\log|\mathcal{G}|}\right) + 2 + 2\log|\mathcal{G}|,$$

where the last equality follows by Fact C.12. When $|\mathcal{G}| > 1$, we can compute the integral to be

$$\mathbb{E}\left[\max_{f \in \mathcal{G}} M_T^{k,f} \mid \tau_{k-1} < \infty\right] \leq \sqrt{2^{k+1}}\left(\frac{|\mathcal{G}|}{2\sqrt{\log|\mathcal{G}|}}\exp(-\log|\mathcal{G}|) + \sqrt{\log|\mathcal{G}|}\right) + 2 + 2\log|\mathcal{G}|$$

$$\leq \sqrt{2^{k-1}}(2 + 2\sqrt{\log|\mathcal{G}|}) + 2 + 2\log|\mathcal{G}|,$$

where the first inequality uses that $\text{erfc}(z) < \frac{\exp(-z^2)}{z\sqrt{\pi}}$. When $|\mathcal{G}| = 1$, we again have

$$\mathbb{E}\left[\max_{f \in \mathcal{G}} M_T^{k,f} \mid \tau_{k-1} < \infty\right] \leq \sqrt{\pi}\sqrt{2^{k-1}} + 2$$

$$\leq \sqrt{2^{k-1}}(2 + 2\sqrt{\log|\mathcal{G}|}) + 2 + 2\log|\mathcal{G}|.$$

$\square$

**Fact C.12.** *For $k, n \in \mathbb{Z}_+$, the following integral equality holds*

$$\int_0^1 \sqrt{2^{k+1}\log(n/p)} + 2\log(n/p) \, \mathrm{d}p = \sqrt{2^{k+1}}\left(\frac{n}{2}\sqrt{\pi} \cdot \text{erfc}(\sqrt{\log n}) + \sqrt{\log n}\right) + 2 + 2\log n$$

*where* erfc *denotes the complementary error function.*

*Proof.* Let us first separate the integral into two parts:

$$\int_0^1 \sqrt{2^{k+1}\log(n/p) + 2\log(n/p)}\ \mathrm{d}p = \int_0^1 \sqrt{2^{k+1}\log(n/p)}\ \mathrm{d}p + \int_0^1 2\log(n/p)\ \mathrm{d}p.$$

We can bound the second integral easily. Since $\log(n/p) = \log n - \log p$,

$$\int_0^1 2\log(n/p)\ \mathrm{d}p = \int_0^1 2(\log n - \log p)\ \mathrm{d}p$$
$$= 2\log n \int_0^1 \mathrm{d}p - 2\int_0^1 \log p\ \mathrm{d}p$$
$$= 2\log n + 2 \tag{13}$$

Now we consider the first integral. Let $u = \log(n/p)$. Then $p = ne^{-u}$ and $\mathrm{d}p = -ne^{-u}\ \mathrm{d}u$. When $p = 1$, $u = \log n$. When $p = 0$, $u$ goes to $\infty$. Thus, the integral becomes:

$$\int_0^1 \sqrt{2^{k+1}\log(n/p)}\ \mathrm{d}p = \int_\infty^{\log n} \sqrt{2^{k+1}u} \cdot (-ne^{-u})\ \mathrm{d}u$$
$$= n\sqrt{2^{k+1}} \int_{\log n}^\infty \sqrt{u}\, e^{-u}\ \mathrm{d}u.$$

The integral involving the error function $\mathrm{erfc}(x)$ can be recognized:

$$\int_{\log n}^\infty \sqrt{u}\, e^{-u}\ \mathrm{d}u = -\sqrt{u}\, e^{-u}\Big|_{\log n}^\infty + \int_{\log n}^\infty \frac{1}{2\sqrt{u}} e^{-u}\ \mathrm{d}u$$
$$= \lim_{u\to\infty}\left(-\sqrt{u}\, e^{-u}\right) - \left(-\sqrt{\log n}\, e^{-\log n}\right) + \int_{\log n}^\infty \frac{1}{2\sqrt{u}} e^{-u}\ \mathrm{d}u$$
$$= \sqrt{\log n}\, e^{-\log n} + \int_{\log n}^\infty \frac{1}{2\sqrt{u}} e^{-u}\ \mathrm{d}u$$
$$= \sqrt{\log n}\, e^{-\log n} + \int_{\sqrt{\log n}}^\infty e^{-t^2}\ \mathrm{d}t$$
$$= \frac{\sqrt{\log n}}{n} + \frac{\sqrt{\pi}}{2}\mathrm{erfc}(\sqrt{\log n}).$$

Thus, the integral $\int_0^1 \sqrt{2^{k+1}\log(n/p)}\ \mathrm{d}p$ is given by

$$n\sqrt{2^{k+1}}\left(\frac{\sqrt{\pi}}{2}\mathrm{erfc}(\sqrt{\log n}) + \frac{\sqrt{\log n}}{n}\right) = \sqrt{2^{k+1}}\left(\frac{n\sqrt{\pi}}{2}\mathrm{erfc}(\sqrt{\log n}) + \sqrt{\log n}\right). \tag{14}$$

Summing (13) and (14) gives the claim. $\qquad\square$

### C.4  Proof of Lemma 5.2

**Lemma 5.2.** *Given a function* $f : [0,1] \to [-1,1]$ *and* $y \in \{0,1\}^T$, *consider the martingale* $M_t(f,y) := \sum_{s=1}^t y_s \cdot f(p_s^\star) \cdot (x_s - p_s^\star)$ *where* $x \sim \mathcal{D}$. *Then, for any finite family* $\mathcal{G}$ *of functions from* $[0,1]$ *to* $[-1,1]$ *and any* $y \in \{0,1\}^T$, *we have*

$$\mathbb{E}_{x\sim\mathcal{D}}\left[\max_{f\in\mathcal{G}} M_T(f,y)\right] \le O\left(\log|\mathcal{G}| \cdot \log T + \sqrt{\log|\mathcal{G}|} \cdot \mathbb{E}_{x\sim\mathcal{D}}\left[\sqrt{\mathrm{Var}_T}\right]\right).$$

*Proof.* Let us decompose the martingale $M_T(f,y)$ into epochs of doubling realized variance with respect to $\mathcal{I} = [0,1]$ as per Definition C.4. Using $\tau$ as defined in (5), we will write $I_k := [\tau_{k-1} + 1, \min\{T, \tau_k\}]$ to denote the time steps composing epoch $k$ and write $K := \max\{k \mid \tau_k < \infty\}$ to denote the number of completed epochs.

Applying a triangle inequality and the law of total expectation gives

$$\mathbb{E}_{x\sim\mathcal{D}}\left[\max_{f\in\mathcal{G}}\sum_{t=1}^{T}y_t\cdot f(p_t^\star)\cdot(x_t-p_t^\star)\right]$$

$$=\mathbb{E}_{x\sim\mathcal{D}}\left[\max_{f\in\mathcal{G}}\sum_{k=1}^{K+1}\sum_{t\in I_k}y_t\cdot f(p_t^\star)\cdot(x_t-p_t^\star)\right]$$

$$\leq\mathbb{E}_{x\sim\mathcal{D}}\left[\sum_{k=1}^{K+1}\max_{f\in\mathcal{G}}\sum_{t\in I_k}y_t\cdot f(p_t^\star)\cdot(x_t-p_t^\star)\right]$$

$$=\sum_{k=1}^{\lceil\log_2(T)\rceil+2}\mathbb{E}_{x\sim\mathcal{D}}\left[\max_{f\in\mathcal{G}}\sum_{t\in I_k}y_t\cdot f(p_t^\star)\cdot(x_t-p_t^\star)\cdot\mathbb{1}\left[t\in I_k\right]\right]$$

$$=\sum_{k=1}^{\lceil\log_2(T)\rceil+2}\Pr\left[\tau_{k-1}<\infty\right]\cdot\mathbb{E}_{x\sim\mathcal{D}}\left[\max_{f\in\mathcal{G}}M_T^{k,f}\mid\tau_{k-1}<\infty\right].\qquad(15)$$

where we define the process $M_T^{k,f}:=\sum_{t=1}^{T}y_t\cdot f(p_t^\star)\cdot(x_t-p_t^\star)\cdot\mathbb{1}\left[t\in I_k\right]$. In the above, the second equality uses Fact C.5, namely that $\tau_{\lceil\log_2(T)\rceil+2}=\infty$. Applying Fact C.10 to each of the martingales $M_T^{k,f}$ in (15) gives us

$$\mathbb{E}_{x\sim\mathcal{D}}\left[\max_{f\in\mathcal{G}}\sum_{t=1}^{T}y_t\cdot f(p_t^\star)\cdot(x_t-p_t^\star)\right]$$

$$\leq\sum_{k=1}^{\lceil\log_2(T)\rceil+2}\Pr\left[\tau_{k-1}<\infty\right]\cdot\left[\sqrt{2^{k-1}}(2+2\sqrt{\log|\mathcal{G}|})+2+2\log|\mathcal{G}|\right].$$

We can upper bound some of the summands in the right-hand side by using Fact C.8 to bound

$$\sum_{k=2}^{\lceil\log_2(T)\rceil+2}\Pr\left[\tau_{k-1}<\infty\right]\sqrt{2^{k-1}}\leq\mathbb{E}\left[\sqrt{\mathrm{Var}_T}\right](2+2\sqrt{2}).$$

This gives that

$$\mathbb{E}_{x\sim\mathcal{D}}\left[\max_{f\in\mathcal{G}}\sum_{t=1}^{T}y_t\cdot f(p_t^\star)\cdot(x_t-p_t^\star)\right]$$

$$\leq(2+2\log|\mathcal{G}|)(\lceil\log_2(T)\rceil+2)+\mathbb{E}\left[\sqrt{\mathrm{Var}_T}\right](2+2\sqrt{2})(2+2\sqrt{\log|\mathcal{G}|}).$$

□

# D   Supplemental Materials for Section 6

**Notation.**  For all stochastic processes $(X_t)$, we use $X_{t_1:t_2}=X_{\min\{t_2,T\}}-X_{t_1}$ to denote the increment within the time interval $(t_1,t_2]$ (with $X_0=0$ by default).

## D.1   Proof of the Weaker Lower Bound

We restate and prove Lemmas 6.2 and 6.3.

**Lemma 6.2.** *For any $x\in\{0,1\}^T$ and $p\in[0,1]^T$, we have $\mathsf{SSCE}(x,p)\geq\Omega\left(\sqrt{N_T}\right)$.*

*Proof.* Recall that $\mathsf{SSCE}$ is defined using $\mathsf{smCE}$, which is in turn a supremum over the family $\mathcal{F}$ of Lipschitz functions. Since both $f\equiv1$ and $f\equiv-1$ are included in $\mathcal{F}$, for any realized sequences $x$ and $p$, we can lower bound $\mathsf{SSCE}(x,p)$ as follows:

$$\mathsf{SSCE}(x,p)\geq\mathbb{E}_{y\sim\mathsf{Unif}(\{0,1\}^T)}\left[\left|\sum_{t=1}^{T}y_t\cdot(x_t-p_t)\right|\right]=\mathbb{E}_y\left[\left|\sum_{t=1}^{T}z_t+\mu\right|\right],$$

where we have defined $z_t := (y_t - 0.5)(x_t - p_t)$ to be zero-mean independent random variables, and $\mu := \sum_{t=1}^{T} 0.5(x_t - p_t)$. Now we partition $[T]$ into $T_1$ and $T_2$, where $T_1$ includes the all time steps such that $|x_t - p_t| \geq \frac{1}{2}$, and $T_2 = T \setminus T_1$ contains the remaining time steps. From the definition of $N_T$, it immediately follows that $N_T = |T_1|$. Letting $Z_1 := \sum_{t \in T_1} z_t$ and $Z_2 := \sum_{t \in T_2} z_t$, it remains to lower bound $\mathbb{E}[|Z_1 + Z_2 + \mu|]$ by $\Omega(\sqrt{N_T})$.

We will first prove that $\mathbb{E}[|Z_1|] \geq C\sqrt{N_T}$ for a universal constant $C > 0$. From the Berry-Esseen theorem (e.g. from [She10]), the CDF of $Z_1$ can be approximated by the CDF of the standard normal distribution as follows:

$$\forall x \in \mathbb{R}, \ \left| \Pr\left[Z_1 \leq x \cdot \sigma_0\right] - \Phi(x) \right| \leq C_0 \cdot \sigma_0^{-1} \cdot \rho_0,$$

where $\Phi(x)$ is the standard Gaussian CDF, $C_0$ is a universal constant no larger than $0.56$, and

$$\sigma_0 = \sqrt{\sum_{t \in T_1} \mathbb{E}[z_t^2]} = \sqrt{\frac{1}{4} \sum_{t \in T_1}(x_t - p_t)^2} \geq \frac{1}{4}\sqrt{N_T};$$

$$\rho_0 = \max_{t \in T_1} \frac{\mathbb{E}[|z_t|^3]}{\mathbb{E}[|z_t|^2]} = \max_{t \in T_1} \frac{|x_t - p_t|^3/8}{|x_t - p_t|^2/4} \leq \frac{1}{2}.$$

As a result, we can lower bound the probability of $|Z_1| \geq 0.05\sqrt{N_T}$ as follows:

$$\Pr\left[|Z_1| \geq 0.05\sqrt{N_T}\right] \geq \Pr\left[|Z_1| > 0.2 \cdot \sigma_0\right] \qquad (\sigma_0 \geq \tfrac{1}{4}\sqrt{N_T})$$

$$= 2\left(1 - \Pr\left[Z_1 \leq 0.2 \cdot \sigma_0\right]\right) \qquad (Z_1 \text{ is symmetric})$$

$$\geq 2\left(1 - \Phi(0.2) - 2C_0/\sqrt{N_T}\right). \qquad \text{(Berry-Esseen theorem)}$$

Since $C_0 \leq 0.56$ and $\Phi(0.2) \leq 0.58$, we can guarantee $\Pr\left[|Z_1| \geq 0.05\sqrt{N_T}\right] \geq \Omega(1)$ for all $N_T \geq 8$. When $N_T \leq 7$, we have $|Z_1| = N_T/2 \geq 0.05\sqrt{N_T}$ when all $\{z_t \mid t \in T_1\}$ are positive, which happens with probability $2^{-N_T} \geq 2^{-7} = \Omega(1)$. Therefore, we can always conclude that

$$\mathbb{E}[|Z_1|] \geq 0.05\sqrt{N_T} \cdot \Pr\left[|Z_1| \geq 0.05\sqrt{N_T}\right] \geq C\sqrt{N_T}$$

for some universal constant $C > 0$.

Finally, we consider the randomness of $Z_2$ and show that $\mathbb{E}[|Z_1 + Z_2 + \mu|] \geq \frac{C}{2}\sqrt{N_T}$. Applying the tower property of expectations, we have

$$\mathbb{E}_y[|Z_1 + Z_2 + \mu|] = \mathbb{E}\left[\mathbb{E}\left[|Z_1 + Z_2 + \mu| \mid Z_2\right]\right].$$

Consider the following two cases for the conditional expectation inside:

- When $|Z_2 + \mu| \geq \frac{C}{2}\sqrt{N_T}$, we use Jensen's inequality and $\mathbb{E}[Z_1] = 0$ to obtain $\mathbb{E}[|Z_1 + Z_2 + \mu| \mid Z_2] \geq |\mathbb{E}[Z_1 + Z_2 + \mu \mid Z_2]| = |Z_2 + \mu| \geq \frac{C}{2}\sqrt{N_T}$.

- When $|Z_2 + \mu| < \frac{C}{2}\sqrt{N_T}$, we apply the triangle inequality and have

$$\mathbb{E}[|Z_1 + Z_2 + \mu| \mid Z_2] \geq \mathbb{E}[|Z_1|] - |Z_2 + \mu| > C\sqrt{N_T} - \frac{C}{2}\sqrt{N_T} = \frac{C}{2}\sqrt{N_T}.$$

Therefore, regardless of the realization of $Z_2$, we always have $\mathbb{E}[|Z_1 + Z_2 + \mu_0| \mid Z_2] \geq \frac{C}{2}\sqrt{N_T}$. Taking an expectation over the randomness of $Z_2$ gives the desired bound $\mathsf{SSCE}(x, p) \geq \frac{C}{2}\sqrt{N_T}$. $\square$

**Lemma 6.3.** *The stochastic process* $(N_t)_{t \in [T]}$ *satisfies* $\mathbb{E}\left[\sqrt{N_T}\right] \geq \Omega(\mathbb{E}\left[\sqrt{\mathrm{Var}_T}\right]) - O(1)$.

*Proof.* Since $N_T \geq \mathrm{Var}_T/16$ implies $\sqrt{N_T} \geq \sqrt{\mathrm{Var}_T}/4$, we have

$$\sqrt{N_T} \geq \frac{\sqrt{\mathrm{Var}_T}}{4} \cdot \mathbb{1}\left[N_T \geq \frac{\mathrm{Var}_T}{16}\right] = \frac{\sqrt{\mathrm{Var}_T}}{4} - \frac{\sqrt{\mathrm{Var}_T}}{4} \cdot \mathbb{1}\left[N_T < \frac{\mathrm{Var}_T}{16}\right].$$

Therefore, to establish the inequality $\mathbb{E}\left[\sqrt{N_T}\right] \geq \Omega(\mathbb{E}\left[\sqrt{\mathrm{Var}_T}\right]) - O(1)$, it suffices to prove that the expectation of the second term—which we denote with $M$—is upper bounded by $O(1)$, i.e.,

$$M := \mathbb{E}\left[\sqrt{\mathrm{Var}_T} \cdot \mathbb{1}\left[N_T < \mathrm{Var}_T/16\right]\right] \leq O(1). \tag{16}$$

We proceed by partitioning the range of $\mathrm{Var}_T$ into subintervals of geometrically increasing length and enumerating all possibilities for which subinterval $\mathrm{Var}_T$ falls into. If $\mathrm{Var}_T \leq 1$, its contribution to $M$ is clearly $O(1)$. Otherwise, we must have $\mathrm{Var}_T \in [2^l, 2^{l+1})$ for some $l \in \mathbb{N}$, which implies that $N_T < \mathrm{Var}_T/16 < 2^{l-3}$. Therefore, we bound $M$ by taking a union bound over all such $l$'s:

$$M \leq O(1) + \sum_{l \in \mathbb{N}} \mathbb{E}\left[\sqrt{\mathrm{Var}_T} \cdot \mathbb{1}\left[N_T < 2^{l-3} \wedge \mathrm{Var}_T \in [2^l, 2^{l+1})\right]\right]$$

$$\leq O(1) + \sum_{l \in \mathbb{N}} \sqrt{2^{l+1}} \cdot \Pr\left[N_T < 2^{l-3} \wedge \mathrm{Var}_T \geq 2^l\right]. \tag{17}$$

Now we bound $\Pr\left[N_T < k/8 \wedge \mathrm{Var}_T \geq k\right]$ for any fixed value of $k$ (that plays the role of $2^l$) by constructing a sub-martingale. We start by partitioning the time horizon $[T]$ into blocks based on the realized variance $\mathrm{Var}_t$, such that each block $B_j := (b_{j-1}, b_j]$ terminates upon the realized variance $\mathrm{Var}_{B_j}$ first exceeds 1. Formally, using notation $X_{t_1:t_2} := X_{\min\{t_2, T\}} - X_{t_1}$ to denote the increment of any process $(X_t)$ in $(t_1, t_2]$ (with $X_0 = 0$ by default), the endpoints $b_j$ are defined recursively as:

$$b_0 := 0, \; b_j := \min\left\{\infty\right\} \cup \left\{t \in [b_{j-1} + 1, T] \mid \mathrm{Var}_{b_{j-1}:t} \geq 1\right\}, \; \forall j \geq 1.$$

We show in the following lemma that for each block $B_j$, the expected increment $N_{B_j}$ within $B_j$ is lower bounded by a constant as long as $B_j$ terminates before $T$.

**Lemma D.1.** *For the constant $c = 1 - 1/e$, $\mathbb{E}\left[\mathbb{1}\left[N_{B_j} \geq 1\right] - c \cdot \mathbb{1}\left[b_j < \infty\right] \mid \mathbb{F}_{b_{j-1}}\right] \geq 0$.*

We prove Lemma D.1 in Appendix D.2. This lemma justifies that if we define $A_j$ as

$$A_0 := 0, \quad A_j - A_{j-1} := \mathbb{1}\left[N_{B_j} \geq 1\right] - c \cdot \mathbb{1}\left[b_j < \infty\right] \; (j \geq 1),$$

then $(A_j)_{j \geq 0}$ forms a sub-martingale of bounded increment $|A_j - A_{j-1}| \leq 1$, making it unlikely for any $A_j$ to deviate significantly below 0. However, if $N_T < k/8$ and $\mathrm{Var}_T \geq k$, then $A_{k/2}$ must witness a large deviation: on the one hand, block $B_{k/2}$ should terminate properly because the variance in each block cannot exceed 2; on the other hand, $N_T < k/8$ implies that at most $k/8$ of these blocks can have a nonzero increment $N_{B_j}$. As a result,

$$A_{k/2} = \sum_{j=1}^{k} \mathbb{1}\left[N_{B_j} \geq 1\right] - c \cdot \sum_{j=1}^{k} \mathbb{1}\left[b_j < \infty\right] \leq N_T - c \cdot (k/2) < -k/8.$$

By applying the Azuma-Hoeffding inequality for submartingales, we can quantitatively bound the probability of such a large deviation by

$$\Pr\left[N_T < k/8 \wedge \mathrm{Var}_T \geq k\right] \leq \Pr\left[A_{k/2} \leq -k/8\right] \leq e^{-k/64}.$$

Finally, plugging the above bound back into equation (17) gives us

$$M \leq O(1) + \sum_{l \in \mathbb{N}} \sqrt{2^{l+1}} \cdot e^{-2^{l-6}} \leq O(1).$$

We have thus established the inequality (16), which in turn proves the lemma. $\square$

## D.2 Proof of Lemma D.1

Now we prove Lemma D.1, which we restate below.

**Lemma D.1.** *For the constant $c = 1 - 1/e$, $\mathbb{E}\left[\mathbb{1}\left[N_{B_j} \geq 1\right] - c \cdot \mathbb{1}\left[b_j < \infty\right] \mid \mathbb{F}_{b_{j-1}}\right] \geq 0$.*

*Proof.* We first show that for all $t \in [T]$, we have $\Pr\left[n_t = 1 \mid \mathbb{F}_{t-1}\right] \geq p_t^\star(1 - p_t^\star)$, where $\mathbb{F}_{t-1}$ denotes the filtration generated by all the randomness up to time $t - 1$. Note that conditioning on $\mathbb{F}_{t-1}$, $x_t$ is distributed according to Bernoulli$(p_t^\star)$. If the forecaster chooses $p_t \geq \frac{1}{2}$, the condition

$|x_t - p_t| \geq \frac{1}{2}$ holds when $x_t = 0$, which happens with probability $1 - p_t^\star$; otherwise it holds when $x_t = 1$, which happens with probability $p_t^\star$. Therefore, regardless of the choice of $p_t$, we have

$$\Pr\left[n_t = 1 \mid \mathbb{F}_{t-1}\right] = \Pr_{x_t \sim \mathsf{Bernoulli}(p_t^\star)}\left[|x_t - p_t| \geq 1/2\right] \geq \min\{p_t^\star, 1 - p_t^\star\} \geq p_t^\star(1 - p_t^\star).$$

This allows us to invoke Lemma D.5 with $q_t := n_t$, $r_t := p_t^\star(1 - p_t^\star)$, and $\theta = 1$, where we only consider the random process inside block $B_j$. In this context, the stopping time $\tau_1$ corresponds to the end of the block, i.e., $b_j$. Therefore, by applying Lemma D.5 at time step $b_{j-1}$, we obtain

$$A_{b_{j-1}} = \Pr\left[N_{B_j} \geq 1 \mid \mathbb{F}_{b_{j-1}}\right] - \left(1 - e^{-1}\right) \cdot \Pr\left[b_j < \infty \mid \mathbb{F}_{b_{j-1}}\right] \geq 0$$

$$\iff \mathbb{E}\left[\mathbb{1}\left[N_{B_j} \geq 1\right] - c \cdot \mathbb{1}\left[b_j < \infty\right] \mid \mathbb{F}_{b_{j-1}}\right] \geq 0, \text{ where } c = 1 - \frac{1}{e}.$$

$\square$

## D.3 A Stronger Lower Bound

In this section, we state and prove the stronger SSCE lower bound for all forecasters.

**Theorem D.2.** *For any $\mathcal{D} \in \Delta(\{0, 1\}^T)$, $\mathsf{OPT}_{\mathsf{SSCE}}(\mathcal{D}) = \Omega(\mathbb{E}\left[\gamma(\mathrm{Var}_T)\right])$, where the function $\gamma$ is defined as $\gamma(x) := x \cdot \mathbb{1}\left[0 \leq x < 1\right] + \sqrt{x} \cdot \mathbb{1}\left[x \geq 1\right]$.*

*Proof of Theorem D.2.* The theorem holds by combining Lemma 6.2, which lower bounds the SSCE by $\Omega(\sqrt{N_T})$, and the stronger lower bound on $\mathbb{E}\left[\sqrt{N_T}\right]$ shown in Lemma D.3. $\square$

**Lemma D.3.** *There exists a universal constant $C > 0$ such that $\mathbb{E}\left[\sqrt{N_T}\right] \geq C \cdot \mathbb{E}\left[\gamma(\mathrm{Var}_T)\right]$, where the function $\gamma$ is defined as $\gamma(x) := x \cdot \mathbb{1}\left[0 \leq x < 1\right] + \sqrt{x} \cdot \mathbb{1}\left[x \geq 1\right]$.*

*Proof of Lemma D.3.* The proof is also based on partitioning the time horizon into blocks $B_j = (b_{j-1}, b_j]$—each with approximately unit variance—similar to the approach used in proving Lemma 6.3. However, this proof involves a more careful analysis of the growth of $\sqrt{N_t}$ by further grouping blocks into "epochs" and giving special treatment to the first epoch, where the cumulative variance is very small.

Specifically, consider the blocks $B_j = (b_{j-1}, b_j]$ defined by

$$b_0 := 0, \ b_j := \min\{\infty\} \cup \left\{t \in [b_{j-1} + 1, T] \mid \mathrm{Var}_{b_{j-1}:t} \geq 1\right\}, \ \forall j \geq 1.$$

Recall that the increment of $\mathrm{Var}_t$ satisfies $\mathrm{Var}_t - \mathrm{Var}_{t-1} = p_t^\star(1 - p_t^\star) \leq 1/4$. Thus, every block $j$ satisfies $\mathrm{Var}_{B_j} = \mathrm{Var}_{b_j} - \mathrm{Var}_{b_{j-1}} = (\mathrm{Var}_{b_j - 1} - \mathrm{Var}_{b_{j-1}}) + (\mathrm{Var}_{b_j} - \mathrm{Var}_{b_j - 1}) \leq 1 + 1/4 = 5/4$. We further group blocks into epochs such that the $k$-th epoch $\mathcal{T}_k := (\tau_{k-1}, \tau_k]$ contains $\approx 2^k$ blocks:

$$\mathcal{T}_0 := B_1, \quad \mathcal{T}_k := \bigcup_{j \in (2^{k-1}, 2^k]} B_j, \ \forall k \geq 1 \quad \text{(or equivalently, } \tau_k := b_{2^k}\text{)}.$$

In addition, we define $\widetilde{N}_t$ as the sum of $n_s$ capped by 1 in each block:

$$\widetilde{N}_t := \sum_{j : b_j \leq t} \min\{N_{B_j}, 1\} = \sum_{j : b_j \leq t} \mathbb{1}\left[N_{B_j} \geq 1\right].$$

Clearly, for all the realized sequences we have $N_T \geq \widetilde{N}_T$ and $\widetilde{N}_{\tau_k} \leq 2^k$, where the latter is because each block contributes at most 1 to $\widetilde{N}_t$. In the following, we will first analyze the growth of $\sqrt{\widetilde{N}_t}$ in epochs $k \geq 1$, then provide a different analysis for the zeroth epoch.

**In each epoch $\mathcal{T}_k$ with $k \geq 1$.** We start by establishing the following lemma, which extends the characterization of Lemma D.1 into epochs.

**Lemma D.4** (Lower bound on $\widetilde{N}_{\mathcal{T}_k}$)**.** *For any $k \geq 1$, we have*

$$\mathbb{E}\left[\widetilde{N}_{\mathcal{T}_k}\right] \geq 2^{k-2} \cdot \Pr\left[\tau_k < \infty\right].$$

*Proof of Lemma D.4.* According to Lemma D.1, we have that in each block $B_j = (b_{j-1}, b_j]$,

$$\mathbb{E}\left[\mathbb{1}\left[N_{B_j} \geq 1\right] - c \cdot \mathbb{1}\left[b_j < \infty\right]\right] = \mathbb{E}\left[\mathbb{E}\left[\mathbb{1}\left[N_{B_j} \geq 1\right] - c \cdot \mathbb{1}\left[b_j < \infty\right] \,\Big|\, \mathbb{F}_{b_{j-1}}\right]\right] \geq 0,$$

where the first step uses the tower property of expectations, and $c = 1 - \frac{1}{e} \geq \frac{1}{2}$.

Summing over all the blocks in epoch $\mathcal{T}_k$, we obtain

$$\mathbb{E}\left[\widetilde{N}_{\mathcal{T}_k}\right] = \sum_{j=2^{k-1}+1}^{2^k} \mathbb{E}\left[\widetilde{N}_{B_j}\right] = \sum_{j=2^{k-1}+1}^{2^k} \mathbb{E}\left[\mathbb{1}\left[N_{B_j} \geq 1\right]\right] \qquad \text{(Definition of } \mathcal{T}_k \text{ and } \widetilde{N}_t)$$

$$\geq c \cdot \mathbb{E}\left[\sum_{j=2^{k-1}+1}^{2^k} \mathbb{1}\left[b_j < \infty\right]\right] \qquad \text{(Lemma D.1)}$$

$$\geq c \cdot \mathbb{E}\left[\sum_{j=2^{k-1}+1}^{2^k} \mathbb{1}\left[\tau_k < \infty\right]\right] \qquad (b_j \leq b_{2^k} = \tau_k \text{ for all } j \leq 2^k)$$

$$\geq 2^{k-2} \cdot \Pr\left[\tau_k < \infty\right]. \qquad (c \geq 1/2)$$

We have thus established Lemma D.4. $\qquad \square$

With Lemma D.4, we obtain a lower bound by linearizing the increment of $\sqrt{\widetilde{N}_t}$ in each block.

$$\mathbb{E}\left[\sqrt{\widetilde{N}_{\tau_k}} - \sqrt{\widetilde{N}_{\tau_{k-1}}}\right]$$

$$\geq \mathbb{E}\left[\frac{1}{2}\left(\widetilde{N}_{\tau_k}\right)^{-\frac{1}{2}} \cdot \left(\widetilde{N}_{\tau_k} - \widetilde{N}_{\tau_{k-1}}\right)\right] \qquad \text{(Concavity of function } \sqrt{x})$$

$$\geq 2^{-\frac{k}{2}-1} \cdot \mathbb{E}\left[\widetilde{N}_{\tau_k} - \widetilde{N}_{\tau_{k-1}}\right] = 2^{-\frac{k}{2}-1} \cdot \mathbb{E}\left[\widetilde{N}_{\mathcal{T}_k}\right] \qquad (\widetilde{N}_{\tau_k} \leq 2^k)$$

$$\geq 2^{\frac{k}{2}-3} \cdot \Pr\left[\tau_k < \infty\right]. \qquad \text{(Lemma D.4)}$$

The first step above can be alternatively justified by $\sqrt{a} - \sqrt{b} = \frac{a-b}{\sqrt{a}+\sqrt{b}} \geq \frac{a-b}{2\sqrt{a}}$, which holds for all $a \geq b \geq 0$.

**In epoch $\mathcal{T}_0$.** We now analyze $\sqrt{\widetilde{N}_{\mathcal{T}_0}}$ in epoch 0. Note that the $\mathcal{T}_0$ contains only the first block $B_1$, so this value is either 0 or 1, depending on whether there exists a $t \in B_1$ such that $n_t = \mathbb{1}\left[|x_t - p_t| \geq \frac{1}{2}\right] = 1$.

Recall that in the proof of Lemma D.1, we have shown that regardless of the choice of $p_t$,

$$\Pr\left[n_t = 1 \mid \mathbb{F}_{t-1}\right] = \Pr_{x_t \sim \text{Bernoulli}(p_t^\star)}\left[|x_t - p_t| \geq 1/2\right] \geq p_t^\star(1 - p_t^\star)$$

Therefore, in the special case of product distributions (i.e., the sequence $(p_t^\star)$ is deterministic and each outcome $x_t \sim p_t^\star$ is independent of other time steps), we can directly bound the probability that $\sqrt{\widetilde{N}_{\mathcal{T}_0}} = 1$ as follows:

$$\Pr\left[\sqrt{\widetilde{N}_{\mathcal{T}_0}} = 1\right] = 1 - \prod_{t=1}^{\tau_1} \Pr\left[n_t = 0\right] \geq 1 - \prod_{t=1}^{\tau_1}\left[1 - p_t^\star(1 - p_t^\star)\right]$$

$$\geq 1 - \exp\left(-\sum_{t=1}^{\tau_1} p_t^\star(1 - p_t^\star)\right) = 1 - \exp(-\text{Var}_{B_1}) \geq \frac{1}{2}\text{Var}_{B_1},$$

where the last step follows from the inequality $1 - e^{-x} \geq x/2$ when $0 \leq x \leq 5/4$, and the fact that $\text{Var}_{B_1} \leq 5/4$.

However, in the general case where the sequence $(p_t^\star)$ is itself random and depends on the history of $x_t$'s, such a direct argument fails. Instead, we use Lemma D.6 that extends the above analysis to this more general setting. Lemma D.6 is itself a similar but more general statement than Lemma D.5, as it is applicable even when the cumulative variance is smaller than the hard threshold $\theta$. Invoking Lemma D.6 with $q_t := n_t$, $r_t := p_t^\star(1 - p_t^\star)$, and the stopping time $\tau$ as the earlier time step between the end of block $B_1$ and the first time where $n_t = 1$, we have

$$\Pr\left[N_\tau \geq 1\right] \geq 1 - \mathbb{E}\left[e^{-\mathrm{Var}_\tau}\right] \geq \frac{1}{2}\,\mathbb{E}\left[\mathrm{Var}_\tau\right],$$

where the last step again uses $1 - e^{-x} \geq x/2$ for $x \in [0, 5/4]$. Moreover, since $\frac{5}{4} \cdot \mathbb{1}\left[N_\tau \geq 1\right] \geq \mathrm{Var}_{\tau:b_1}$, we also have

$$\Pr\left[N_\tau \geq 1\right] \geq \frac{4}{5}\,\mathbb{E}\left[\mathrm{Var}_{\tau:b_1}\right] \geq \frac{1}{2}\,\mathbb{E}\left[\mathrm{Var}_{\tau:b_1}\right].$$

Combining the two inequalities, we obtain

$$\mathbb{E}\left[\sqrt{\widetilde{N}_{\mathcal{T}_0}}\right] = \Pr\left[N_{B_1} \geq 1\right] \geq \Pr\left[N_\tau \geq 1\right]$$

$$\geq \frac{1}{4}\,\mathbb{E}\left[\mathrm{Var}_\tau + \mathrm{Var}_{\tau:b_1}\right] = \frac{1}{4}\,\mathbb{E}\left[\mathrm{Var}_{B_1}\right]$$

$$\geq \frac{1}{4}\,\mathbb{E}\left[\mathrm{Var}_T \cdot \mathbb{1}\left[\tau_1 = \infty\right]\right]. \qquad (\tau_1 = \infty \implies \mathrm{Var}_T = \mathrm{Var}_{B_1})$$

**Putting everything together.** Combining the lower bounds for epoch $0$ and epochs $k \geq 1$, we obtain

$$\mathbb{E}\left[\sqrt{\widetilde{N}_T}\right] = \mathbb{E}\left[\sqrt{\widetilde{N}_{\mathcal{T}_0}}\right] + \sum_{k \geq 1}\mathbb{E}\left[\sqrt{\widetilde{N}_{\tau_k}} - \sqrt{\widetilde{N}_{\tau_{k-1}}}\right]$$

$$\geq \frac{1}{4}\,\mathbb{E}\left[\mathrm{Var}_T \cdot \mathbb{1}\left[\tau_1 = \infty\right]\right] + \sum_{k \geq 1} 2^{\frac{k}{2} - 3} \cdot \Pr\left[\tau_k < \infty\right]$$

$$= \frac{1}{4}\,\mathbb{E}\left[\mathrm{Var}_T \cdot \mathbb{1}\left[\tau_1 = \infty\right]\right] + \sum_{k \geq 1}\Pr\left[\tau_{k-1} < \infty, \tau_k = \infty\right]\sum_{k' < k} 2^{\frac{k'}{2} - 3}$$

$$\geq \frac{1}{8\sqrt{2}}\,\mathbb{E}\left[\mathrm{Var}_T \cdot \mathbb{1}\left[\tau_1 = \infty\right] + \sum_{k \geq 1}\mathbb{1}\left[\tau_{k-1} < \infty, \tau_k = \infty\right] \cdot 2^{\frac{k}{2}}\right]$$

$$\geq \frac{1}{16}\,\mathbb{E}\left[\mathrm{Var}_T \cdot \mathbb{1}\left[\tau_1 = \infty\right] + \sum_{k \geq 1}\mathbb{1}\left[\tau_{k-1} < \infty, \tau_k = \infty\right] \cdot \sqrt{\mathrm{Var}_T}\right],$$

where the last step follows from the observation that the cumulative variance in each block cannot exceed $2$, so $\tau_k = \infty$ implies that $\mathrm{Var}_T < 2^{k+1}$, i.e., $2^{k/2} \geq \sqrt{\mathrm{Var}_T}/\sqrt{2}$. Finally, since $\tau_1 = \infty$ is equivalent to $\mathrm{Var}_T < 1$, we have established that

$$\mathbb{E}\left[\sqrt{\widetilde{N}_T}\right] \geq \frac{1}{16}\,\mathbb{E}\left[\mathrm{Var}_T \cdot \mathbb{1}\left[\mathrm{Var}_T < 1\right] + \mathbb{1}\left[\mathrm{Var}_T \geq 1\right] \cdot \sqrt{\mathrm{Var}_T}\right] = \frac{1}{16}\,\mathbb{E}\left[\gamma(\mathrm{Var}_T)\right].$$

The lemma follows from the fact that $N_T \geq \widetilde{N}_T$ always holds, which implies $\mathbb{E}\left[\sqrt{N_T}\right] \geq \mathbb{E}\left[\sqrt{\widetilde{N}_T}\right] \geq \frac{1}{16}\,\mathbb{E}\left[\gamma(\mathrm{Var}_T)\right]$. $\qquad\square$

## D.4 Auxiliary Lemmas

**Lemma D.5.** *Let $Q_t = \sum_{s \leq t} q_s$, $R_t = \sum_{s \leq t} r_s$ be two (coupled) stochastic processes such that $q_t \in \{0, 1\}$, $r_t \in [0, 1]$ for all $t \in [T]$. Let $\mathbb{F}_t$ denote the filtration generated by all the randomness up to time $t$. Suppose $r_t$ is a deterministic function on $\mathbb{F}_{t-1}$, and $s_t := \Pr\left[q_t = 1 \mid \mathbb{F}_{t-1}\right] \geq r_t$.*

*For any constant $\theta > 0$, define $\tau_\theta$ to be a stopping time chosen as the first time that $R_t$ reaches $\theta$, i.e.,*

$$\tau_\theta := \min\{\infty\} \cup \{t \in [T] \mid R_t \geq \theta\}.$$

*Let $Q_t^+ := Q_{t:\tau_\theta}$ be the sum of $q_s$ in the future until the stopping time $\tau_\theta$. If $t > \tau_\theta$, then we let $Q_t^+ := 0$. Consider random variables $A_t$'s defined on the filtration $\mathbb{F}_t$ as follows:*

$$A_t := \Pr\left[Q_t^+ \geq 1 \mid \mathbb{F}_t\right] - \left(1 - e^{-(\theta - R_t)}\right) \cdot \Pr\left[\tau_\theta < \infty \mid \mathbb{F}_t\right].$$

*Then we have $A_t \geq 0$ for every $t \leq T$ and every event in $\mathbb{F}_t$.*

*Proof of Lemma D.5.* It suffices to prove the inequality conditioning on events in $\mathcal{F}_t$ that are "atomic" in the sense that they uniquely determine the values of $q_{1:t}$ and $r_{1:t}$. The general case would follow from the law of total probability. In particular, in the following proof, we may view the value of $R_t$ as fixed when we analyze the quantity $A_t$.

We perform a backwards induction from $t = T$ to $t = 0$. Consider the base case of $t = T$. If $R_T \geq \theta$, we have

$$A_T = \underbrace{\Pr\left[Q_T^+ \geq 1 \mid \mathbb{F}_T\right]}_{=0} - \underbrace{\left(1 - e^{-(\theta - R_T)}\right)}_{\leq 0} \cdot \Pr\left[\tau_\theta < \infty \mid \mathbb{F}_T\right] \geq 0.$$

Otherwise when $R_T < \theta$, we have $\Pr\left[\tau_\theta < \infty \mid \mathbb{F}_T\right] = 0$, which also implies $A_T = 0 \geq 0$.

We then assume $A_t \geq 0$, and show that the same holds for $A_{t-1}$, where $t \leq T$. If $R_{t-1} \geq \theta$, we clearly have $A_{t-1} \geq 0$, as the factor $-\left(1 - e^{-(\theta - R_{t-1})}\right)$ would be non-negative. Therefore, it suffices to consider the case that $R_{t-1} < \theta$. In this case, the stopping time $\tau_\theta$ should be $\geq t$, so we have $Q_{t-1}^+ = q_t + Q_t^+$. We bound $A_{t-1}$ by breaking the event $Q_{t-1}^+ \geq 1$ into two cases: either $q_t = 1$, or $q_t = 0$ but $Q_t^+ \geq 1$. We have

$$\Pr\left[Q_{t-1}^+ \geq 1 \mid \mathbb{F}_{t-1}\right] = \Pr[q_t = 1 \mid \mathbb{F}_{t-1}] + \Pr[q_t = 0 \mid \mathbb{F}_{t-1}] \cdot \Pr\left[Q_t^+ \geq 1 \mid \mathbb{F}_{t-1}, q_t = 0\right]$$

$$= s_t + (1 - s_t)\,\mathbb{E}\left[\Pr\left[Q_t^+ \geq 1 \mid \mathbb{F}_t\right] \mid \mathbb{F}_{t-1}, q_t = 0\right]$$

For the second term, we apply the induction hypothesis of $A_t \geq 0$ and get

$$\mathbb{E}\left[\Pr\left[Q_t^+ \geq 1 \mid \mathbb{F}_t\right] \mid \mathbb{F}_{t-1}, q_t = 0\right] \geq \mathbb{E}\left[\left(1 - e^{-(\theta - R_t)}\right) \cdot \Pr\left[\tau_\theta < \infty \mid \mathbb{F}_t\right] \mid \mathbb{F}_{t-1}, q_t = 0\right]$$

$$= \left(1 - e^{-(\theta - R_{t-1} - r_t)}\right) \cdot \Pr\left[\tau_\theta < \infty \mid \mathbb{F}_{t-1}, q_t = 0\right],$$

where the second step uses the fact that conditioning on $\mathcal{F}_{t-1}$, $R_t = R_{t-1} + r_t$. As a result, we obtain

$$\Pr\left[Q_{t-1}^+ \geq 1 \mid \mathbb{F}_{t-1}\right] \geq s_t + (1 - s_t)\left(1 - e^{-(\theta - R_{t-1} - r_t)}\right) \cdot \Pr\left[\tau_\theta < \infty \mid \mathbb{F}_{t-1}, q_t = 0\right]. \quad (18)$$

We also expand the conditional probability $\Pr\left[\tau_\theta < \infty \mid \mathbb{F}_{t-1}\right]$ as follows:

$$\Pr\left[\tau_\theta < \infty \mid \mathbb{F}_{t-1}\right] = s_t \cdot \Pr\left[\tau_\theta < \infty \mid \mathbb{F}_{t-1}, q_t = 1\right] + (1 - s_t)\Pr\left[\tau_\theta < \infty \mid \mathbb{F}_{t-1}, q_t = 0\right]$$

$$\leq s_t + (1 - s_t)\Pr\left[\tau_\theta < \infty \mid \mathbb{F}_{t-1}, q_t = 0\right] \quad (19)$$

Combining the bounds in (18) and (19), we obtain

$$A_{t-1} \geq s_t + (1 - s_t)\left(1 - e^{-(\theta - R_{t-1} - r_t)}\right) \cdot \Pr\left[\tau_\theta < \infty \mid \mathbb{F}_{t-1}, q_t = 0\right]$$

$$- \left(1 - e^{-(\theta - R_{t-1})}\right) \cdot \left(s_t + (1 - s_t)\Pr\left[\tau_\theta < \infty \mid \mathbb{F}_{t-1}, q_t = 0\right]\right)$$

$$= s_t \cdot e^{-(\theta - R_{t-1})} + (1 - s_t) \cdot \left(e^{-(\theta - R_{t-1})} - e^{-(\theta - R_{t-1} - r_t)}\right) \cdot \Pr\left[\tau_\theta < \infty \mid \mathbb{F}_{t-1}, q_t = 0\right]$$

$$\geq e^{-(\theta - R_{t-1})} \cdot \left(s_t \cdot e^{r_t} + 1 - e^{r_t}\right) \qquad \text{(bounding the probability by 1)}$$

$$\geq e^{-(\theta - R_{t-1})} \cdot \left(r_t \cdot e^{r_t} + 1 - e^{r_t}\right) \qquad (s_t \geq r_t \text{ from assumption})$$

$$= e^{-(\theta - R_{t-1}) + r_t} \cdot \left(r_t + e^{-r_t} - 1\right) \geq 0. \qquad (\forall x,\ e^{-x} \geq 1 - x)$$

We have thus proved that the claim also holds for $t - 1$. This completes the induction.

$\square$

**Lemma D.6.** *Let* $Q_t = \sum_{s \leq t} q_s, R_t = \sum_{s \leq t} r_s$ *be two (coupled) stochastic processes such that* $q_t \in \{0, 1\}$, $r_t \in [0, 1]$ *for all* $t \in [T]$. *Let* $\mathbb{F}_t$ *denote the filtration generated by all the randomness up to time* $t$. *Suppose* $r_t$ *is a deterministic function on* $\mathbb{F}_{t-1}$, *and* $s_t := \Pr[q_t = 1 \mid \mathbb{F}_{t-1}] \geq r_t$.

*For any constant* $\theta > 0$, *define* $\tau$ *to be a stopping time chosen as the first time that either* $R_t$ *reaches* $1$ *or* $q_t = 1$, *i.e.,*

$$\tau := \min\{\infty\} \cup \{t \in [T] \mid R_t \geq 1\} \cup \{t \in [T] \mid q_t = 1\}.$$

*Let* $Q_t^+ := Q_{t:\tau}$ *and* $R_t^+ := Q_{t:\tau}$ *be the sum of* $q_s$ *and* $r_s$ *in the future until the stopping time* $\tau$, *respectively. We also let* $Q_t^+ = R_t^+ = 0$ *when* $t > \tau$. *Consider random variables* $A_t$'s *defined on the filtration* $\mathbb{F}_t$ *as follows:*

$$A_t := \Pr\left[Q_t^+ \geq 1 \,\middle|\, \mathbb{F}_t\right] - \mathbb{E}\left[1 - e^{-R_t^+} \,\middle|\, \mathbb{F}_t\right].$$

*Then we have* $A_t \geq 0$ *for every* $t \leq T$ *and every event in* $\mathbb{F}_t$.

*Proof of Lemma D.6.* Using a similar approach to that for Lemma D.5, we prove this claim via a backwards induction from $t = T$ to $t = 0$. Again, we only consider the "atomic" events in $\mathcal{F}_t$ that uniquely determines the values of $q_{1:t}$ and $r_{1:t}$, and thus whether $\tau \leq t$; the general case follows from the law of total probability.

For the base case of $t = T$, we have

$$A_T = \underbrace{\Pr\left[Q_T^+ \geq 1 \,\middle|\, \mathbb{F}_T\right]}_{=0 \text{ as } Q_T^+ \equiv 0} + \underbrace{\mathbb{E}\left[e^{-R_T^+} \,\middle|\, \mathbb{F}_T\right]}_{=1 \text{ as } R_T^+ \equiv 0} - 1 = 0.$$

Now for $t \leq T$, we assume the claim holds for $t$ and analyze $A_{t-1}$. If $\tau \leq t - 1$, we immediately obtain $A_{t-1} \geq 0$ since $Q_{t-1}^+ = R_{t-1}^+ = 0$. It remains to consider the case of $\tau \geq t$. For the first term of $A_{t-1}$ (the conditional probability), we have

$$\Pr\left[Q_{t-1}^+ \geq 1 \,\middle|\, \mathbb{F}_{t-1}\right] = \Pr\left[q_t = 1 \,\middle|\, \mathbb{F}_{t-1}\right] + \Pr[q_t = 0 \mid \mathbb{F}_{t-1}] \cdot \Pr\left[Q_t^+ \geq 1 \mid \mathbb{F}_{t-1}, q_t = 0\right]$$

$$= s_t + (1 - s_t) \Pr\left[Q_t^+ \geq 1 \,\middle|\, \mathbb{F}_{t-1}, q_t = 0\right]$$

$$\geq s_t + (1 - s_t) \cdot \mathbb{E}\left[1 - e^{-R_t^+} \,\middle|\, \mathbb{F}_{t-1}, q_t = 0\right]$$

$$= 1 - (1 - s_t) \cdot \mathbb{E}\left[e^{-R_t^+} \,\middle|\, \mathbb{F}_{t-1}, q_t = 0\right],$$

where the inequality step follows from the induction hypothesis $A_t \geq 0$.

On the other hand, the second term of $A_{t-1}$ (the conditional expectation) can be bounded as

$$\mathbb{E}\left[1 - e^{-R_{t-1}^+} \,\middle|\, \mathbb{F}_{t-1}\right]$$

$$= \Pr\left[q_t = 1 \,\middle|\, \mathbb{F}_{t-1}\right] \cdot \left(1 - e^{-r_t}\right) \qquad\qquad (q_t = 1 \text{ implies } \tau = t \text{ and } R_{t-1}^+ = r_t)$$

$$\quad + \Pr\left[q_t = 0 \,\middle|\, \mathbb{F}_{t-1}\right] \cdot \mathbb{E}\left[1 - e^{-r_t - R_t^+} \,\middle|\, \mathbb{F}_{t-1}, q_t = 0\right]$$

$$= 1 - s_t \cdot e^{-r_t} - (1 - s_t) \cdot \mathbb{E}\left[e^{-r_t - R_t^+} \,\middle|\, \mathbb{F}_{t-1}, q_t = 0\right]$$

$$\geq 1 - \mathbb{E}\left[e^{-r_t - (1 - s_t) R_t^+} \,\middle|\, \mathbb{F}_{t-1}, q_t = 0\right]. \qquad \text{(Jensen's inequality for the convex function } e^{-x})$$

Finally, combining the bounds for both terms of $A_{t-1}$, we obtain

$$A_{t-1} = \Pr\left[Q_{t-1}^+ \geq 1 \,\middle|\, \mathbb{F}_{t-1}\right] - \mathbb{E}\left[1 - e^{-R_{t-1}^+} \,\middle|\, \mathbb{F}_{t-1}\right]$$

$$\geq \mathbb{E}\left[e^{-r_t - (1 - s_t) R_t^+} - (1 - s_t) \cdot e^{-R_t^+} \,\middle|\, \mathbb{F}_{t-1}, q_t = 0\right]$$

$$\geq \mathbb{E}\left[e^{-R_t^+} \cdot \left(e^{-r_t} - (1 - s_t)\right) \,\middle|\, \mathbb{F}_{t-1}, q_t = 0\right] \qquad\qquad (e^{s_t R_t^+} \geq 1)$$

$$\geq \mathbb{E}\left[e^{-R_t^+} \cdot \left(e^{-s_t} - (1 - s_t)\right) \,\middle|\, \mathbb{F}_{t-1}, q_t = 0\right] \qquad\qquad (s_t \geq r_t \text{ by assumption})$$

$$\geq 0. \qquad\qquad (e^{-x} \geq 1 - x, \ \forall x \geq 0)$$

We have proved that $A_{t-1} \geq 0$. As a result, $A_t \geq 0$ for all $t \leq T$ and all events in $\mathcal{F}_t$. $\qquad\square$

# E   Proof of Theorem 1.2

In this section, we prove our main theorem (Theorem 1.2) by combining the theorems established in the previous sections, and then verifying the completeness and soundness of the SSCE.

*Proof of Theorem 1.2.* Let $\mathcal{D} \in \Delta(\{0,1\}^T)$ be an arbitrary distribution and define the random variable

$$\text{Var}_T := \sum_{t=1}^{T} p_t^\star (1 - p_t^\star),$$

over $x \sim \mathcal{D}$, where $p_t^\star := \Pr_{x' \sim \mathcal{D}} \left[ x_t' = 1 \mid x_{1:(t-1)}' = x_{1:(t-1)} \right]$. By Theorems C.1 and D.2, the truthful forecaster gives

$$\text{err}_{\text{SSCE}}(\mathcal{D}, \mathcal{A}^{\text{truthful}}(\mathcal{D})) = O\left( \mathop{\mathbb{E}}_{x \sim \mathcal{D}} [\gamma(\text{Var}_T)] \right),$$

whereas

$$\text{OPT}_{\text{SSCE}}(\mathcal{D}) = \Omega\left( \mathop{\mathbb{E}}_{x \sim \mathcal{D}} [\gamma(\text{Var}_T)] \right).$$

In the above, the $O(\cdot)$ and $\Omega(\cdot)$ notations hide universal constants that do not depend on $\mathcal{D}$. Therefore, there exists a universal constant $c > 0$ such that the SSCE is $(c, 0)$-truthful.

**Completeness.**   Now we verify that the SSCE is complete. For any $x \in \{0,1\}^T$, we have

$$\text{SSCE}(x, x) = \mathop{\mathbb{E}}_{y \sim \text{Unif}(\{0,1\}^T)} \left[ \sup_{f \in \mathcal{F}} \sum_{t=1}^{T} y_T \cdot f(x_t) \cdot (x_t - x_t) \right] = 0.$$

For any $\alpha \in [0,1]$, the upper bound

$$\mathop{\mathbb{E}}_{x_1, \ldots, x_T \sim \text{Bernoulli}(\alpha)} \left[ \text{SSCE}(x, \alpha \cdot \vec{1}_T) \right] = O(\sqrt{T \cdot \alpha \cdot (1 - \alpha)}) = o_\alpha(T)$$

follows from applying Theorem C.1 to the product distribution $\mathcal{D} = \prod_{t=1}^{T} \text{Bernoulli}(\alpha)$ and the fact that $\gamma(x) \leq \sqrt{x}$ for all $x \geq 0$.

**Soundness.**   To show that the SSCE is sound, we first consider the case that $x \in \{0,1\}^T$ is arbitrary and the predictions are $p = \vec{1}_T - x$. Noting that the function $x \mapsto 1/2 - x$ is in the family $\mathcal{F}$ of 1-Lipschitz functions from $[0,1]$ to $[-1,1]$, we have

$$\begin{aligned}
\text{SSCE}(x, \vec{1}_T - x) &= \mathop{\mathbb{E}}_{y \sim \text{Unif}(\{0,1\}^T)} \left[ \sup_{f \in \mathcal{F}} \sum_{t=1}^{T} y_t \cdot f(1 - x_t) \cdot (x_t - (1 - x_t)) \right] \\
&\geq \mathop{\mathbb{E}}_{y \sim \text{Unif}(\{0,1\}^T)} \left[ \sum_{t=1}^{T} y_t \cdot (x_t - 1/2) \cdot (2x_t - 1) \right] \\
&= \mathop{\mathbb{E}}_{y \sim \text{Unif}(\{0,1\}^T)} \left[ \frac{1}{2} \sum_{t=1}^{T} y_t \right] = \frac{T}{4} = \Omega(T),
\end{aligned}$$

where the third step holds since $(x - 1/2) \cdot (2x - 1) = 1/2$ holds for every $x \in \{0,1\}$.

Finally, we fix $\alpha, \beta \in [0,1]$ such that $\alpha \neq \beta$. For fixed $x, y \in \{0,1\}^T$, we have

$$\sup_{f \in \mathcal{F}} \sum_{t=1}^{T} y_t \cdot f(\beta) \cdot (x_t - \beta) = \left| \sum_{t=1}^{T} y_t \cdot (x_t - \beta) \right|.$$

Taking an expectation over $x_1, \ldots, x_T \sim \mathsf{Bernoulli}(\alpha)$ and $y \sim \mathsf{Unif}(\{0,1\}^T)$ gives

$$
\mathop{\mathbb{E}}_{x_1,\ldots,x_T \sim \mathsf{Bernoulli}(\alpha)} \left[ \mathsf{SSCE}(x, \beta \cdot \vec{1}_T) \right] = \mathop{\mathbb{E}}_{x,y} \left[ \sup_{f \in \mathcal{F}} \sum_{t=1}^{T} y_t \cdot f(\beta) \cdot (x_t - \beta) \right]
$$

$$
= \mathop{\mathbb{E}}_{x,y} \left[ \left| \sum_{t=1}^{T} y_t \cdot (x_t - \beta) \right| \right]
$$

$$
\geq \left| \mathop{\mathbb{E}}_{x,y} \left[ \sum_{t=1}^{T} y_t \cdot (x_t - \beta) \right] \right|
$$

$$
= \left| \frac{\alpha - \beta}{2} \cdot T \right| = \Omega_{\alpha,\beta}(T),
$$

where the third step follows from Jensen's inequality $\mathbb{E}\left[|X|\right] \geq |\mathbb{E}\left[X\right]|$. $\qquad \square$

## F    Proof of Lemma 7.1

**Lemma 7.1.** *For any $x \in \{0,1\}^T$ and $p \in [0,1]^T$,*

$$
\mathsf{SSCE}(x, p) \leq \frac{1}{2} \mathsf{smCE}(x, p) + O(\sqrt{T}),
$$

*where the $O(\cdot)$ notation hides a universal constant that does not depend on $T$, $x$ or $p$.*

We prove Lemma 7.1 via a standard chaining argument.

*Proof of Lemma 7.1.* We decompose the SSCE as follows:

$$
\mathsf{SSCE}(x, p)
$$
$$
= \mathop{\mathbb{E}}_{y \sim \mathsf{Unif}(\{0,1\}^T)} \left[ \sup_{f \in \mathcal{F}} \sum_{t=1}^{T} y_t \cdot f(p_t) \cdot (x_t - p_t) \right]
$$
$$
\leq \mathop{\mathbb{E}}_{y \sim \mathsf{Unif}(\{0,1\}^T)} \left[ \sup_{f \in \mathcal{F}} \sum_{t=1}^{T} \left( y_t - \frac{1}{2} \right) \cdot f(p_t) \cdot (x_t - p_t) \right] + \frac{1}{2} \sup_{f \in \mathcal{F}} \sum_{t=1}^{T} f(p_t) \cdot (x_t - p_t).
$$

Note that the second term is exactly $\frac{1}{2}\mathsf{smCE}(x, p)$, so it suffices to bound the first term by $O(\sqrt{T})$.

For notational convenience, let $M_T^{(f)} := \sum_{t=1}^{T} \left( y_t - \frac{1}{2} \right) \cdot f(p_t) \cdot (x_t - p_t)$ for function $f \in \mathcal{F}$. We will establish the following bound for any $N \geq 1$ and functions $f_1, f_2, \ldots, f_N$ from $[0,1]$ to $[-1,1]$:

$$
\mathop{\mathbb{E}}_{y \sim \mathsf{Unif}(\{0,1\}^T)} \left[ \sup_{i \in [N]} M_T^{(f_i)} \right] \leq O(\sqrt{T \log N}). \tag{20}
$$

Assuming Inequality (20), applying Dudley's chaining technique [Dud87] to the $\delta$-covering $\mathcal{F}_\delta$ defined in Lemma C.2 would give

$$
\mathop{\mathbb{E}}_{y \sim \mathsf{Unif}(\{0,1\}^T)} \left[ \sup_{f \in \mathcal{F}} M_T^{(f)} \right] \lesssim \int_0^1 \sqrt{T \log |\mathcal{F}_\delta|} \ \mathrm{d}\delta \qquad\qquad\qquad \text{(chaining)}
$$

$$
\lesssim \sqrt{T} \cdot \int_0^1 \delta^{-\frac{1}{2}} \ \mathrm{d}\delta \qquad (\log |\mathcal{F}_\delta| \leq O(1/\delta) \text{ from Lemma C.2})
$$

$$
\leq O(\sqrt{T}),
$$

which implies the lemma.

Therefore, it remains to establish Inequality (20). We prove this using Hoeffding's inequality and a union bound. For each $i \in [N]$ and every $\varepsilon > 0$, we have

$$\Pr\left[\sup_{i \in [N]} M_T^{(f_i)} \geq \varepsilon\right] \leq \sum_{i=1}^{N} \Pr\left[M_T^{(f_i)} \geq \varepsilon\right] \qquad \text{(union bound)}$$

$$\leq \sum_{i=1}^{N} \exp\left(-\frac{2\varepsilon^2}{\sum_{t=1}^{T}(x_t - p_t)^2 f_i(p_t)^2}\right) \qquad \text{(Hoeffding's inequality)}$$

$$\leq N \cdot \exp\left(-\frac{2\varepsilon^2}{T}\right). \qquad (\|f_i\|_\infty \leq 1, \ \forall i \in [N])$$

Finally, the bound (20) holds by taking an integral over $\varepsilon > 0$: shorthanding $X := \sup_{i \in [N]} M_T^{(f_i)}$, we have

$$\mathbb{E}[X] \leq \int_0^{+\infty} \Pr[X \geq \tau] \ \mathrm{d}\tau \leq \int_0^{+\infty} \min\{N \cdot e^{-2\tau^2/T}, 1\} \ \mathrm{d}\tau = O(\sqrt{T \log N}).$$

This completes the proof. $\qquad\square$

## G  Supplemental Materials for Section 8

We justify the claim in Section 8 that it is impossible for the SSCE (and most natural calibration measures) to incentivize truthful prediction against all adaptive adversaries.

Suppose that the adversary draws $x_1$ from Bernoulli$(1/2)$. If the forecaster predicts $p_1 = 0$, all the subsequent bits are zeros; otherwise, the adversary keeps producing independent samples from Bernoulli$(1/2)$.

Clearly, the truthful forecaster predicts $p_t = 1/2$ at every step $t \in [T]$, and the resulting outcome sequence $x$ is uniform over $\{0,1\}^T$. The resulting SSCE is then $\Theta(T^{1/2})$ in expectation. If the forecaster keeps predicting $p_t = 0$ instead, the expectation of SSCE$(x, p)$ is only $O(1)$. Note that this impossibility holds for any calibration measure CM that satisfies

$$\mathop{\mathbb{E}}_{x_1,\ldots,x_T \sim \text{Bernoulli}(1/2)}\left[\mathsf{CM}_T(x, \vec{1}_T/2)\right] = \omega(1)$$

and

$$\mathop{\mathbb{E}}_{x_1 \sim \text{Bernoulli}(1/2)}\left[\mathsf{CM}_T(x_1 \circ \vec{0}_{T-1}, \vec{0}_T)\right] = O(1),$$

where $\circ$ denotes concatenation.

