# OpenReview forum: "Truthfulness of Calibration Measures"
_NeurIPS.cc/2024/Conference — NeurIPS 2024 poster_

### Official Review · Reviewer_NYRM · 2024-07-12

**Soundness:** 4
**Presentation:** 3
**Contribution:** 3
**Rating:** 7
**Confidence:** 3

**Summary:**

The authors evaluate a variety of existing calibration measures according to their completeness, soundness, along with truthfulness which is their main focus, by theoretically showing how the measures satisfy these aforementioned criteria. They discover that existing measures, despite being complete and sound, have large truthfulness gaps. They introduce a new method, SSCE, that makes a simple modification of smooth calibration error by subsampling time steps, that is also approximately truthful and the authors provide proofs for this.

**Strengths:**

- The paper is the first systematic, theoretical investigation of the truthfulness of a range of calibration measures.

- Extensive and rigorous proofs are provided for all claims.

- The paper introduces a novel calibration measure, the Subsampled Smooth Calibration Error (SSCE), which is both complete and sound while also being approximately truthful.  The proposed SSCE has the potential to be widely used.

**Weaknesses:**

- Given your definition of truthfulness that you base your analysis on, is it consistent with what previous works have defined, and do other works stress the crucialness for this aspect of calibration? I’d like to see more discussion on related works tackling truthfulness in calibration and its importance, particularly to help establish why such an approach like SSCE should be used in practice in the future over ECE or proper scoring rules that have trade offs.


- Despite the introduction claims that Part III of the paper (about the adversarial setting and theorem 1.3) is one of the main contributions, it is not featured prominently in the rest of the main body of the paper, nor is much written about it, so it is difficult to ascertain its importance in the context of the work. It would be beneficial to discuss this more, as most of the paper is devoted to Part II.

- The paper lacks many practical examples which can make it hard to follow at times, particularly when the formulation (of notions such as soundness) differs from previous works. It would be helpful to have more examples such as in [FRST11].

**Questions:**

- What are the main implications of adaptive adversary results? If a sublinear penalty for SSCE is achievable, how would it perform on average or in a typical use case?

- The subsampling time steps technique is applied to smooth calibration error to improve its truthfulness. Can it be more generalized and applied to different calibration measures to improve their truthfulness? Discussing this could help shed light on the generalizability of the approach.

**Limitations:**

yes.

---

> ### Author Rebuttal · Authors · 2024-08-05
>
> **Related works tackling truthfulness.**
>
> As we mention in the paragraph starting on Line 34, to the best of our knowledge, no prior work has systematically investigated or formally defined the notion of truthfulness. Instead, they observed the lack of truthfulness in certain calibration measures, and then used this observation towards the design of algorithms / hard instances for those specific measures. Lack of the systematic study and definition of truthfulness also came as a surprise to us! This is the reason we believe our work makes a compelling contribution to the study of calibration.
>
> For the camera-ready version, we will add more details of how and where truthfulness has played a role in prior work.  At a high level,
> for the ECE, [FH21, QV21] noted that the forecaster can lower their ECE by predicting according to the past. This observation was applied in the algorithm of [FH21] and motivated the "sidestepping" technique in the lower bound proof of [QV21]. For smooth calibration and distance from calibration, [QZ24] used their observation (that both measures can be made $\mathrm{polylog}(T)$ on independent coin flips) as a justification for why a strong lower bound cannot be easily proved on a sequence of random bits.
>
> **Part III of the paper.**
> We will use the extra page afforded by the camera-ready version to provide more details of Part III of our contributions.
> We agree that (while the rest of the paper focuses on showing that the calibration measure we introduce, $\mathsf{SSCE}$, satisfies desirable properties like truthfulness) it is also important to know whether there even exist forecasting algorithms that perform well with respect to our proposed measure, $\mathsf{SSCE}$.
> This is the purpose of Part III, where we show that---just as there exist algorithms guaranteeing $O(\sqrt T)$ $\mathsf{smCE}$---there is an algorithm guaranteeing $O(\sqrt T)$ $\mathsf{SSCE}$.
> At a technical level, our proof here (specifically Theorem 1.3 as proved in Appendix F) uses a key step from prior work [ACRS24]. In particular, we combine their deterministic algorithm, which guarantees $O(\sqrt{T})$ $\mathsf{CalDist}$ (and thus $\mathsf{smCE}$), with our Lemma F.1 that bounds the difference between $\mathsf{smCE}$ and $\mathsf{SSCE}$ via standard chaining techniques.
>
>
> **Implications of adaptive adversary results.**
>
> One particular implication is: In terms of the optimal error, predicting against an adaptive adversary is as easy as predicting a sequence of independent coin flips---it is possible to get an $O(\sqrt{T})$ $\mathsf{SSCE}$ in the former case while, by our Theorem 6.1, the optimal error in the latter case is $\Omega(\sqrt{T})$. This is not the case for every calibration measure; for example, expected calibration error $\mathsf{ECE}$ does not have this property.
> We also note that Theorem 6.1 provides a $O(\sqrt{T})$ bound on $\mathsf{SSCE}$ in the worst-case; in nicer cases, one should achieve lower $\mathsf{SSCE}$.
>
> **Apply subsampling to other measures.**
>
> We believe that understanding the extent to which the subsampling more generally helps with truthfulness is an excellent direction for future work! While, applying this technique more broadly is beyond the scope of this work, let us say a bit more about what we already suspect. Due to the discontinuities of expected calibration error $\mathsf{ECE}$ and maximum swap regret $\mathsf{MSR}$, we believe it to be impossible for our technique of subsampling timesteps to imbue either of the two calibration measures with bounded truthfulness guarantees.
> Our current analysis might still apply though for calibration measures similar to smoothed calibration error but where the Lipschitz function class (see L488) is replaced with other function classes.

---

### Official Review · Reviewer_ZWFw · 2024-07-13

**Soundness:** 3
**Presentation:** 3
**Contribution:** 4
**Rating:** 7
**Confidence:** 4

**Summary:**

The paper initiates the study of truthful calibration measures. It introduces a set of requirements --- truthfulness, completeness, soundness, and asymptotic calibration --- that together define a novel and fruitful set of requirements on calibration measures. It explores the classic sequential binary prediction setting, and proves in that setting both that (1) an entire array of existing calibration metrics fail to satisfy at least one of these requirements, and (2) that a simple novel metric called SSCE (which is a smooth and subsampled calibration measure) actually satisfies all of these, in particular being only a constant factor away from perfect truthfulness.

**Strengths:**

This paper is in my opinion strong both on the modeling (principled definitions leading to nontrivial theory) and on the technical side.

The paper presents a new array of results that evaluate calibration measures under a new angle not explicitly considered before: truthfulness (along with several other dimensions that make this requirement nontrivial, as without further restrictions even constant measures can be truthful as the authors remark). It is very appealing that this work successfully tackles both sides of the issue — on the one hand, it carefully analyzes an entire array of existing (including recent) calibration measures and derives principled bounds showcasing their lack of (optimal) truthfulness, but it also proposes a novel — even if simple — calibration metric that “smooths out” lack of truthfulness in prior measures via subsampling. In this way, it presents a principled framework on which future work could build to both explore and exploit truthful calibrated elicitation.

The technique, with which the main result — O(1) truthfulness of SSCE — is derived, is quite impressive; despite a simple formulation, the proof requires a careful martingale-based argument which necessitates careful twists on standard concentration/anticoncentration/other measure-theoretic results. Based on familiarity with related work, I believe this complexity is fully justified even the “elementary” nature of the statement; and I concur with the authors that this technique may be useful for further work as well.

**Weaknesses:**

Overall, this is a strong paper. It is not a flaw per se given that this is the first paper to model the “truthful” setting — and thus any and all reasonable modeling assumptions are great to have at this stage — but the main question I am not fully certain about is: how principled is the fashion in which “soundness” — i.e. the property that “bad” predictions should lead to linear (in T) calibration measure value — is defined? Meaning, the requirements imposed (that predicting the complement of the truth, or consistently predicting a bias different from the true underlying bias, are both bad) seem necessary but possibly not sufficient to correspond to what may be viewed as “consistently bad predictions”. It would be great if the authors could shed further light on this. (On the other hand, the other dimensions of inquiry (truthfulness/completeness/asymptotic calibration), I would argue, are indeed defined in a very principled way.)

**Questions:**

Can you give some crude but reasonable constant bound on the truthfulness constant c of SSCE? If that is possible, along with an explanation of how this bounding propagates through the outlined proof, that might also serve as additional explanation of the proof pipeline for the reader.

---

> ### Author Rebuttal · Authors · 2024-08-05
>
> **Definition of soundness.**
>
> At a high level, soundness mirrors the completeness and requires that intuitively bad predictions should result in a high calibration error. In this paper, we establish minimal conditions for soundness by considering two particular types of bad predictions, (1) predicting the complement of the true outcomes, and (2) making consistently biased predictions relative to the true mean of the product of Bernoulli distributions. We agree with the reviewer that these conditions are necessary but not sufficient, and that it is an important open direction to explore more principled definitions of soundness. For example, our second requirement can be naturally extended to multiple Bernoulli distributions with different means, where predictions are deemed "bad" if they consistently deviate from the true means.
> However, in our paper, we have chosen to focus on the simplest version, as none of the measures we examined seems to have a problem with soundness.
>
> **Constant factors.**
>
> While we prioritized simplifying the proof over optimizing the constant factors, these constants can be easily tracked. We emphasize that these constants are only for demonstrations and likely can be improved with more careful analysis:
>
> - From Theorem C.1 (Line 897), the ratio between the truthful predictor's expected SSCE and $\mathbb{E}[\gamma(\text{Var}_T)]$ is upper bounded by $(48 + 8\ln 2)\cdot(2 + 2\sqrt{2}) \le 259$.
>
> - From the proof of Lemma 6.2 (Lines 1062, 1063 and 1073), the ratio between the the optimal forecaster's expected SSCE and $\mathbb{E}[\sqrt{N_T}]$ is lower bounded by $0.05\cdot 2^{-7}\cdot (1/2) = 1/5120$. We expect that this ratio can be improved using a more careful analysis of the case where $N_T$ is small.
>
> - From Lemma D.3 (Line 1167), the ratio between $\mathbb{E}[\sqrt{N_T}]$ and $\mathbb{E}[\gamma(\text{Var}_T)]$ is lower bounded by $1/16$.
>
> As a result, the ratio between the expected SSCE of the truthful forecaster and that of the optimal forecaster is upper bounded by $259\cdot 5120\cdot 16 \le 2.2\cdot 10^7$.

---

> > ### Comment · Reviewer_ZWFw · 2024-08-12
> > **Response to Authors**
> >
> > Thank you for your response. Having read it along with the other reviews and your rebuttals to those, I am keeping my positive score and assessment of the paper. A couple clarifying remarks: regarding the constant that I asked about, thank you for providing the quick derivation --- it makes sense to me --- but I would still like to emphasize that if you were to expand your explanation in the rebuttal somewhat, and include it in the paper, that would provide additional ropes for the reader to hold on to while parsing the long and technical proof; I would encourage you to do that. Also, I believe we are in agreement regarding the soundness component, but it might be worth to additionally emphasize the point that this condition is minimal, as in the context of the flow of the paper, the definition comes out almost out of nowhere and might confuse some readers by its unambitiousness relative to the definitions of the other properties.

---

### Official Review · Reviewer_M6FP · 2024-07-13

**Soundness:** 3
**Presentation:** 3
**Contribution:** 3
**Rating:** 6
**Confidence:** 1

**Summary:**

The paper studies calibration measures in a sequential prediction setup. In addition to rewarding accurate predictions (completeness) and penalizing incorrect ones (soundness), the paper formalizes another desideratum of calibration measures — truthfulness. A calibration measure is truthful if the forecaster (approximately) minimizes the expected penalty by predicting the conditional expectation of the next outcome, given the prior distribution of outcomes.

The paper first shows that the existing calibration measures fail to simultaneously meet all these criteria (completeness, soundness, truthfulness). The paper then proposes a new calibration measure — Subsampled Smooth Calibration Error (SSCE) — that is shown to be approximately truthful via a non-trivial analysis.

**Strengths:**

The proposed truthfulness desideratum for the calibration measure is well-motivated. The authors have provided several rigorous technical results to support the need of the proposed calibration measure (SSCE), including the failure of being truthful of existing calibration measures. The analysis and the results are non-trivial.

**Weaknesses:**

The proposed Subsampled Smooth Calibration Error is essentially the expected of the smooth calibration error over the uniformly randomly selected events. This seems to be related to the definition of “Event-conditional unbiasedness” proposed in “High-Dimensional Prediction for Sequential Decision Making”. Could authors comment a bit on this connection?

It is a bit unclear on how to understand the truthfulness in an adversary setting, especially given that in Definition 2.5 (Truthfulness of calibration measures), it requires a distribution over the sequence of the events. It might be helpful to clarify here a bit.

Does the authors have intuitions on how the current results could be extended if one consider multi-outcome space, instead of being binary?

**Questions:**

please see above.

---

> ### Author Rebuttal · Authors · 2024-08-05
>
> **Connection to "event-conditional unbiasedness".**
>
> In our view, the event-conditional unbiasedness in [Definition 2.3, NRRX23] is a strengthening of the notion of "calibration with checking rules" from [FRST11], where each checking rule specifies a subset of time horizon $[T]$ on which the calibration error is evaluated, as discussed in Lines 109--115. Roughly speaking, event-conditional unbiasedness allows each checking rule to be violated up to a degree commensurate with the number of steps that the checking rule applies to. Therefore, event-conditional unbiasedness, like the notion from [FRST11], is of an worst-case flavor as it takes the maximum over the event family. Note that taking the worst case over all $2^T$ events lead to vacuous guarantees due to the large number of events. On the other hand, our proposed notion of SSCE is qualitatively different as it is an average-case notion which takes the expectation over all subsets of the time horizon.
>
> **Truthfulness in an adversarial setting.**
>
> In our view, the truthfulness is a property of a calibration measure, and is independent of whether the measure is used in a stochastic or adversarial setup.
> Intuitively, truthfulness requires that whenever the forecaster has additional side information about the next outcome, they should be encouraged to use it.
> Our current formulation focuses on the "stochastic" scenario where the outcome sequence $y_{1:T}$ is drawn from some prior distribution, and this prior distribution is provided to the forecaster as side information. In this case, the forecaster is able to calculate the ground-truth probability of the next outcome $y_t$ from the history $y_{1:t-1}$, and they should truthfully forecast this ground-truth probability. Arguably, this serves as a minimal condition of truthfulness.
>
> Following our methodology, truthfulness can be naturally defined in adversarial settings as well, which we discuss in Section 7. In the adversarial setting, the next outcome is selected by an adversary, which may depend on both the historical outcomes $y_{1:t-1}$ and the forecaster's previous predictions $p_{1:t-1}$. In this setting, truthfulness would require that whenever the adversary's strategy is provided to the forecaster as side information, they should use that information to truthfully predict the ground-truth probability of $y_t$ under this strategy.
>
> **Extension to multi-outcome spaces.**
>
> Central to our current proofs are the analyses of concentration (specifically, uniform convergence over the Lipschitz class) and anti-concentration. So, we expect that the results can be extended to the multi-outcome spaces via the high-dimensional analogues of such results.
> We agree that the exact analysis and the bounds on the truthfulness for multi-outcome spaces would indeed be an interesting direction for future work.

---

> > ### Comment · Reviewer_M6FP · 2024-08-13
> >
> > I thank the authors' response, I do not have further questions.

---

### Official Review · Reviewer_74Zg · 2024-07-26

**Soundness:** 3
**Presentation:** 1
**Contribution:** 2
**Rating:** 6
**Confidence:** 1

**Summary:**

This paper proposes revisiting the calibration measure by emphasizing three main desirable properties for the metric's behavior: rewarding accurate predictions, penalizing incorrect ones, and ensuring truthfulness. The authors define truthfulness as the ability to accurately predict the conditional expectation of future outcomes. The paper's core findings are twofold: first, it demonstrates that most calibration metrics do not adequately capture truthfulness; second, it argues that a small modification to the classical Expected Calibration Error (ECE)—specifically, subsampling the window in which the ECE is calculated and then averaging the results—is sufficient to meet all the desired properties. The whole article is a series of theoretical arguments supporting these claims.

**Strengths:**

The claims appear to be supported by rigorous mathematical arguments, which, I found,  are also quite lengthy and challenging to follow.

**Weaknesses:**

The theoretical argument for including the notion of truthfulness is intriguing, but I couldn't discern any practical implications for omitting it. At the very least, we would expect some numerical demonstrations, even on simple toy examples.

**Questions:**

- Could you numerically highlight any meaningful difference between smECE and the proposed SSCE?

- Does the introduced subsampling breaks to possibility to easily draw a reliability diagram as in (Blasiok & Nakkiran, 2023)?


- The definition in line 41-42 is crucial to the paper and deserves to be more clearly introduced. It is formalized in page 5. This in general makes the paper quite convoluted (most of the point -even simple argument or definition- are understood after reading a long detour. For a > 50 pages technical paper it is quite painful and I personally just gave up reading it.)

**Limitations:**

Ok

---

> ### Author Rebuttal · Authors · 2024-08-05
>
> **Summary of paper.**
>
> We believe that there is a possible typo or misunderstanding in the summary written by the reviewer---our new calibration measure, $\mathsf{SSCE}$, is obtained from the smooth calibration error ($\mathsf{smCE}$) plus subsampling, rather than from the ECE plus subsampling.
>
> **Practical implications for lack of truthfulness.**
>
> Lines 30--32 include an explanation of the practical importance of truthfulness: If a calibration measure is far from being truthful, the forecaster is incentivized to "overfit" to the measure rather than figuring out the exact distribution of the next outcome. In many cases, this leads to predictions that are far from the true probabilities, and would weaken the trustworthiness of the forecasts.
>
> For practical impact of the lack of truthfulness on a toy example, our informal discussion and calculations in Lines 195--206 as well as the formal proofs in Appendices A.2 and A.3 indeed provide one such example. In short, we gave simple examples on which: (1) the "right" predictions lead to a large error of $\Omega(\sqrt{T})$ or $\Omega(T)$; whereas: (2) a different strategy that frequently makes "wrong" predictions has a much lower error of $0$. The analyses of these examples are indeed simple and elementary and we believe they serve as good examples for demonstrating how lack of truthfulness can lead to highly biased and qualitatively poor predictions.
>
> We can also numerically compute the example on Lines 195---206, say for $T = 1000$ timesteps.
> The expected calibration error $\mathsf{smCE}$ of truthful prediction (average over 100 seeds) is $11.27$; the $\mathsf{smCE}$ of the dishonest prediction strategy is deterministically $0$.
> In contrast, the subsampled smooth calibration error $\mathsf{SSCE}$ of truthful prediction is $8.83$, while the $\mathsf{SSCE}$ of the dishonest prediction strategy is $8.85$.
> That is, the incentive to lie according to the dishonest strategy is near-zero under our proposed calibration error.
>
>
> **Difference between $\mathsf{smECE}$ of [Blasiok-Nakkiran, ICLR'24] and the proposed $\mathsf{SSCE}$.**
>
> We thank the reviewer for pointing us to this calibration measure. We believe that the smooth ECE is qualitatively different from the SSCE, both in terms of their definitions and their truthfulness guarantees. At a high level, the definition of $\mathsf{smECE}$ aims to mitigate the discontinuity of the ECE introduced by the binning; this is in a similar spirit to $\mathsf{smCE}$. However, even after this smoothing, the measure still allows the forecaster to "predict to the past" in an effort to minimize their penalty. On the other hand, our definition of $\mathsf{SSCE}$ introducing an additional amount of randomness by subsampling the horizon, so that the benefit from "predicting to the past" is minimized.
>
> In more detail, in our proof of Proposition A.2, we give a distribution $\mathcal{D}$ on which: (1) truthful prediction typically leads to an $\Omega(\sqrt{T})$ bias at value $1/2$; (2) a different forecasting strategy achieves perfect calibration. Property (2) implies that the optimal error is $\mathsf{OPT}_{\mathsf{smECE}}(\mathcal{D}) = 0$. Property (1), with a simple calculation, shows that truthful prediction leads to an $\Omega(\sqrt{T})$ $\mathsf{smECE}$ (if we scale up the definition in [BN24] by a factor of $T$). In other words, $\mathsf{smECE}$ has a $0$-$\Omega(\sqrt{T})$ truthfulness gap.
>
> **Impact of subsampling on reliability diagrams.**
>
> While the visualization of miscalibration is beyond the scope of this work, we believe that existing methods for producing reliability diagrams (e.g., the one introduced by [BN24]) can easily accommodate the subsampling. It suffices to sample a few independent subsets of the time horizon and generate the reliability diagram corresponding to each subsampled set. Then, we can either stack these diagrams together, or plot the confidence intervals computed from these trials.
>
> **Definition in Lines 41--42.**
>
> We thank the reviewer for the comment, and would like to explain our decision on the organization. The formal definition of truthfulness requires the introduction of many concepts (those in Definitions 2.3, 2.4 and 2.5)
> and thus is relegated to the Preliminaries section. However, we do have informal definitions and explanations in the introduction (with forward pointers to Section 2). We believe this allows the readers to choose their own pace, e.g., either reading the paper in order and gradually building on the formality of the definitions or jumping to formal definitions and back.

---

> > ### Comment · Reviewer_74Zg · 2024-08-12
> >
> > Thanks for the clarifications.
> >
> > - Despite the nice theoretical contributions, a minimal set of experiments illustrating the notion would be beneficial if the authors believe that the *truthfullness* additional metric is useful in practice. In all case, I think it can help the readers understanding.
> >
> > > If a calibration measure is far from being truthful, the forecaster is incentivized to "overfit" to the measure rather than figuring out the exact distribution of the next outcome.
> >
> > I am not sure to understand this point. Perhaps my confusion comes from the fact that I see calibration measure as essentially distances to ground-truth conditional distribution. It is more formally shown in https://arxiv.org/abs/2402.10046
> > As such, as the continuity issue almost surely does not occur, I suspect that same might happen with the truthfulness criteria. The example in Line 195-206 is quite artificial. So for any algorithm with a bit of randomness, these edge cases should not matter .
> >
> > That was why I asked for numerical benchmarks. I would be curious to have authors opinion on that.
> >
> >
> > - Paper Organization
> >
> > Ok, I was not sensitive with authors choice but this is subjective. Unfortunately, I find the paper quite lengthy and hard to follow.
> >
> > >  SSCE is obtained from the smooth calibration error (smCE) plus subsampling, rather than from the ECE plus subsampling.
> >
> > Yes, of course. Thanks for the precision.

---

> > > ### Author Response · Authors · 2024-08-14
> > >
> > > Thank you for the follow up!
> > >
> > > **Regarding “calibration measure as essentially distances to ground-truth conditional distribution”.**
> > >
> > > We think this is a very important point to clarify. Calibration measures are usually defined as a generic function that takes as input a prediction sequence and an outcome sequence (Lines 151–152). Although they are commonly understood as "distances to ground-truth conditional distributions", our work proves this is not the case---calibration can significantly incentivize one to significantly deviate from predicting the ground-truth conditional distribution.
> > >
> > > We also wanted to note that, in adversarial settings for example, there may not be a well-defined notion of an underlying ground-truth conditional distribution. Even in settings where such a conditional distribution does exist, a calibration measure should still be a function of predictions and outcomes, rather than the conditional distribution. As a result, it might not be the case that “the continuity issue almost surely does not occur”.
> > >
> > > **"For any algorithm with a bit of randomness, these edge cases should not matter".**
> > >
> > > We do not see  truthfulness as an edge case similar to the sensitivity of ECE to output perturbations (where even vanishingly small perturbations to predictions can significantly increase ECE), which is the edge case cited in your reference. Even if such perturbations were added to one's predictions (we believe this is what you are referring to as an algorithm "with a bit of randomness"), the lack of truthfulness we observe with smECE still holds; for example the numerical experiment we provided in our experiment remains unchanged if one adds vanishingly small perturbations. We do not believe there is any dimension along which perturbing the setup of the example in Lines 195-206 significantly changes the observed truthfulness problem---as such we do not believe it to be an artificial edge case.

---

> > > > ### Comment · Reviewer_74Zg · 2024-08-14
> > > >
> > > > Thanks for the insightful discussions.
> > > >
> > > > As a final note, I still think numerical experiments are needed in this paper.
> > > >
> > > > Based on authors clarification, I will raise my score accordingly.

---

### Decision · Program_Chairs · 2024-09-25

**Decision:**

Accept (poster)

**Comment:**

All reviewers agree that this is a solid paper.